# Do Neural Networks Learn Similar Subspaces? An Empirical Exploration of Joint Parametric Subspaces in Deep Neural Networks

## Abstract

We show that deep neural networks trained across diverse tasks exhibit remarkably similar low-dimensional parametric subspaces. We provide the first large-scale empirical evidence that demonstrates that neural networks systematically converge to shared spectral subspaces regardless of initialization, task, or domain. Through mode-wise spectral analysis of over 1100 models - including 500 Mistral-7B LoRAs, 500 Vision Transformers, and 50 LLaMA-8B models - we identify universal subspaces capturing majority variance in just a few principal directions. By applying spectral decomposition techniques to the weight matrices of various architectures trained on a wide range of tasks and datasets, we identify sparse, joint subspaces that are consistently exploited, within shared architectures across diverse tasks and datasets. Our findings offer new insights into the intrinsic organization of information within deep networks and raise important questions about the possibility of discovering these universal subspaces without the need for extensive data and computational resources. Furthermore, this inherent structure has significant implications for model reusability, multi-task learning, model merging, and the development of training and inference-efficient algorithms, potentially reducing the carbon footprint of large-scale neural models.

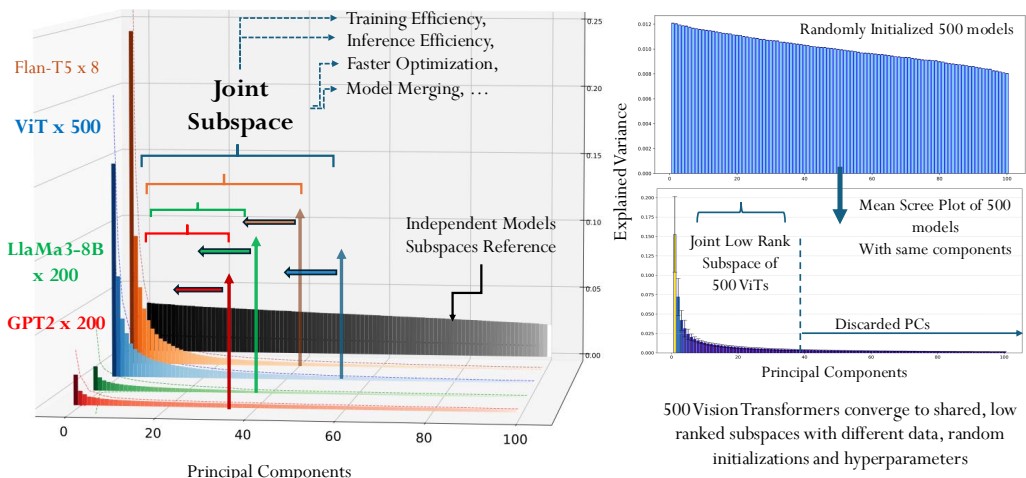

Figure 1: **Deep Networks Converge to Shared, Low-Rank (Universal) Subspaces.** Across distinct architectures and modalities, neural networks systematically learn to operate within remarkably similar low-dimensional parameter subspaces. **Left:** Principal component analysis of 200 GPT2, 500 Vision Transformers, 50 LLaMA-8B, and 8 Flan-T5 models reveals consistent sharp spectral decay - strong evidence that a small number of weight directions capture dominant variance despite vast differences in training data, objectives, and initialization. The black baseline (independent subspaces reference) represents the naive expectation that models would learn distinct directions; our empirical findings contradict this. **Right:** Strikingly, 500 randomly initialized ViT models converge to a common low-rank subspace, demonstrating this is a fundamental neural network property. This emergent structure unlocks powerful applications: parameter-efficient adaptation, efficient model merging, compressed storage, and accelerated training and inference. Further discussion in Section A.

# 1 INTRODUCTION

We show that backpropagated neural networks trained on a variety of datasets - which could be disjoint and unrelated - diverse hyper-parameter settings, initializations and regularization methods, often learn an architecture-specific, layer-wise similar, low-rank joint subspaces (we refer to this as the Universal Subspace). We provide the first large-scale empirical analysis - across a diverse set of models - that neural networks tend to converge to these joint subspaces, largely independent of their initialization or the specific data used for training. Our study encompasses different model architectures trained on a variety of datasets, sometimes with different loss functions and tasks. Our spectral subspace analysis of the weights of all these models (Figure 1) suggests that although individual tasks appear to induce distinct subspaces, individually, they are all part of an unusually low-ranked joint subspace. Our work extends the scientific community's understanding of what neural networks learn. This universality could explain several puzzling neural properties: why overparameterized models with millions more parameters than training samples still generalize; how different initializations converge to similar representations; and why techniques like weight sharing and parameter-efficient fine-tuning succeed across architectures. If networks indeed learn within shared subspaces, this would provide a supporting explanation for implicit regularization, transferability, and the effectiveness of sparse training methods, while also opening up avenues for applications like efficient merging, new optimization techniques, faster and more efficient learning and inference.

Several works have hinted at phenomena consistent with our joint (universal) subspace hypothesis. For example, Neural Tangent Kernel (NTK) theory demonstrates that, in the infinite-width limit, the training dynamics of deep networks are governed by a kernel that is largely invariant to task specifics (Jacot et al., 2018). Similarly, research in mechanistic interpretability's own universality hypothesis (Olah et al., 2020; Chughtai et al., 2023) has uncovered recurring circuits and patterns within some layers of toy or vision networks, lending indirect support to the universality hypothesis. Other works, including the lottery ticket hypothesis (Frankle & Carbin, 2019) and studies on mode connectivity (Garipov et al., 2018), provide further evidence for the existence of reusable, low-dimensional representations in neural networks. Notably, Krizhevsky et al. (2012) observed that the first layer of convolutional networks tends to learn Gabor-like filters across various vision tasks. Recent studies by (Guth & Ménard, 2024; Guth et al., 2024) have also shown initial evidence of recurring eigenvectors for some layers of convolutional neural networks trained on natural images.

In our analysis, we present compelling empirical evidence for the existence of universal subspaces within LoRA adapters across different modalities and tasks. We initially focus on LoRA adapters due to their ease of training and the ability to collect a large number of adapters for diverse tasks, models, and datasets, which enables robust evaluation of our hypothesis. E.g., we demonstrate the emergence of a universal subspace across approximately 500 LoRA adapters for the Mistral-7B (Jiang et al., 2023) model. We further extend our investigation to the full weight space, where we observe similar universality, extracting sparse, low-rank universal subspaces from about 500 Vision Transformer models and 50 LLaMA3-8B models, each trained on different datasets and initializations.

Although the underlying causes and broader implications of this universal property remain an open area of investigation, even an initial understanding of parameter subspace universality has profound implications for neural network efficiency and interpretability. Shared subspaces could enable: (1) massive model compression by storing only subspace coefficients rather than full weights; (2) rapid adaptation to new tasks within learned subspaces; (3) theoretical insights into generalization bounds and optimization landscapes; and (4) environmental benefits through reduced computational requirements for training and inference.

The remainder of this paper is organized as follows. We first define the problem set up formally in Section 2 followed by listing of essential properties and conditions with corresponding empirical justifications. Section 3.3.1 proposes the method to adapt to new tasks leveraging the shared approximate universal subspace. Section 3.1 explains our analysis methodology and Section 3.2 presents the comprehensive empirical evidence of the Universal subspaces. Section 4 briefly discusses the analysis providing useful insights and answers the fundamental questions raised in the introduction. We discuss related work in appendix A.1 and discuss limitations and scope for future work in Section 5. Our primary contributions include

- We empirically demonstrate the existence of a lower-dimensional shared universal subspace in backpropagated neural networks, and also provide relevant theoretical analysis.
- Illustrate the approach to learning an approximate low-dimensional shared subspace using the available set of tasks. Propose conditions for convergence of this learned subspace to the true universal shared subspace.
- Reuse the learned shared subspace to efficiently adapt to new unseen tasks with significantly fewer of trainable parameters. Our experiments across a wide variety of large pretrained models across various architectures and data modalities extensively verify and validate our hypothesis and theoretical findings.

## 2 NOTATIONS, DEFINITIONS AND THEORETICAL ANALYSIS

Our theoretical analysis models predictors as elements of a Hilbert space, for example a reproducing kernel Hilbert space (RKHS), while our experiments are conducted with practical large-scale models such as transformers and LoRA-based variants. Modeling predictors in a Hilbert space (kernel) framework is standard when analyzing aspects such as generalization and inductive bias of modern deep architectures, and has been widely used to approximate or interpret the behavior of large neural networks in practice (Ortiz-Jimenez et al., 2023; Wei et al., 2019; Chen & Xu, 2021; Belfer et al., 2024; Bietti et al., 2019). We aim to understand whether the shared structure across tasks can be consistently recovered from data as number of tasks increase. Specifically, each task has an associated ground-truth predictor $f_t^\star$, and we are interested in the covariance (second-moment) operator $\mathcal{S}$ that captures the common subspace spanned by these predictors. Since in practice we only observe finite samples per task and learn approximate predictors $\hat{f}_t$, two sources of error arise: (i) variability due to having finitely many tasks, and (ii) estimation noise within each task. Our goal is to establish conditions under which the empirical operators built from $\hat{f}_t$ concentrate around $\mathcal{S}$, and to show that the learned top-$k$ subspace converges to the true one, with convergence rates that separately reflect the number of tasks and the accuracy of per-task learning.

**Setup.** Let $(\mathcal{H}, \langle \cdot, \cdot \rangle)$ be a separable Hilbert space with norm $\|\cdot\| = \|\cdot\|_{\mathcal{H}}$. For $a, b \in \mathcal{H}$, the rank-one operator $a \otimes b : \mathcal{H} \to \mathcal{H}$ is $(a \otimes b)g = \langle b, g \rangle \, a$; in particular $\|a \otimes b\|_{\text{op}} = \|a\| \, \|b\|$. Tasks $t = \{1, 2, 3 ..., T\}$ are drawn i.i.d. from distribution $\mathcal{T}$ and each task dataset $S_t = \{(x_{t,i}, y_{t,i})\}_{i=1}^{n_t}$ with $n_t$ samples is drawn independently from $D_t$. Let $f_t^\star \in \mathcal{H}$ denote the (unknown) ground-truth predictor for task $t$ and $\hat{f}_t \in \mathcal{H}$ be the learned predictor for the task.

**Definition 2.1** (Task second-moment operator). The *population*, *true empirical*, and *learned empirical* task second-moment operators are respectively,

$$\mathcal{S} := \mathbb{E}_{t \sim \tau}[f_t^\star \otimes f_t^\star], \qquad \hat{\mathcal{S}} := \frac{1}{T} \sum_{t=1}^{T} f_t^\star \otimes f_t^\star, \qquad \tilde{\mathcal{S}} := \frac{1}{T} \sum_{t=1}^{T} \hat{f}_t \otimes \hat{f}_t.$$

where $\mathcal{S}, \hat{\mathcal{S}}, \tilde{\mathcal{S}}$ are self-adjoint and positive semi-definite such that $\text{tr}(\mathcal{S}) < \infty$. Its top-$k$ eigenspace $\mathcal{H}_k^\star$ is the population rank-$k$ *shared subspace* of tasks.

*Remark 2.2.* We work with the second-moment operator (rather than centered covariance), so the top eigenspace may include the mean direction of $\{f_t^\star\}_{t \sim \mathcal{T}}$.

Let $\lambda_1 \geq \lambda_2 \geq \cdots$ be the eigenvalues of $\mathcal{S}$ with orthonormal eigenvectors $\{\phi_i\}_{i \geq 1}$. Write $P_k = \sum_{i=1}^{k} \phi_i \otimes \phi_i$ for the projector onto the population top-$k$ subspace $\mathcal{H}_k^\star = \text{span}\{\phi_1, \ldots, \phi_k\}$, and let $\tilde{P}_k$ be the projector onto the top-$k$ eigenspace of $\tilde{\mathcal{S}}$ (the learned shared subspace). Define the eigengap $\gamma_k := \lambda_k - \lambda_{k+1} > 0$.

**Assumption 2.3** (Realizability, bounded second moment and effective rank). *For a constant $B > 0$ and for all tasks, $f_t^\star \in \mathcal{H}$ almost surely, $\|f_t^\star\| \leq B$ a.s., $\mathbb{E}_{t \sim \tau} \|f_t^\star\|^2 = \text{tr}(S) < \infty$. In addition, $\mathcal{S}$ has bounded effective rank, $\frac{tr(\mathcal{S})}{\|\mathcal{S}\|_{\text{op}}} \leq \kappa$*

Assumption 2.3 ensures that all ground-truth predictors are bounded and have finite second moment, so the population covariance operator $S$ is well-defined. The bounded effective rank condition further guarantees that the shared structure of the tasks is not arbitrarily infinite-dimensional, making subspace recovery feasible.

**Assumption 2.4** (Per-task estimation accuracy in $\mathcal{H}$). *For any $\delta_t \in (0,1)$ with probability at least $1 - \delta_t$ over the draw of $S_t$,*

$$\left\| \hat{f}_t - f_t^\star \right\| \leq \eta_t, \quad ...where \ \eta_t = \mathcal{R}_{n_t, D_t}(\mathcal{H}) + \sqrt{\frac{\ln(1/\delta_t)}{2n_t}}$$

*Here $\mathcal{R}_{n_t, D_t}(\mathcal{H})$ represents Rademacher complexity of the solutions within Hilbert space $\mathcal{H}$ over $n_t$ samples drawn i.i.d. from $D_t$ This form is satisfied, for example, by strongly convex regularized ERM in an RKHS (e.g., kernel ridge regression or NTK ridge), under bounded kernel norm and sub-Gaussian response noise (Bartlett & Mendelson, 2003).*

Assumption 2.4 requires that each task predictor $\hat{f}_t$ is learned accurately from its finite dataset. In other words, $\hat{f}_t$ is close to the true $f_t^\star$ in $\mathcal{H}$-norm with high probability, at a rate governed by sample size and complexity of the hypothesis space.

**Theorem 2.5** (Two-level convergence to the shared subspace). *Assume 2.3–2.4. Let $c_1, c_2$ be any absolute constants. For any $\delta \in (0,1)$, choose $\delta_t = \delta/(2T)$ and set $\delta_T = \delta/2$. With probability at least $1 - \delta$ (over tasks and all per-task samples),*

$$\left\| \tilde{\mathcal{S}} - \mathcal{S} \right\|_{\text{op}} \leq c_1 B^2 \sqrt{\frac{\log(c_2/\delta)}{T}} + (2B\bar{\eta} + \overline{\eta^2}) \tag{1}$$

*If moreover $\gamma_k > 0$, then*

$$\left\| \tilde{P}_k - P_k \right\|_{\text{op}} \leq \frac{2}{\gamma_k} \left( c_1 B^2 \sqrt{\frac{\log(c_2/\delta)}{T}} + (2B\bar{\eta} + \overline{\eta^2}) \right). \tag{2}$$

*where $\bar{\eta} = \frac{1}{T} \sum_{t=1}^{T} \eta_t$, $\overline{\eta^2} = \frac{1}{T} \sum_{t=1}^{T} \eta_t^2$ and $\eta_t$ is defined same as in assumption 2.4*

Proof of Theorem 2.5 can be found in appendix Section A.2. The Theorem 2.5 shows that the empirical second-moment operator built from the learned predictors converges to the true operator $\mathcal{S}$, and the learned top-$k$ subspace $\hat{P}_k$ converges to the true subspace $P_k$. The rates capture two sources of error: averaging across tasks (scaling with $1/\sqrt{T}$) and per-task estimation errors (through $\bar{\eta}$ and $\overline{\eta^2}$). A larger eigengap $\gamma_k$ makes the subspace recovery more stable. In practice, we obtain the eigenvectors of $\tilde{\mathcal{S}}$ using HOSVD (Higher-Order Singular Value Decomposition) of the concatenated weight matrix $\mathcal{X}$ highlighted in Section 3. Motivated by our theoretical analysis, we try to approximate $\hat{\mathcal{S}}$ for a set of tasks by extracting principal directions from as many trained models as possible.

## 3 ANALYSIS

### 3.1 ANALYSIS METHODOLOGY

Since there is no current method that enables us to compare subspaces of models with different architectures, we focus on large number of models trained on the same architecture. To this end, we perform analysis using Low rank adapters (Hu et al., 2021) (LoRA) as well as classical weights of transformer and CNN (Convolutional Neural Network) architectures. For all our experiments, unless stated otherwise, we perform Order 1-2 HOSVD only, to ensure that our methodology works even in the simplest case. Algorithm 1 provides the algorithm we implement. Refer to Section B for discussion regarding secondary subspace and how to choose the number of top components.

### 3.2 RESULTS FROM JOINT SUBSPACES' ANALYSIS

We present empirical results using method shown in Section 3.1, extracting our layer wise universal subspace approximations using thousands of publicly available models for most of our experiments. This choice allows us to have *no training costs* whatsoever, for extracting the universal subspace. Spectral analysis relies on efficient spectral decomposition libraries, and can even be run on CPUs. We run all our analysis and experiments on one Nvidia A5000 GPU. The presented large scale empirical results forms the crux of our work and provide strong evidence for the presence of such

---

**Algorithm 1** Truncated Zero-Centered Higher-Order SVD (HOSVD)

---

**Require:** A high-order tensor $\mathcal{X} \in \mathbb{R}^{I_1 \times \cdots \times I_N}$ constructed by stacking $N$ rank-$r_n$ task matrices along mode $n$, where $1 \leq r_n \leq I_n$ and $n \in [1, N]$.

**Ensure:** Mean tensor $\boldsymbol{\mu}$; factor matrices $U^{(n)} \in \mathbb{R}^{I_n \times \hat{r}_n}$ (orthonormal columns), where $\hat{r}_n$ is chosen as the smallest number of left singular vectors whose cumulative explained variance is at least $\tau$; and the truncated core tensor $\mathcal{S} \in \mathbb{R}^{\hat{r}_1 \times \cdots \times \hat{r}_N}$. Reconstruction is given by $\widehat{\mathcal{X}} = \boldsymbol{\mu} + \mathcal{S} \times_1 U^{(1)} \cdots \times_N U^{(N)}$, where $\times_n$ denotes mode-$n$ tensor–matrix multiplication.

1: **Zero-centering:** $\boldsymbol{\mu} \leftarrow \text{mean}(\mathcal{X})$         ▷ elementwise mean over all entries
2: $\mathcal{X}_c \leftarrow \mathcal{X} - \boldsymbol{\mu}$         ▷ broadcast $\boldsymbol{\mu}$ to the shape of $\mathcal{X}$
3: **for** $n = 1$ **to** $N$ **do**
4:     $X_{(n)} \leftarrow \text{unfold}(\mathcal{X}_c, n)$         ▷ mode-$n$ matricization; $X_{(n)} \in \mathbb{R}^{I_n \times \prod_{m \neq n} I_m}$
5:     Compute thin SVD: $X_{(n)} = \tilde{U}^{(n)} \Sigma^{(n)} \tilde{V}^{(n)\top}$
6:     $U^{(n)} \leftarrow \tilde{U}^{(n)}(:, 1:\hat{r}_n)$         ▷ keep first $\hat{r}_n$ left singular vectors (variance $\geq \tau$)
7: **end for**
8: **Truncated core:** $\mathcal{S} \leftarrow \mathcal{X}_c \times_1 U^{(1)\top} \times_2 U^{(2)\top} \cdots \times_N U^{(N)\top}$
9: **return** $\boldsymbol{\mu}$, $\{U^{(n)}\}_{n=1}^N$, $\mathcal{S}$         ▷ Optionally compute $\widehat{\mathcal{X}} = \boldsymbol{\mu} + \mathcal{S} \times_1 U^{(1)} \cdots \times_N U^{(N)}$

---

low ranked joint subspaces across a wide range of task, architecture and modalities. In summary, we present a total of **eight** set of analysis and applications, including tasks like image classification, natural language understanding, text to image generation, model merging, etc for different model architectures and modalities.

### 3.2.1 LOWER-RANK JOINT SUBSPACES IN CNNS, LORA AND FINETUNED MODELS

In smaller and conventional architectures such as CNNs, evidence for universal structure has been more limited but suggestive. Early work observed that the first convolutional layer often learns Gabor-like filters across diverse vision tasks (Krizhevsky et al., 2012). More recently, works report recurring eigenvectors in certain CNN layers trained on natural images (Guth et al., 2024; Guth & Ménard, 2024).

We extend these observations and examine whether a shared low-rank joint subspace emerges across tasks. Specifically, we train ResNet-50 models from random initialization for image classification on five disjoint datasets (CIFAR-10, CIFAR-100, ImageNet, Oxford-IIIT Pets, and EuroSAT), ensuring no overlap in samples. While our theoretical analysis indicates that a small number of models may lead to an under-approximation of the joint universal subspace, training CNNs from scratch at scale constrains the number of models we can include in this study.

(a) Comparison of model performance across datasets.

| Method | ImageNet | EuroSat | CIFAR-10 | CIFAR-100 | Oxford Pets | Avg |
|---|---|---|---|---|---|---|
| ResNet50 | 80.86 | 98.96 | 97.35 | 83.82 | 93.48 | 90.89 |
| Universal R50 | 77.89 | 98.83 | 95.89 | 81.49 | 83.81 | 87.58 |

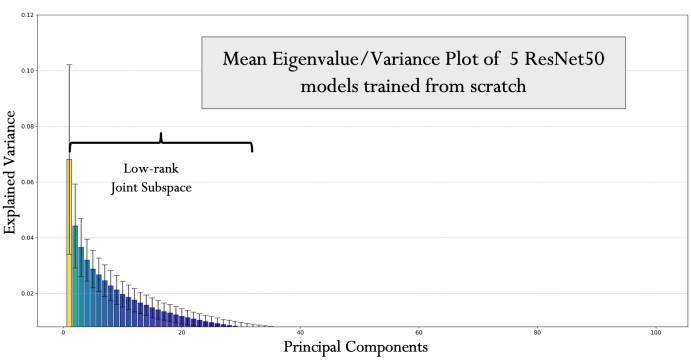

(b) Summarized (averaged for all layers) eigenvalue plot of all model weights corresponding to all 31 layers of 5 ResNet50 models. Mean refers to the fact that it has been averaged for all layers for conciseness. The vertical axis is Explained Variance (for *all* models) and X axis indicated Principal Components. We will follow this setup throughout the paper. We also refer to the low-ranked shared subspace as 'Universal' subspace and may refer to a specific model consisting of extracted basis as the 'Universal variant'.

Figure 2: **Proving existence of universal subspaces in CNNs.** Decomposing 5 ResNet50 models trained on different tasks shows the emergence of a low rank, universal subspace where the majority of the information is present in only 16 (or fewer) distinct subspace directions for all layers of the network.

Despite these limitations, Figure 2b reports the average explained variance across all layers of ResNet-50 and reveals a distinct, shared low-rank structure spanning these disjoint tasks. Moreover, even when the estimated universal subspace is relatively coarse, projecting to this subspace to obtain a low-rank ResNet-50 (thereby reducing parameters) preserves competitive performance relative to full fine-tuning, further supporting the presence and utility of a joint subspace (2a).

In order to conduct a more real-world experiment, we choose to run the subspace analysis for LoRA Hu et al. (2021) models simply because they are available in abundance in public domain. Given LoRA models distinctly capture task specific directions as they show weak alignment with the original weights Hu et al. (2021), they form a good main model parameter alternative to run our subspace analysis and verify whether this holds true. We spectrally decompose (Section 3.1) LoRA's submatrices individually, each concatenated across all the available finetuned LoRAs and choose top $k$ spectral basis. This setup allows us to truly stress test the Universal Subspace.

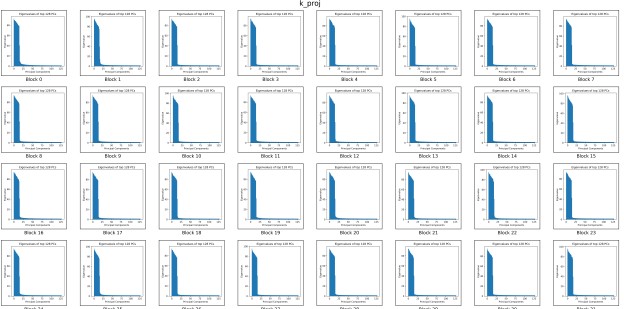

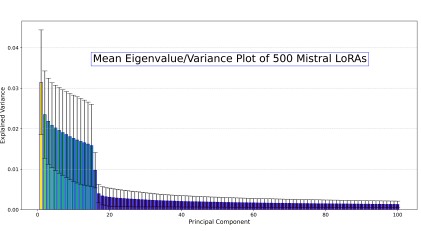

(a) Eigenvalue/Variance plot for Orthogonal Spectral Components for 500 unique LoRAs of different layers of Mistral-7B model

(b) Summarized eigenvalue plot of all LoRAs corresponding to all 31 layers of all 500 Mistral 7B models

Figure 3: **Proving existence of universal subspaces in deep networks.** Decomposing 500 sets of LoRAs trained on different tasks using the Mistral-7B model shows the emergence of a low rank, universal subspace where the majority of the information is present in only 16 (or less) distinct subspace directions for all layers of the network. Plots of other layers are present in the Section B.1.

We first study **500 LoRA models** trained on distinct Natural Instructions (Wang et al., 2022) using Mistral-7B-Instruct-v0.2 (Jiang et al., 2023) as the base (Brüel-Gabrielsson et al., 2024). Each LoRA has at least rank 16. Figure 3 shows that the top spectral components capture most of the variance in each layer, indicating a low-rank structure shared across tasks. Figure 3a visualizes the eigenvalue decay per layer, while Figure 3b summarizes the pattern across all layers and models.

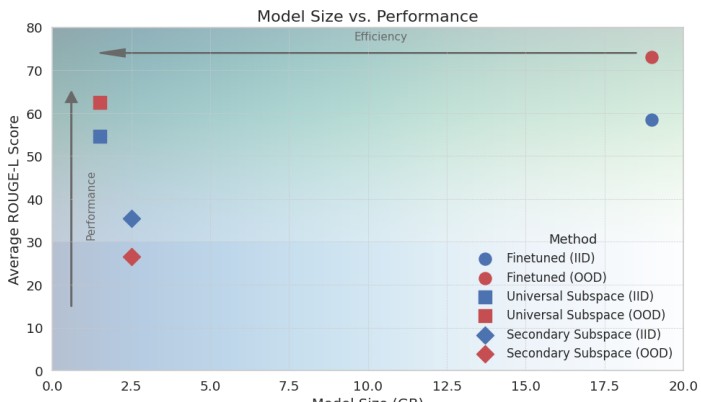

Figure 4: Lots of LoRAs Model Size vs Performance plot.

To test subspace expressiveness, we reconstruct LoRA weights for both seen (IID) and unseen (OOD) tasks by projecting them into the universal subspace. As shown in Figure 4, the reconstructed models retain high performance in both cases. In contrast, projection into the residual *Secondary Subspace* leads to a sharp performance drop, underscoring the importance of the principal subspace. Our method is also **19× more memory-efficient**, as it eliminates the need to store all 500 LoRAs.

We extend our analysis to **text-to-image generation** using Stable Diffusion-XL (Podell et al., 2023). A universal subspace is extracted from publicly available LoRAs on HuggingFace (von Platen et al.,

2022). When projecting individual LoRAs into this subspace, the resulting generations preserve visual quality and style (Figure 5). CLIP-based evaluations (Table 1) show that the universal subspace even outperforms individual LoRAs in some cases, possibly due to denoising effects previously observed in (Sharma et al., 2023).

Table 1: CLIP scores (higher is better) of images generated using SDXL.

| Method | Style 1 | Style 2 | Style 3 | Style 4 | Style 5 | Style 6 | Style 7 | Style 8 | Style 9 | Style 10 | Avg |
|---|---|---|---|---|---|---|---|---|---|---|---|
| LoRA | 21.95 | 15.59 | 22.18 | 18.84 | 16.65 | 17.99 | 24.66 | 17.47 | 22.07 | 19.93 | 19.73 |
| Universal SDXL LoRA | 21.96 | 16.07 | 22.07 | 18.79 | 16.68 | 17.99 | 24.66 | 17.56 | 22.46 | 20.09 | **19.83** |

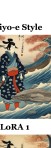 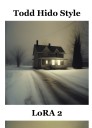 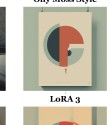 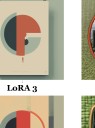 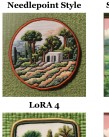 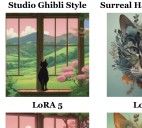 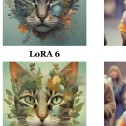 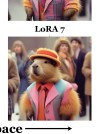 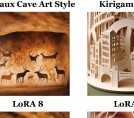 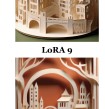 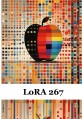

Figure 5: Text-to-Image Generation Results for Individual models vs. our Universal Subspace model. We notice no visual reduction in style quality despite significant reduction in total model size.

In order to test the ability of condensing many models into a single universal subspace, we compare our method with SOTA model merging/combination methods in Table 2. We compare our universal subspace inspired combination approach against six state-of-the-art, gradient-free baselines: RegMean (Jin et al., 2023), Task Arithmetic (TA) (Ilharco et al., 2023), TIES (Yadav et al., 2023), DARE-TIES (Yu et al., 2024), KnOTS-TIES, and KnOTS-DARE-TIES (Stoica et al., 2025). RegMean aligns task-specific updates by solving a layer-wise linear regression problem, requiring transformation matrices for each model. TA merges models by linearly combining parameters, but relies on tuning scaling coefficients on a validation set for optimal performance. TIES extends TA with magnitude-based pruning and sign conflict resolution, introducing additional hyperparameters such as pruning thresholds, while DARE-TIES combines random Bernoulli pruning with TIES' sign resolution, also requiring tuning of pruning probability. KnOTS-TIES and KnOTS-DARE-TIES further apply SVD-based subspace alignment before merging, but still inherit the need for coefficient or pruning hyperparameter selection. In contrast, our universal subspace method, analytically computes the merging coefficients based solely on the geometry of a shared, low-rank universal subspace identified across models, requiring no iterative tuning or validation data-although optional finetuning is possible if data is available. Furthermore, because our subspace is intrinsically low-rank, the merged model contains significantly fewer parameters than any individual models, offering both computational efficiency and theoretical alignment guarantees not present in the baselines. Empirically, our approach achieves higher average accuracy (see Table 2), while reducing parameter count, thus enabling scalable and robust model merging without heuristic pruning or validation overhead. We note that we did not optimize our merging process and better results nearing finetuned performance may be achieved.

In summary, these four experiments provide strong empirical support for our universal subspace hypothesis and demonstrate its practical advantages in terms of memory efficiency, model merging, model reusability, and scalable deployment across diverse tasks and modalities.

### 3.3 Low rank shared universal subspaces in classical weights

While aforementioned experiments on CNNs trained from scratch, and LoRAs provide strong evidence for the presence of the joint subspace, we further rigorously test on large scale finetuned models (500 pretrained ViT, 50 LLaMA3-8B models, 177 GPT-2 and Flan-T5).

First, we collect ~500 pretrained Vision Transformer (ViT) models from HuggingFace, spanning diverse domains - medical imaging, satellite data, and synthetic - and

Table 3: Image Classification Accuracy

| Method | IID | OOD |
|---|---|---|
| Full Training | 94.4 ± 1.7 | 91.3 ± 2.1 |
| Universal ViT | 94.1 ± 2.0 | 87.8 ± 1.5 |

Table 2: Per-task results for eight ViT-B/32 models, each finetuned with LoRA on a different image classification dataset. "Finetuned" indicates the accuracy of each model on its respective training dataset. For each merging baseline, we report the normalized accuracy on every task, as well as the average across all tasks.

| Method | Datasets | | | | | | | | Avg |
|---|---|---|---|---|---|---|---|---|---|
| | Cars | DTD | EuroSAT | GTSRB | MNIST | RESISC45 | SUN397 | SVHN | |
| **Per-Task Absolute Accuracies (%)** | | | | | | | | | |
| Finetuned | 74.0 | 58.3 | 99.0 | 92.7 | 99.3 | 88.4 | 64.5 | 96.2 | 84.1 |
| **Per-Task Accuracies of Combined Models Normalized Against Finetuned Models (%)** | | | | | | | | | |
| RegMean | 80.2 | 71.3 | 37.9 | 47.3 | 43.1 | 70.5 | 99.3 | 43.0 | 60.9 |
| TA | 82.0 | 73.6 | 48.8 | 42.1 | 53.1 | 71.5 | 97.5 | 41.2 | 63.7 |
| TIES | 82.4 | 72.8 | 50.8 | 39.0 | 50.3 | 70.9 | 99.4 | 40.5 | 63.7 |
| DARE-TIES | 81.4 | 74.5 | 50.8 | 39.2 | 55.0 | 70.7 | 96.7 | 40.4 | 63.7 |
| KnOTS-TIES | 82.7 | 73.7 | 49.3 | 48.9 | 70.9 | 95.5 | 53.8 | 68.0 | 68.0 |
| KnOTS-DARE-TIES | 81.8 | 75.9 | 50.7 | 40.3 | 53.2 | 70.2 | 97.9 | 41.0 | 63.9 |
| **Ours** | **88.1** | **82.3** | **65.9** | **61.3** | **88.3** | **98.1** | **98.5** | **85.1** | **83.5** |

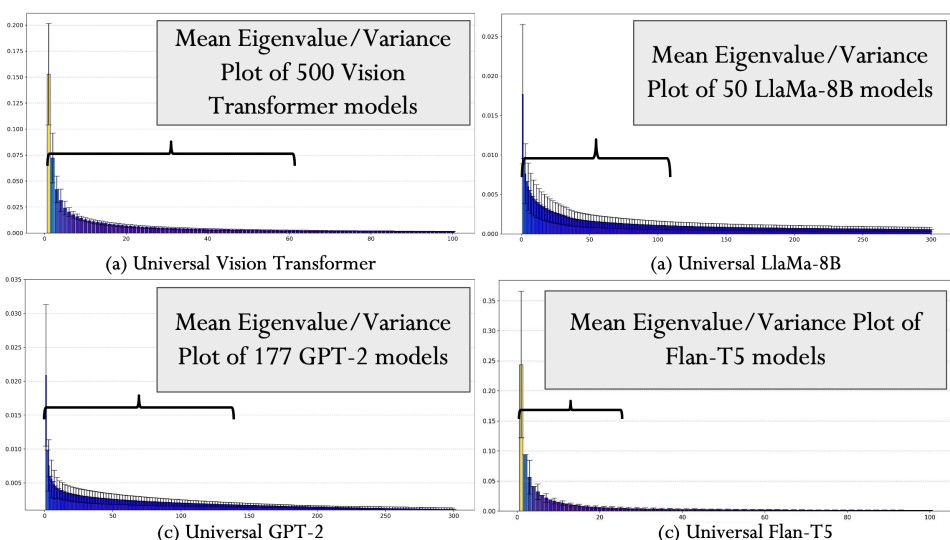

(a) Universal Vision Transformer  (a) Universal LlaMa-8B  (c) Universal GPT-2  (c) Universal Flan-T5

Figure 6: **Universal Subspaces in Classical Weights.** Spectral decomposition of weight matrices from (a) ∼500 Vision Transformers (b) 50 LLaMa-8B models (c) 177 GPT-2 models (d) GLUE Flan-T5 models - each trained independently across diverse tasks, datasets, and configurations - reveals a consistent low-rank structure: most variance is captured by the top few spectral basis. This suggests that, despite significant variation in training conditions, the learned weights consistently align along a shared low-dimensional subspace. For visualization clarity, only a fraction of the basis are shown; extended plots are provided in the Section B.2.

trained with varying losses, optimizers, and initializations. These models are used as-is, without curation or access to training data, to reflect real-world variability. See Section B.2 for details. Following our method (Section 3.1), we spectrally decompose all layers (excluding first and last) and observe, in Figure 6, that the majority of variance is captured by the top few spectral components, revealing a highly compressible, shared subspace across layers. Only the top 100 components are visualized for clarity.

To evaluate universal generalization, we project five held-out ViT models onto this 16-dim subspace and measure classification accuracy. As shown in Table 3, performance remains robust, indicating that a shared low-rank subspace spans a wide range of ViT model configurations and domains.

A major outcome of this experiment is that we can replace these 500 ViT models with a single Universal Subspace model. Ignoring the task-variable first and last layer (weight matrices vary due to different number of categories and input size and formats), we observe a requirement of **100× less memory**, and these savings are prone to increase as the number of trained models increases. We note that we are, to the best of our knowledge, the first work, to be able to *merge* 500 (and theoretically

more) Vision Transformer into a single universal subspace model. This result implies that hundreds of ViTs can be represented using a single subspace model - excluding task-specific layers - yielding up to **100×** **memory reduction**. To our knowledge, this is the first demonstration of merging over 500 ViTs into a single universal representation.

We further extend this analysis to 50 finetuned LLaMA3-8B models, 177 GPT-2 models, and Flan-T5 models (trained on GLUE Wang et al. (2019) datasets) again sourced from HuggingFace without filtering. As shown in Figure 6, a small number of directions capture dominant structure across models spanning diverse and distinct datasets and tasks. More details are provided in the Section B.2. This is, to our knowledge, the first instance of compressing such a large and diverse collection of foundation models into a unified subspace, highlighting its potential for large-scale model reuse and environmental efficiency.

### 3.3.1 FINDING UNIVERSAL SUBSPACES AND APPLYING THEM TO FUTURE TASKS

In this section, the low-rank shared subspaces estimated from a set of available tasks are leveraged to adapt to new, previously unseen tasks. While we do not make theoretical guarantees about reuse on unseen tasks, our experiments show that the approximate shared subspace is empirically reusable across a wide range of practical settings. Concretely, we reuse the shared principal directions and learn only their task-specific coefficients for the new task. Learning these low-rank coefficients is substantially cheaper than optimizing full-rank weights of size, reducing both computation and memory. The resulting trainable parameter counts are reported in Table 5. We find our universal subspace models can have significant impact on the carbon footprint issues of large AI models by making the training, inference and scaling of these models efficient and cheap. As shown in the previous section, we can effectively recycle and replace available pretrained models with a universal subspace model with every individual being represented by a sparse set of coefficients. In this section, we show a set of experiments where we utilize the universal subspaces to learn new tasks by freezing the components and simply learning the coefficients using gradient descent. We find that since we are only learning the coefficients, it drastically cuts down the number of parameters required to train the new models. Further, since these coefficients are simply linear scaling values, the optimization is smoother and faster.

Table 4: Performance on the GLUE Benchmark.

| Method | Speedup | CoLA | MRPC | RTE | QNLI | SST-2 | STS-B | Avg |
|---|---|---|---|---|---|---|---|---|
| LoRA | 1× | 59.56 | 86.76 | 77.61 | 92.53 | 94.72 | 90.81 | 83.67 |
| Universal order-2 | 2× | 61.82 | 87.25 | 77.62 | 92.71 | 94.15 | 90.48 | 84.01 |
| HOOI (order-2) | 2× | 61.96 | 87.55 | 77.50 | 92.83 | 94.45 | 90.40 | 84.12 |
| Universal order-3 | 1.8× | 62.06 | 86.52 | 75.81 | 92.98 | 94.26 | 90.39 | 83.67 |

We present two experiments - Image Classification using ViT-base and Natural Language Understanding using GLUE benchmark Wang et al. (2019) with RoBERTa-base model. Both involve creating a universal subspace using publicly available LoRA adapters. Details are provided in the Section C. For the GLUE benchmark, we follow the same setup as (Kopiczko et al., 2023) considering the 6 tasks - CoLA, MRPC, SST-2, QNLI, RTE and STS-B while omitting the time-intensive MNLI and QQP tasks. We initialize our universal subspace using a leave-one-out-setup, where the subspace is calculated using components of all but one LoRA adapter for which the coefficients are learned. For image classification, we utilize publicly available ViT LoRAs to extract our universal subspaces taking care that the data any of these pretrained LoRAs have not seen the data we will be training our coefficients on. Table 5 and Table 4 show that our universal subspace enables significantly more

Table 5: Image Classification with Vision Transformer.

| | # Training Params | CIFAR100 | Food101 | Flowers102 | CIFAR10 | Pets |
|---|---|---|---|---|---|---|
| Full Training | 86M | 92.8 | 90.7 | 98.82 | 99.0 | 91.2 |
| Universal ViT | 10K | 90.1 | 89.1 | 90.1 | 96.7 | 89.4 |

efficient and effective learning since only compact coefficients are trained. The storage required to

save all these models is also drastically reduced. The ViT models require 150 GB and LLaMA models require 1.6TB of memory in total. Our universal subspace reduces that memory requirement by more than **100**×.

# 4 DISCUSSION

This work provides, to the best of our knowledge, the first large-scale, cross-domain analysis showing that neural networks trained across diverse tasks, modalities, initializations, and hyperparameters consistently exhibit an architecture-specific shared low-rank universal subspace at the layer level. Concretely, by performing layer-wise spectral decompositions and retaining only the leading principal directions, an accurate approximation of these universal subspaces can be extracted. Empirically, this behavior emerges broadly: in fully finetuned models and LoRA-based adapters, in models trained from scratch, in both generative and discriminative settings, and in multimodal configurations. Moreover, the approximated subspaces generalize to out-of-distribution tasks, where projecting models and learning only a small set of coefficients suffices to recover strong performance. This enables adapting to new tasks without retraining or storing full weights, and supports robust multi-task learning, scalable fine-tuning, and principled model merging within a single unifying framework.

The practical implications are substantial. By learning only lightweight coefficients for shared layer-wise principal directions, large models can be extended with dramatically reduced computational and memory overhead. This lowers deployment costs while enabling more accessible AI development and data-free model merging. These results suggest a path toward scalable model reuse grounded in a simple geometric principle: most task variation lies in a shared, low-dimensional subspace.

**Why do these universal subspaces emerge?** Neural networks may exhibit spectral bias toward low-frequency functions, potentially creating polynomial eigenvalue decay that concentrates learning dynamics in a small number of dominant directions. Modern architectures also impose strong inductive biases - convolutional structures might favor local patterns, attention mechanisms could prioritize relational reasoning - that may constrain parameter variations to similar subspaces across tasks. The ubiquity of gradient-based optimization, with its inherent preference for smooth solutions, could further channel different learning trajectories toward shared geometric structures. If true, this would suggest that the universal subspace captures fundamental computational patterns that transcend specific tasks - potentially explaining why transfer learning works and why diverse problems often benefit from similar architectural modifications. However, the precise mechanisms remain an open question, making our empirical investigation all the more important to understand this surprising regularity in neural network learning.

# 5 LIMITATIONS AND FUTURE WORK

Although we provide conclusive results towards the existence and utility of universal shared subspaces, the current analysis has scope for future research, such as limited interpretability of the shared subspace and the corresponding directions. While it is a critical area of research, it is extremely cumbersome to demonstrate interpretability of the principal directions for each layer of the network. To the best of our knowledge we are not aware of any other literature that performs such an in-depth analysis of the weight space of large models across diverse tasks, data modalities and model architectures. The current approach to approximating a universal subspace relies on pretrained task-specific models (predictors) for tasks, which may not be readily available for new tasks. An interesting direction for future research would be to explore model independent methods for learning a universal shared subspace, potentially derived directly from data. Furthermore, the conditions proposed in Ortiz-Jimenez et al. (2023) for enabling task arithmetic rely on localized eigenfunctions which are not conducive to learning a shared universal subspace. As a result, performing task arithmetic within the current framework of a shared universal subspace is non-trivial and warrants further investigation.

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

Table 6: Notation reference.

| Notation | Description |
|---|---|
| $\mathcal{H}$ | Separable Hilbert space with inner product $\langle \cdot, \cdot \rangle$, norm $\|\cdot\|$. |
| $a \otimes b$ | Rank-one operator $g \mapsto \langle b, g \rangle a$, $\|a \otimes b\|_{\text{op}} = \|a\| \|b\|$. |
| $T$ | Number of tasks. |
| $\mathcal{T}$ | Distribution over tasks. |
| $D_t$ | Data distribution for task $t$. |
| $S_t = \{(x_{t,i}, y_{t,i})\}_{i=1}^{n_t}$ | Dataset of size $n_t$ for task $t$. |
| $f_t^\star \in \mathcal{H}$ | Ground-truth predictor for task $t$. |
| $\hat{f}_t \in \mathcal{H}$ | Learned predictor for task $t$. |
| $B$ | Uniform bound: $\|f_t^\star\| \leq B$ almost surely. |
| $\mathcal{R}_{n_t, D_t}(\mathcal{H})$ | Per-task estimation error rate (e.g. $\tilde{O}(1/\sqrt{n_t})$). |
| $\eta_t$ | Per-task error: $\eta_t := \mathcal{R}_{n_t, D_t}(\mathcal{H}) + \sqrt{\frac{\ln(2T/\delta)}{2n_t}}$. |
| $\bar{\eta}$ | Average error: $\frac{1}{T} \sum_{t=1}^{T} \eta_t$. |
| $\overline{\eta_t^2}$ | Average squared error: $\frac{1}{T} \sum_{t=1}^{T} \eta_t^2$. |
| $\mathcal{S}$ | Population operator: $\mathcal{S} = \mathbb{E}_{t \sim \mathcal{T}}[f_t^\star \otimes f_t^\star]$. |
| $\hat{\mathcal{S}}$ | Empirical operator (true predictors): $\frac{1}{T} \sum_{t=1}^{T} f_t^\star \otimes f_t^\star$. |
| $\tilde{\mathcal{S}}$ | Empirical operator (learned predictors): $\frac{1}{T} \sum_{t=1}^{T} \hat{f}_t \otimes \hat{f}_t$. |
| $\lambda_1 \geq \lambda_2 \geq \ldots$ | Eigenvalues of $\mathcal{S}$. |
| $\phi_i$ | Orthonormal eigenvectors of $\mathcal{S}$. |
| $P_k$ | Projector onto top-$k$ eigenspace of $\mathcal{S}$. |
| $\tilde{P}_k$ | Projector onto top-$k$ eigenspace of $\tilde{\mathcal{S}}$. |
| $\gamma_k$ | Eigengap: $\gamma_k := \lambda_k - \lambda_{k+1} > 0$. |
| $\|A\|_{\text{op}}$ | Operator (spectral) norm. |
| $\|A\|_{HS}$ | Hilbert–Schmidt norm. |
| $r(V)$ | Intrinsic/Effective rank: $\text{tr}(V)/\|V\|_{\text{op}}$. |
| $X_t$ | Centered operator: $X_t := f_t^\star \otimes f_t^\star - \mathcal{S}$. |
| $V$ | Variance operator: $V := \sum_{t=1}^{T} \mathbb{E}[X_t^2]$. |
| $\delta, \delta_t, \delta_T$ | Failure probabilities (global, per-task, across-task). |

# A APPENDIX

## A.1 RELATED WORK

Several lines of prior research support the core intuition behind our universal subspace hypothesis, though they do not provide a unified, scalable framework for identifying and leveraging such subspaces across architectures, tasks, and modalities. The Neural Tangent Kernel framework reinforces this idea, demonstrating that, in the infinite-width regime, training dynamics are governed by a kernel largely invariant to task specifics, implying the presence of common functional subspaces. (Jacot et al., 2018). This result implies that training is implicitly constrained to a shared function space, suggesting the existence of low-dimensional structures that generalize across tasks. Complementing this, works in mechanistic interpretability has uncovered modular and recurring patterns that consistently re-emerge in independently trained models (Olah et al., 2020; Chughtai et al., 2023), supporting the notion of structural universality in network representations.

Empirical studies further strengthen this perspective. The lottery ticket hypothesis (Frankle & Carbin, 2019) demonstrates that overparameterized networks contain sparse subnetworks capable of matching full-model performance, implying that task-relevant information resides in a small, structured subset of weights. Similarly, mode connectivity studies (Garipov et al., 2018) reveal that seemingly isolated optima in parameter space are often connected by low-loss paths, suggesting that task solutions lie on a shared manifold. In convolutional models, Krizhevsky et al. (Krizhevsky et al., 2012) famously observed that early layers consistently learn Gabor-like filters, indicating a universal inductive bias in early representations. More recent works (Guth et al., 2024; Guth & Ménard, 2024) extends this

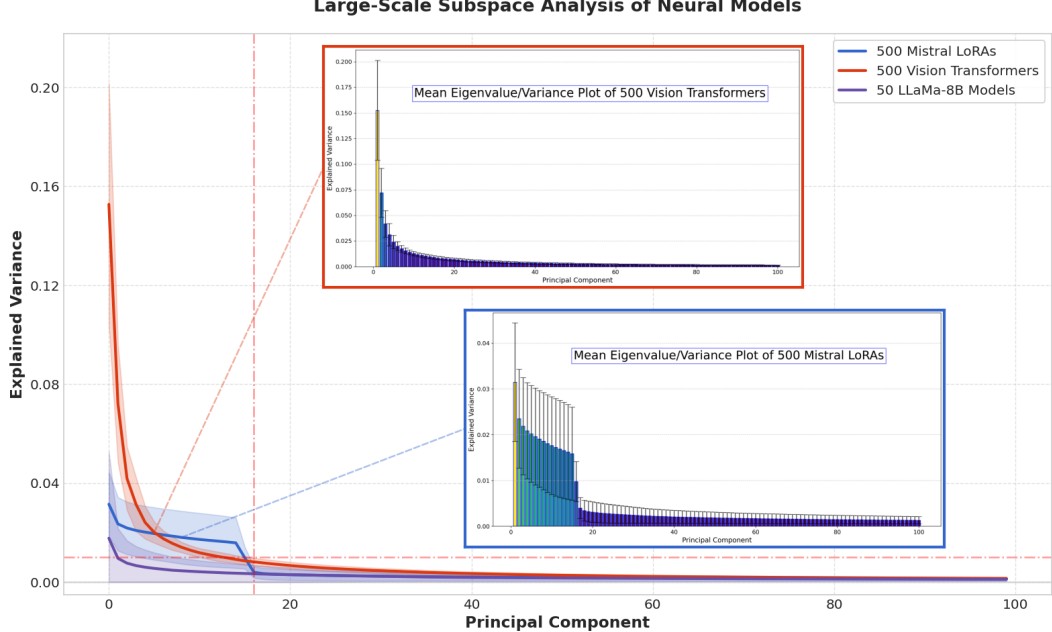

Figure 7: **Empirical Evidence for (Universal) Joint Weight Subspaces.** This figure illustrates the existence of joint low-dimensional subspaces across models trained on diverse tasks. We plot the average explained variance of the top few principal components of weight matrices from 500 Mistral-7B LoRAs, 500 Vision Transformers, and 50 LLaMA-8B models. Despite differences in modality, data, and training objective, all models exhibit rapid spectral decay - indicating that a small number of directions dominate across layers and settings. This consistent structure provides strong evidence for the presence of joint/universal subspaces, supporting our hypothesis that deep networks systematically reuse a common representational basis. Often, this shared subspace can be seen distinctly. The presence of the subspace has significant implications for deep learning. Not only can large number of models be compressed into a single, lighter Universal model with difference represented as lightweight coefficients, training on future tasks simply becomes tuning those coefficients. Since the basis are fixed, training becomes simpler and quicker. However, this convergence to similar subspace raises few important questions - is it possible to recover the "true" Universal Subspace without learning with huge amounts of data? Is this lack of diversity a bottleneck from current family of deep models?

observation to deeper layers, showing that certain eigenvectors of trained convolutional layers recur across networks trained on different datasets.

While these studies are suggestive of shared structures in neural representations or parameters, they remain limited in their focus, application and analysis. Our work fills this critical gap by presenting a principled and empirically validated method for discovering and utilizing universal parametric subspaces that span across architectures, tasks, and modalities. By conducting large-scale spectral analyses of over large number of diverse architectures, models and tasks, we demonstrate that a small number of principal directions consistently capture the majority of task-relevant variation. We then operationalize these findings by developing a practical framework for reusing these subspaces for parameter-efficient finetuning, task adaptation, and model merging, achieving competitive performance while dramatically reducing memory and compute requirements.

## A.2 THEORETICAL ANALYSIS

We apply a standard generalization bound over the squared error between the task function and its projection onto the shared subspace:

$$\ell(f_t, x) = \|f_t(x) - f_{t,k}(x)\|^2$$

To justify the application of PAC-style bounds, we verify that this loss is bounded. We assume that each task predictor $f_t$ lies in a Reproducing Kernel Hilbert Space (RKHS) with norm bounded by $B$, i.e., $\|f_t\|_{\mathcal{H}} \leq B$, and that the projection $f_{t,k}$ onto the learned shared subspace $\hat{\mathcal{H}}_k$ also satisfies $\|f_{t,k}\|_{\mathcal{H}} \leq B$.

Using the reproducing property and assuming a kernel bound $\kappa^2 = \sup_{x \in \mathcal{X}} \|\phi(x)\|^2$, we have for any $x$:

$$\|f_t(x)\| \leq \kappa B \quad \text{and} \quad \|f_{t,k}(x)\| \leq \kappa B$$

Thus, the pointwise squared loss is bounded as:

$$\|f_t(x) - f_{t,k}(x)\|^2 \leq (\|f_t(x)\| + \|f_{t,k}(x)\|)^2 \leq (2\kappa B)^2 = 4\kappa^2 B^2$$

Therefore, the loss function is bounded in $[0, 4\kappa^2 B^2]$, satisfying the conditions required for PAC-style generalization bounds to hold.

**Lemma A.1** (Matrix Bernstein for self-adjoint operators)**.** *There exist absolute constants $C > 0$ such that, for any $\delta_T \in (0,1)$, we have with probability at least $1 - \delta_T$,*

$$\left\|\hat{\mathcal{S}} - \mathcal{S}\right\|_{\mathrm{op}} \leq C\, B^2 \left[\sqrt{\frac{\ln(c/\delta_T)}{T}} + \frac{\ln(c/\delta_T)}{T}\right]$$

*Proof.* Operator Bernstein (intrinsic form).
Let $X_1, \ldots, X_T$ be independent, mean-zero, self-adjoint, bounded operators on a separable Hilbert space. Suppose

$$\|X_t\|_{\mathrm{op}} \leq L \quad \text{a.s. for all } t.$$

Then from (Minsker, 2017; Koltchinskii & Lounici, 2014) there exist absolute constants $C, c > 0$ such that for every $\delta \in (0,1)$,

$$\left\|\frac{1}{T}\sum_{t=1}^T X_t\right\|_{\mathrm{op}} \leq C\left[\sqrt{\frac{\left\|\sum_{t=1}^T \mathbb{E}[X_t^2]\right\|_{\mathrm{op}}}{T^2} \ln\left(\frac{c\left(1 + \frac{\mathrm{tr}\left(\sum_{t=1}^T \mathbb{E}[X_t^2]\right)}{\left\|\sum_{t=1}^T \mathbb{E}[X_t^2]\right\|_{\mathrm{op}}}\right)}{\delta_T}\right)} + \frac{L}{T}\ln\left(\frac{c\left(1 + \frac{\mathrm{tr}\left(\sum_{t=1}^T \mathbb{E}[X_t^2]\right)}{\left\|\sum_{t=1}^T \mathbb{E}[X_t^2]\right\|_{\mathrm{op}}}\right)}{\delta_T}\right)\right]$$

with probability at least $1 - \delta_T$.

Application to $X_t = f_t^\star \otimes f_t^\star - \mathcal{S}$ with $\|f_t^\star\| \leq B$ a.s.

We have

$$\|X_t\|_{\mathrm{op}} \leq \|f_t^\star\|^2 + \|\mathcal{S}\|_{\mathrm{op}} \leq B^2 + \mathbb{E}\|f^\star\|^2 \leq 2B^2.$$

so $L \leq 2B^2$. Moreover, for $X_t = f_t^\star \otimes f_t^\star - \mathcal{S}$ we have

$$\mathbb{E}[X_t^2] \preceq 2B^2 \mathcal{S}.$$

Hence

$$\left\|\sum_{t=1}^T \mathbb{E}[X_t^2]\right\|_{\mathrm{op}} \leq 2TB^2\|\mathcal{S}\|_{\mathrm{op}}, \qquad \mathrm{tr}\left(\sum_{t=1}^T \mathbb{E}[X_t^2]\right) \leq 2TB^2\,\mathrm{tr}(\mathcal{S}).$$

By asumption 2.3,

$$\frac{\mathrm{tr}(\sum_{t=1}^T \mathbb{E}[X_t^2])}{\left\|\sum_{t=1}^T \mathbb{E}[X_t^2]\right\|_{\mathrm{op}}} \leq \frac{\mathrm{tr}(\mathcal{S})}{\|\mathcal{S}\|_{\mathrm{op}}} \leq \kappa.$$

Therefore the intrinsic logarithmic factor in Bernstein reduces to

$$\ln\left(\frac{c(1 + \kappa)}{\delta_T}\right),$$

and since $\kappa$ is a fixed constant, $1 + \kappa$ can be absorbed into $c$.

Plugging into Bernstein gives

$$\|\hat{\mathcal{S}} - \mathcal{S}\|_{\mathrm{op}} \leq C \left[ \sqrt{\frac{2B^2 \|\mathcal{S}\|_{\mathrm{op}} \ln(c/\delta_T)}{T}} + \frac{2B^2 \ln(c/\delta_T)}{T} \right],$$

with probability at least $1 - \delta_T$.

$\square$

**Lemma A.2** (Davis–Kahan, sin-$\Theta$). *Let $\gamma_k > 0$. Then*

$$\left\|\tilde{P}_k - P_k\right\|_{\mathrm{op}} \leq \frac{2}{\gamma_k} \left\|\tilde{\mathcal{S}} - \mathcal{S}\right\|_{\mathrm{op}}.$$

*using definition of $\gamma_k$ from definition 2.1.*

**Theorem A.3** (Restating Two-level convergence to the shared subspace theorem). *Assume 2.3–2.4. Let $c_1, c_2$ be any absolute constants. For any $\delta \in (0, 1)$, choose $\delta_t = \delta/(2T)$ and set $\delta_T = \delta/2$. With probability at least $1 - \delta$ (over tasks and all per-task samples),*

$$\left\|\tilde{\mathcal{S}} - \mathcal{S}\right\|_{\mathrm{op}} \leq c_1 B^2 \sqrt{\frac{\ln(c_2/\delta)}{T}} + (2B\bar{\eta} + \overline{\eta^2}) \tag{3}$$

*If moreover $\gamma_k > 0$, then*

$$\left\|\tilde{P}_k - P_k\right\|_{\mathrm{op}} \leq \frac{2}{\gamma_k} \left( c_1 B^2 \sqrt{\frac{\ln(c_2/\delta)}{T}} + (2B\bar{\eta} + \overline{\eta^2}) \right). \tag{4}$$

*where $\bar{\eta} = \frac{1}{T} \sum_{t=1}^{T} \eta_t$, $\overline{\eta^2} = \frac{1}{T} \sum_{t=1}^{T} \eta_t^2$ and $\eta_t$ is defined same as in assumption 2.4*

*Proof of Theorem 2.5.* **(i) Triangle split.** $\left\|\tilde{\mathcal{S}} - \mathcal{S}\right\|_{\mathrm{op}} \leq \left\|\tilde{\mathcal{S}} - \hat{\mathcal{S}}\right\|_{\mathrm{op}} + \left\|\hat{\mathcal{S}} - \mathcal{S}\right\|_{\mathrm{op}}$.

**(ii) Within-task term.** We know that,

$$\begin{aligned}
\left\|\hat{f}_t \otimes \hat{f}_t - f_t^\star \otimes f_t^\star\right\|_{\mathrm{op}} &\leq \left\|\hat{f}_t - f_t^\star\right\| (\left\|\hat{f}_t\right\| + \|f_t^\star\|) \\
&\leq \|\hat{f}_t - f_t^\star\|(\|\hat{f}_t\| + \|f_t^\star\|) \\
&\leq \eta_t (2B + \eta_t) \qquad (\text{since } \|\hat{f}_t\| \leq \|f_t^\star\| + \|\hat{f}_t - f_t^\star\| \leq B + \eta_t) \\
&= 2B\eta_t + \eta_t^2.
\end{aligned}$$

Averaging and using the triangle inequality for operator norms,

$$\left\|\tilde{\mathcal{S}} - \hat{\mathcal{S}}\right\|_{\mathrm{op}} \leq 2B\bar{\eta} + \overline{\eta^2}$$

This holds on the event $\bigcap_{t=1}^{T} \{\left\|\hat{f}_t - f_t^\star\right\| \leq \eta_t\}$, whose probability is at least $1 - \sum_t \delta_t = 1 - \delta/2$.

**(iii) Across-task term.** Let $X_t := f_t^\star \otimes f_t^\star - \mathbb{E}[f^\star \otimes f^\star]$. Then $X_t$ are independent, mean-zero, self-adjoint, and $\|X_t\|_{\mathrm{op}} \leq \|f_t^\star\|^2 + \|S\|_{\mathrm{op}} \leq 2B^2$. Lemma A.1 (with $R \asymp B^2$) yields

$$\left\|\hat{\mathcal{S}} - \mathcal{S}\right\|_{\mathrm{op}} \leq c_1 B^2 \sqrt{\frac{\ln(c_2/\delta)}{T}}$$

$$\left\|\tilde{\mathcal{S}} - \mathcal{S}\right\|_{\mathrm{op}} \leq c_1 B^2 \sqrt{\frac{\ln(c_2/\delta)}{T}} + 2B \left( \sum_{t=1}^{T} \mathcal{R}_{n_t, D_t}(\mathcal{H}) + \sqrt{\frac{\ln(2T/\delta)}{2n_t}} \right) + \left( \sum_{t=1}^{T} \mathcal{R}_{n_t, D_t}^2(\mathcal{H}) + \frac{\ln(2T/\delta)}{2n_t} \right)$$

$$\leq c_1 B^2 \sqrt{\frac{\ln(c_2/\delta)}{T}} + O \left( \sum_{t=1}^{T} \mathcal{R}_{n_t, D_t}^2(\mathcal{H}) + \frac{\ln(2T/\delta)}{2n_t} \right)$$

with probability at least $1 - \delta_T = 1 - \delta/2$.

**(iv) Union bound and Davis–Kahan.** Combining (ii)–(iii) with a union bound gives equation 1. Lemma A.2 then implies equation 2. $\square$

**Definition A.4** (Population projection risk)**.** For a $k$-dimensional subspace $\mathcal{H}_k^\star \subset \mathcal{H}$, define

$$\mathcal{R}(\mathcal{H}_k^\star) := \mathbb{E}_{t \sim \tau} \left\| f_t^\star - P_{\mathcal{H}_k^\star} f_t^\star \right\|^2 .$$

**Corollary A.5** (Excess projection risk of the learned subspace)**.** *Under the event of Theorem 2.5,*

$$\mathcal{R}(\tilde{\mathcal{H}}_k) \;\leq\; \sum_{i>k} \lambda_i \;+\; \frac{2 \operatorname{tr}(S)}{\gamma_k} \left( c_1 B^2 \sqrt{\frac{\ln(c_2/\delta)}{T}} + 2B\,\bar{\eta} + \overline{\eta^2} \right) .$$

*Proof.* Optimality of $P_k$ gives $\mathcal{R}(\mathcal{H}_k^\star) = \sum_{i>k} \mu_i$. Moreover,

$$\mathcal{R}(\tilde{\mathcal{H}}_k) - \mathcal{R}(\mathcal{H}_k^\star) = \mathbb{E} \left\langle f_t^\star, (P_k - \tilde{P}_k) f_t^\star \right\rangle \leq \left\| \tilde{P}_k - P_k \right\|_{\mathrm{op}} \mathbb{E} \left\| f_t^\star \right\|^2 = \operatorname{tr}(S) \left\| \tilde{P}_k - P_k \right\|_{\mathrm{op}} .$$

Apply equation 2. $\qquad\square$

*Remark* A.6 (Where Rademacher complexity enters)*.* Assumption 2.4 is instantiated by your learning procedure. For strongly-convex ERM (e.g., kernel ridge), a standard Rademacher-based excess-risk bound together with curvature yields an $\eta_t = \eta_t(n_t, \delta_t)$ that vanishes with $n_t$. Plugging these $\eta_t$ into $\bar{\eta}$ and $\overline{\eta^2}$ makes the rate explicit.

# B  UNIVERSAL SUBSPACE ANALYSIS

Similar methodology is followed for subspace analysis for both LoRA and classical weight models. In fact, LoRA analysis' results can be theoretically extended to classical weights, as LoRA weights can be construed to be simple translations from a mean weight matrix. However, in order to solidify our universal subspace hypothesis, we conduct extensive experiments for both types of models. LoRA is chosen because of the recent spurt in the availability of LoRA models trained on diverse kinds of datasets and models. We do this universal subspace analysis on all weight parameters in every neural network layer except the first (or few initial) and last neural network layer. This is because these layers may differ across models due to differences in input shapes and types, loss functions, and the tasks being trained. We also focus our analysis on linear/fully-connected and matrix weights, as the analysis done on these are straightforward and the results observed can be trivially extended to other types of neural parameters (Ma & Lu, 2017).

**Secondary Subspace** refers to the residual subspace that remains after removing the top $k$ principal directions associated with the low-rank universal subspace. This subspace is orthogonal to the universal subspace and serves as a control for evaluating the uniqueness and effectiveness of the learned shared subspace. To make computation tractable when the residual subspace is high-dimensional, we focus on the top components beyond rank $k$, as computing a full SVD is often impractical. This approximation is justified, since the lower components typically capture noise, which has been shown to degrade performance (Sharma et al., 2023).

**How to choose top $k$ components?** As shown in all eigenvalue (scree) plots, a trivial way to choose is a simple visual inspection, since we can see a discontinuity in the spectral analysis. Another way is to define a threshold on the explained variance, all components whose explained variance is close to zero <.01 are considered secondary subspace, and can be discarded. A more structured way is to define an optimal singular value threshold for the HOSVD, as found by previous works (Gavish & Donoho, 2014).

## B.1  LOWER RANK SHARED UNIVERSAL SUBSPACES WITHIN LOW RANK ADAPTATION (LoRA) MODELS

Spectral Decomposition is employed to extract the top `k` principal directions for each of the LoRA matrices `B` and `A`, which are concatenated across all available models. Subsequently, the top `k` principal directions are selected to define the low-rank subspace shared among the LoRA matrices. This process is conducted separately for each layer of the model to derive a low-rank approximated shared subspace for every individual layer. In practice, for every layer, the rank vectors of all available

LoRA matrices are extracted and concatenated into a single matrix. This matrix is then normalized by subtracting the feature-wise mean from each vector, after which principal directions are extracted. The mode-1(order-1) variant of our method is mathematically equivalent to Principal Component Analysis (PCA), hence we can use `torch.pca_lowrank` or `sklearn.decomposition.PCA` to extract the principal directions. The data matrix corresponding to a specific layer for 500 LoRA models is structured as $500r \times d$, where $r$ denotes the rank of each LoRA and $d$ specifies the dimension of each rank vector. The same calculation can be applied to the `BA` matrix instead of individually to `B` and `A`, thereby increasing the computational cost of the Spectral Decomposition without affecting the outcome.

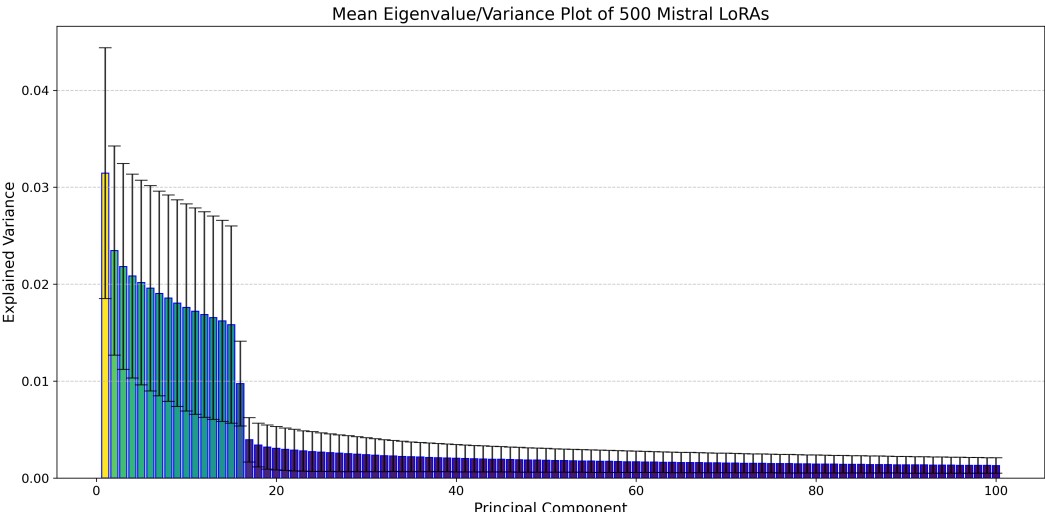

Figure 8: Spectral analysis of the Mistral-7B-Instruct-v0.2 model: Aggregated eigenvalue (scree) plot across 500 LoRA models and all layers. The plot demonstrates that the majority of the variance is consistently captured by the top 16 principal directions, indicating the presence of a shared low-dimensional universal subspace.

**Universal Mistral-7B/Lots of LoRAs experiment details**   In our first experimental analysis, we use 500 LoRA models trained on distinct Natural Instructions (Wang et al., 2022) using Mistral-7B-Instruct-v0.2 (Jiang et al., 2023) as the base (Brüel-Gabrielsson et al., 2024). Please refer to Brüel-Gabrielsson et al. (2024) for more details on how the LoRA models were trained.

Table 7: Models from HuggingFace for the Universal Mistral LoRA. Models in blue indicate the OOD models and the ones in red are the IID models used for evaluation.

| | |
|---|---|
| Lots-of-LoRAs/Mistral-7B-Instruct-v0.2-4b-r16-task391 | Lots-of-LoRAs/Mistral-7B-Instruct-v0.2-4b-r16-task290 |
| Lots-of-LoRAs/Mistral-7B-Instruct-v0.2-4b-r16-task442 | Lots-of-LoRAs/Mistral-7B-Instruct-v0.2-4b-r16-task1598 |
| Lots-of-LoRAs/Mistral-7B-Instruct-v0.2-4b-r16-task039 | |
| Lots-of-LoRAs/Mistral-7B-Instruct-v0.2-4b-r16-task076 | Lots-of-LoRAs/Mistral-7B-Instruct-v0.2-4b-r16-task627 |
| Lots-of-LoRAs/Mistral-7B-Instruct-v0.2-4b-r16-task664 | Lots-of-LoRAs/Mistral-7B-Instruct-v0.2-4b-r16-task819 |
| Lots-of-LoRAs/Mistral-7B-Instruct-v0.2-4b-r16-task1631 | |
| Lots-of-LoRAs/Mistral-7B-Instruct-v0.2-4b-r16-task190 | Lots-of-LoRAs/Mistral-7B-Instruct-v0.2-4b-r16-task1391 |
| Lots-of-LoRAs/Mistral-7B-Instruct-v0.2-4b-r16-task1342 | Lots-of-LoRAs/Mistral-7B-Instruct-v0.2-4b-r16-task620 |
| Lots-of-LoRAs/Mistral-7B-Instruct-v0.2-4b-r16-task769 | Lots-of-LoRAs/Mistral-7B-Instruct-v0.2-4b-r16-task1448 |
| Lots-of-LoRAs/Mistral-7B-Instruct-v0.2-4b-r16-task247 | Lots-of-LoRAs/Mistral-7B-Instruct-v0.2-4b-r16-task513 |
| Lots-of-LoRAs/Mistral-7B-Instruct-v0.2-4b-r16-task875 | Lots-of-LoRAs/Mistral-7B-Instruct-v0.2-4b-r16-task515 |
| Lots-of-LoRAs/Mistral-7B-Instruct-v0.2-4b-r16-task1534 | Lots-of-LoRAs/Mistral-7B-Instruct-v0.2-4b-r16-task1551 |
| Lots-of-LoRAs/Mistral-7B-Instruct-v0.2-4b-r16-task583 | Lots-of-LoRAs/Mistral-7B-Instruct-v0.2-4b-r16-task1431 |
| Lots-of-LoRAs/Mistral-7B-Instruct-v0.2-4b-r16-task270 | Lots-of-LoRAs/Mistral-7B-Instruct-v0.2-4b-r16-task1487 |
| Lots-of-LoRAs/Mistral-7B-Instruct-v0.2-4b-r16-task679 | Lots-of-LoRAs/Mistral-7B-Instruct-v0.2-4b-r16-task456 |
| Lots-of-LoRAs/Mistral-7B-Instruct-v0.2-4b-r16-task385 | Lots-of-LoRAs/Mistral-7B-Instruct-v0.2-4b-r16-task1607 |
| Lots-of-LoRAs/Mistral-7B-Instruct-v0.2-4b-r16-task278 | Lots-of-LoRAs/Mistral-7B-Instruct-v0.2-4b-r16-task022 |
| Lots-of-LoRAs/Mistral-7B-Instruct-v0.2-4b-r16-task210 | Lots-of-LoRAs/Mistral-7B-Instruct-v0.2-4b-r16-task137 |
| Lots-of-LoRAs/Mistral-7B-Instruct-v0.2-4b-r16-task574 | Lots-of-LoRAs/Mistral-7B-Instruct-v0.2-4b-r16-task629 |
| Lots-of-LoRAs/Mistral-7B-Instruct-v0.2-4b-r16-task1378 | Lots-of-LoRAs/Mistral-7B-Instruct-v0.2-4b-r16-task1194 |
| Lots-of-LoRAs/Mistral-7B-Instruct-v0.2-4b-r16-task1529 | Lots-of-LoRAs/Mistral-7B-Instruct-v0.2-4b-r16-task453 |
| Lots-of-LoRAs/Mistral-7B-Instruct-v0.2-4b-r16-task102 | Lots-of-LoRAs/Mistral-7B-Instruct-v0.2-4b-r16-task460 |
| Lots-of-LoRAs/Mistral-7B-Instruct-v0.2-4b-r16-task1204 | Lots-of-LoRAs/Mistral-7B-Instruct-v0.2-4b-r16-task1384 |
| Lots-of-LoRAs/Mistral-7B-Instruct-v0.2-4b-r16-task1572 | Lots-of-LoRAs/Mistral-7B-Instruct-v0.2-4b-r16-task699 |
| Lots-of-LoRAs/Mistral-7B-Instruct-v0.2-4b-r16-task1722 | Lots-of-LoRAs/Mistral-7B-Instruct-v0.2-4b-r16-task580 |

| | |
|---|---|
| Lots-of-LoRAs/Mistral-7B-Instruct-v0.2-4b-r16-task605 | Lots-of-LoRAs/Mistral-7B-Instruct-v0.2-4b-r16-task1152 |
| Lots-of-LoRAs/Mistral-7B-Instruct-v0.2-4b-r16-task1283 | Lots-of-LoRAs/Mistral-7B-Instruct-v0.2-4b-r16-task637 |
| Lots-of-LoRAs/Mistral-7B-Instruct-v0.2-4b-r16-task723 | Lots-of-LoRAs/Mistral-7B-Instruct-v0.2-4b-r16-task084 |
| Lots-of-LoRAs/Mistral-7B-Instruct-v0.2-4b-r16-task201 | Lots-of-LoRAs/Mistral-7B-Instruct-v0.2-4b-r16-task956 |
| Lots-of-LoRAs/Mistral-7B-Instruct-v0.2-4b-r16-task167 | Lots-of-LoRAs/Mistral-7B-Instruct-v0.2-4b-r16-task1192 |
| Lots-of-LoRAs/Mistral-7B-Instruct-v0.2-4b-r16-task300 | Lots-of-LoRAs/Mistral-7B-Instruct-v0.2-4b-r16-task1714 |
| Lots-of-LoRAs/Mistral-7B-Instruct-v0.2-4b-r16-task388 | Lots-of-LoRAs/Mistral-7B-Instruct-v0.2-4b-r16-task516 |
| Lots-of-LoRAs/Mistral-7B-Instruct-v0.2-4b-r16-task127 | Lots-of-LoRAs/Mistral-7B-Instruct-v0.2-4b-r16-task362 |
| Lots-of-LoRAs/Mistral-7B-Instruct-v0.2-4b-r16-task1158 | Lots-of-LoRAs/Mistral-7B-Instruct-v0.2-4b-r16-task322 |
| Lots-of-LoRAs/Mistral-7B-Instruct-v0.2-4b-r16-task697 | Lots-of-LoRAs/Mistral-7B-Instruct-v0.2-4b-r16-task1566 |
| Lots-of-LoRAs/Mistral-7B-Instruct-v0.2-4b-r16-task1451 | Lots-of-LoRAs/Mistral-7B-Instruct-v0.2-4b-r16-task1135 |
| Lots-of-LoRAs/Mistral-7B-Instruct-v0.2-4b-r16-task341 | Lots-of-LoRAs/Mistral-7B-Instruct-v0.2-4b-r16-task267 |
| Lots-of-LoRAs/Mistral-7B-Instruct-v0.2-4b-r16-task1720 | Lots-of-LoRAs/Mistral-7B-Instruct-v0.2-4b-r16-task1452 |
| Lots-of-LoRAs/Mistral-7B-Instruct-v0.2-4b-r16-task131 | Lots-of-LoRAs/Mistral-7B-Instruct-v0.2-4b-r16-task685 |
| Lots-of-LoRAs/Mistral-7B-Instruct-v0.2-4b-r16-task727 | Lots-of-LoRAs/Mistral-7B-Instruct-v0.2-4b-r16-task1590 |
| Lots-of-LoRAs/Mistral-7B-Instruct-v0.2-4b-r16-task1731 | Lots-of-LoRAs/Mistral-7B-Instruct-v0.2-4b-r16-task047 |
| Lots-of-LoRAs/Mistral-7B-Instruct-v0.2-4b-r16-task929 | Lots-of-LoRAs/Mistral-7B-Instruct-v0.2-4b-r16-task1592 |
| Lots-of-LoRAs/Mistral-7B-Instruct-v0.2-4b-r16-task1326 | Lots-of-LoRAs/Mistral-7B-Instruct-v0.2-4b-r16-task615 |
| Lots-of-LoRAs/Mistral-7B-Instruct-v0.2-4b-r16-task1216 | Lots-of-LoRAs/Mistral-7B-Instruct-v0.2-4b-r16-task689 |
| Lots-of-LoRAs/Mistral-7B-Instruct-v0.2-4b-r16-task1156 | Lots-of-LoRAs/Mistral-7B-Instruct-v0.2-4b-r16-task1657 |
| Lots-of-LoRAs/Mistral-7B-Instruct-v0.2-4b-r16-task833 | Lots-of-LoRAs/Mistral-7B-Instruct-v0.2-4b-r16-task1206 |
| Lots-of-LoRAs/Mistral-7B-Instruct-v0.2-4b-r16-task1151 | Lots-of-LoRAs/Mistral-7B-Instruct-v0.2-4b-r16-task244 |
| Lots-of-LoRAs/Mistral-7B-Instruct-v0.2-4b-r16-task1562 | Lots-of-LoRAs/Mistral-7B-Instruct-v0.2-4b-r16-task043 |
| Lots-of-LoRAs/Mistral-7B-Instruct-v0.2-4b-r16-task044 | Lots-of-LoRAs/Mistral-7B-Instruct-v0.2-4b-r16-task722 |
| Lots-of-LoRAs/Mistral-7B-Instruct-v0.2-4b-r16-task183 | Lots-of-LoRAs/Mistral-7B-Instruct-v0.2-4b-r16-task563 |
| Lots-of-LoRAs/Mistral-7B-Instruct-v0.2-4b-r16-task155 | Lots-of-LoRAs/Mistral-7B-Instruct-v0.2-4b-r16-task353 |
| Lots-of-LoRAs/Mistral-7B-Instruct-v0.2-4b-r16-task616 | Lots-of-LoRAs/Mistral-7B-Instruct-v0.2-4b-r16-task1724 |
| Lots-of-LoRAs/Mistral-7B-Instruct-v0.2-4b-r16-task288 | Lots-of-LoRAs/Mistral-7B-Instruct-v0.2-4b-r16-task092 |
| Lots-of-LoRAs/Mistral-7B-Instruct-v0.2-4b-r16-task707 | Lots-of-LoRAs/Mistral-7B-Instruct-v0.2-4b-r16-task577 |
| Lots-of-LoRAs/Mistral-7B-Instruct-v0.2-4b-r16-task742 | Lots-of-LoRAs/Mistral-7B-Instruct-v0.2-4b-r16-task706 |
| Lots-of-LoRAs/Mistral-7B-Instruct-v0.2-4b-r16-task1401 | Lots-of-LoRAs/Mistral-7B-Instruct-v0.2-4b-r16-task1393 |
| Lots-of-LoRAs/Mistral-7B-Instruct-v0.2-4b-r16-task1198 | Lots-of-LoRAs/Mistral-7B-Instruct-v0.2-4b-r16-task966 |
| Lots-of-LoRAs/Mistral-7B-Instruct-v0.2-4b-r16-task219 | Lots-of-LoRAs/Mistral-7B-Instruct-v0.2-4b-r16-task1211 |
| Lots-of-LoRAs/Mistral-7B-Instruct-v0.2-4b-r16-task050 | Lots-of-LoRAs/Mistral-7B-Instruct-v0.2-4b-r16-task494 |
| Lots-of-LoRAs/Mistral-7B-Instruct-v0.2-4b-r16-task1379 | Lots-of-LoRAs/Mistral-7B-Instruct-v0.2-4b-r16-task176 |
| Lots-of-LoRAs/Mistral-7B-Instruct-v0.2-4b-r16-task068 | Lots-of-LoRAs/Mistral-7B-Instruct-v0.2-4b-r16-task566 |
| Lots-of-LoRAs/Mistral-7B-Instruct-v0.2-4b-r16-task333 | Lots-of-LoRAs/Mistral-7B-Instruct-v0.2-4b-r16-task593 |
| Lots-of-LoRAs/Mistral-7B-Instruct-v0.2-4b-r16-task667 | Lots-of-LoRAs/Mistral-7B-Instruct-v0.2-4b-r16-task1670 |
| Lots-of-LoRAs/Mistral-7B-Instruct-v0.2-4b-r16-task733 | Lots-of-LoRAs/Mistral-7B-Instruct-v0.2-4b-r16-task472 |
| Lots-of-LoRAs/Mistral-7B-Instruct-v0.2-4b-r16-task1168 | Lots-of-LoRAs/Mistral-7B-Instruct-v0.2-4b-r16-task075 |
| Lots-of-LoRAs/Mistral-7B-Instruct-v0.2-4b-r16-task148 | Lots-of-LoRAs/Mistral-7B-Instruct-v0.2-4b-r16-task683 |
| Lots-of-LoRAs/Mistral-7B-Instruct-v0.2-4b-r16-task1315 | Lots-of-LoRAs/Mistral-7B-Instruct-v0.2-4b-r16-task121 |
| Lots-of-LoRAs/Mistral-7B-Instruct-v0.2-4b-r16-task370 | Lots-of-LoRAs/Mistral-7B-Instruct-v0.2-4b-r16-task856 |
| Lots-of-LoRAs/Mistral-7B-Instruct-v0.2-4b-r16-task891 | Lots-of-LoRAs/Mistral-7B-Instruct-v0.2-4b-r16-task140 |
| Lots-of-LoRAs/Mistral-7B-Instruct-v0.2-4b-r16-task609 | Lots-of-LoRAs/Mistral-7B-Instruct-v0.2-4b-r16-task344 |
| Lots-of-LoRAs/Mistral-7B-Instruct-v0.2-4b-r16-task1703 | Lots-of-LoRAs/Mistral-7B-Instruct-v0.2-4b-r16-task070 |
| Lots-of-LoRAs/Mistral-7B-Instruct-v0.2-4b-r16-task072 | Lots-of-LoRAs/Mistral-7B-Instruct-v0.2-4b-r16-task504 |
| Lots-of-LoRAs/Mistral-7B-Instruct-v0.2-4b-r16-task695 | Lots-of-LoRAs/Mistral-7B-Instruct-v0.2-4b-r16-task1434 |
| Lots-of-LoRAs/Mistral-7B-Instruct-v0.2-4b-r16-task095 | Lots-of-LoRAs/Mistral-7B-Instruct-v0.2-4b-r16-task346 |
| Lots-of-LoRAs/Mistral-7B-Instruct-v0.2-4b-r16-task274 | Lots-of-LoRAs/Mistral-7B-Instruct-v0.2-4b-r16-task1325 |
| Lots-of-LoRAs/Mistral-7B-Instruct-v0.2-4b-r16-task1190 | Lots-of-LoRAs/Mistral-7B-Instruct-v0.2-4b-r16-task568 |
| Lots-of-LoRAs/Mistral-7B-Instruct-v0.2-4b-r16-task1482 | Lots-of-LoRAs/Mistral-7B-Instruct-v0.2-4b-r16-task924 |
| Lots-of-LoRAs/Mistral-7B-Instruct-v0.2-4b-r16-task761 | Lots-of-LoRAs/Mistral-7B-Instruct-v0.2-4b-r16-task596 |
| Lots-of-LoRAs/Mistral-7B-Instruct-v0.2-4b-r16-task382 | Lots-of-LoRAs/Mistral-7B-Instruct-v0.2-4b-r16-task926 |
| Lots-of-LoRAs/Mistral-7B-Instruct-v0.2-4b-r16-task065 | Lots-of-LoRAs/Mistral-7B-Instruct-v0.2-4b-r16-task1421 |
| Lots-of-LoRAs/Mistral-7B-Instruct-v0.2-4b-r16-task323 | Lots-of-LoRAs/Mistral-7B-Instruct-v0.2-4b-r16-task1310 |
| Lots-of-LoRAs/Mistral-7B-Instruct-v0.2-4b-r16-task110 | Lots-of-LoRAs/Mistral-7B-Instruct-v0.2-4b-r16-task1288 |
| Lots-of-LoRAs/Mistral-7B-Instruct-v0.2-4b-r16-task1503 | Lots-of-LoRAs/Mistral-7B-Instruct-v0.2-4b-r16-task269 |
| Lots-of-LoRAs/Mistral-7B-Instruct-v0.2-4b-r16-task821 | Lots-of-LoRAs/Mistral-7B-Instruct-v0.2-4b-r16-task565 |
| Lots-of-LoRAs/Mistral-7B-Instruct-v0.2-4b-r16-task867 | Lots-of-LoRAs/Mistral-7B-Instruct-v0.2-4b-r16-task755 |
| Lots-of-LoRAs/Mistral-7B-Instruct-v0.2-4b-r16-task378 | Lots-of-LoRAs/Mistral-7B-Instruct-v0.2-4b-r16-task518 |
| Lots-of-LoRAs/Mistral-7B-Instruct-v0.2-4b-r16-task195 | Lots-of-LoRAs/Mistral-7B-Instruct-v0.2-4b-r16-task618 |
| Lots-of-LoRAs/Mistral-7B-Instruct-v0.2-4b-r16-task638 | Lots-of-LoRAs/Mistral-7B-Instruct-v0.2-4b-r16-task1217 |
| Lots-of-LoRAs/Mistral-7B-Instruct-v0.2-4b-r16-task118 | Lots-of-LoRAs/Mistral-7B-Instruct-v0.2-4b-r16-task1564 |
| Lots-of-LoRAs/Mistral-7B-Instruct-v0.2-4b-r16-task1429 | Lots-of-LoRAs/Mistral-7B-Instruct-v0.2-4b-r16-task687 |
| Lots-of-LoRAs/Mistral-7B-Instruct-v0.2-4b-r16-task640 | Lots-of-LoRAs/Mistral-7B-Instruct-v0.2-4b-r16-task1328 |
| Lots-of-LoRAs/Mistral-7B-Instruct-v0.2-4b-r16-task1311 | Lots-of-LoRAs/Mistral-7B-Instruct-v0.2-4b-r16-task285 |
| Lots-of-LoRAs/Mistral-7B-Instruct-v0.2-4b-r16-task1341 | Lots-of-LoRAs/Mistral-7B-Instruct-v0.2-4b-r16-task1603 |
| Lots-of-LoRAs/Mistral-7B-Instruct-v0.2-4b-r16-task162 | Lots-of-LoRAs/Mistral-7B-Instruct-v0.2-4b-r16-task063 |
| Lots-of-LoRAs/Mistral-7B-Instruct-v0.2-4b-r16-task686 | Lots-of-LoRAs/Mistral-7B-Instruct-v0.2-4b-r16-task1483 |
| Lots-of-LoRAs/Mistral-7B-Instruct-v0.2-4b-r16-task146 | Lots-of-LoRAs/Mistral-7B-Instruct-v0.2-4b-r16-task1729 |
| Lots-of-LoRAs/Mistral-7B-Instruct-v0.2-4b-r16-task852 | Lots-of-LoRAs/Mistral-7B-Instruct-v0.2-4b-r16-task1622 |
| Lots-of-LoRAs/Mistral-7B-Instruct-v0.2-4b-r16-task1704 | Lots-of-LoRAs/Mistral-7B-Instruct-v0.2-4b-r16-task145 |
| Lots-of-LoRAs/Mistral-7B-Instruct-v0.2-4b-r16-task648 | Lots-of-LoRAs/Mistral-7B-Instruct-v0.2-4b-r16-task151 |
| Lots-of-LoRAs/Mistral-7B-Instruct-v0.2-4b-r16-task1212 | Lots-of-LoRAs/Mistral-7B-Instruct-v0.2-4b-r16-task045 |
| Lots-of-LoRAs/Mistral-7B-Instruct-v0.2-4b-r16-task893 | Lots-of-LoRAs/Mistral-7B-Instruct-v0.2-4b-r16-task936 |
| Lots-of-LoRAs/Mistral-7B-Instruct-v0.2-4b-r16-task1425 | Lots-of-LoRAs/Mistral-7B-Instruct-v0.2-4b-r16-task1533 |
| Lots-of-LoRAs/Mistral-7B-Instruct-v0.2-4b-r16-task093 | Lots-of-LoRAs/Mistral-7B-Instruct-v0.2-4b-r16-task389 |
| Lots-of-LoRAs/Mistral-7B-Instruct-v0.2-4b-r16-task670 | Lots-of-LoRAs/Mistral-7B-Instruct-v0.2-4b-r16-task461 |
| Lots-of-LoRAs/Mistral-7B-Instruct-v0.2-4b-r16-task113 | Lots-of-LoRAs/Mistral-7B-Instruct-v0.2-4b-r16-task066 |
| Lots-of-LoRAs/Mistral-7B-Instruct-v0.2-4b-r16-task497 | Lots-of-LoRAs/Mistral-7B-Instruct-v0.2-4b-r16-task343 |
| Lots-of-LoRAs/Mistral-7B-Instruct-v0.2-4b-r16-task908 | Lots-of-LoRAs/Mistral-7B-Instruct-v0.2-4b-r16-task1606 |
| Lots-of-LoRAs/Mistral-7B-Instruct-v0.2-4b-r16-task381 | Lots-of-LoRAs/Mistral-7B-Instruct-v0.2-4b-r16-task355 |
| Lots-of-LoRAs/Mistral-7B-Instruct-v0.2-4b-r16-task850 | Lots-of-LoRAs/Mistral-7B-Instruct-v0.2-4b-r16-task1557 |
| Lots-of-LoRAs/Mistral-7B-Instruct-v0.2-4b-r16-task356 | Lots-of-LoRAs/Mistral-7B-Instruct-v0.2-4b-r16-task1394 |
| Lots-of-LoRAs/Mistral-7B-Instruct-v0.2-4b-r16-task1428 | Lots-of-LoRAs/Mistral-7B-Instruct-v0.2-4b-r16-task087 |

| | |
|---|---|
| Lots-of-LoRAs/Mistral-7B-Instruct-v0.2-4b-r16-task582 | Lots-of-LoRAs/Mistral-7B-Instruct-v0.2-4b-r16-task770 |
| Lots-of-LoRAs/Mistral-7B-Instruct-v0.2-4b-r16-task035 | Lots-of-LoRAs/Mistral-7B-Instruct-v0.2-4b-r16-task1317 |
| Lots-of-LoRAs/Mistral-7B-Instruct-v0.2-4b-r16-task1203 | Lots-of-LoRAs/Mistral-7B-Instruct-v0.2-4b-r16-task1444 |
| Lots-of-LoRAs/Mistral-7B-Instruct-v0.2-4b-r16-task1585 | Lots-of-LoRAs/Mistral-7B-Instruct-v0.2-4b-r16-task1508 |
| Lots-of-LoRAs/Mistral-7B-Instruct-v0.2-4b-r16-task740 | Lots-of-LoRAs/Mistral-7B-Instruct-v0.2-4b-r16-task429 |
| Lots-of-LoRAs/Mistral-7B-Instruct-v0.2-4b-r16-task1427 | Lots-of-LoRAs/Mistral-7B-Instruct-v0.2-4b-r16-task328 |
| Lots-of-LoRAs/Mistral-7B-Instruct-v0.2-4b-r16-task955 | Lots-of-LoRAs/Mistral-7B-Instruct-v0.2-4b-r16-task130 |
| Lots-of-LoRAs/Mistral-7B-Instruct-v0.2-4b-r16-task161 | Lots-of-LoRAs/Mistral-7B-Instruct-v0.2-4b-r16-task507 |
| Lots-of-LoRAs/Mistral-7B-Instruct-v0.2-4b-r16-task1502 | Lots-of-LoRAs/Mistral-7B-Instruct-v0.2-4b-r16-task505 |
| Lots-of-LoRAs/Mistral-7B-Instruct-v0.2-4b-r16-task633 | Lots-of-LoRAs/Mistral-7B-Instruct-v0.2-4b-r16-task1645 |
| Lots-of-LoRAs/Mistral-7B-Instruct-v0.2-4b-r16-task1486 | Lots-of-LoRAs/Mistral-7B-Instruct-v0.2-4b-r16-task1146 |
| Lots-of-LoRAs/Mistral-7B-Instruct-v0.2-4b-r16-task1380 | Lots-of-LoRAs/Mistral-7B-Instruct-v0.2-4b-r16-task1088 |
| Lots-of-LoRAs/Mistral-7B-Instruct-v0.2-4b-r16-task033 | Lots-of-LoRAs/Mistral-7B-Instruct-v0.2-4b-r16-task085 |
| Lots-of-LoRAs/Mistral-7B-Instruct-v0.2-4b-r16-task1294 | Lots-of-LoRAs/Mistral-7B-Instruct-v0.2-4b-r16-task080 |
| Lots-of-LoRAs/Mistral-7B-Instruct-v0.2-4b-r16-task489 | Lots-of-LoRAs/Mistral-7B-Instruct-v0.2-4b-r16-task1721 |
| Lots-of-LoRAs/Mistral-7B-Instruct-v0.2-4b-r16-task1713 | Lots-of-LoRAs/Mistral-7B-Instruct-v0.2-4b-r16-task721 |
| Lots-of-LoRAs/Mistral-7B-Instruct-v0.2-4b-r16-task1403 | Lots-of-LoRAs/Mistral-7B-Instruct-v0.2-4b-r16-task746 |
| Lots-of-LoRAs/Mistral-7B-Instruct-v0.2-4b-r16-task728 | Lots-of-LoRAs/Mistral-7B-Instruct-v0.2-4b-r16-task889 |
| Lots-of-LoRAs/Mistral-7B-Instruct-v0.2-4b-r16-task1583 | Lots-of-LoRAs/Mistral-7B-Instruct-v0.2-4b-r16-task1665 |
| Lots-of-LoRAs/Mistral-7B-Instruct-v0.2-4b-r16-task708 | Lots-of-LoRAs/Mistral-7B-Instruct-v0.2-4b-r16-task1419 |
| Lots-of-LoRAs/Mistral-7B-Instruct-v0.2-4b-r16-task963 | Lots-of-LoRAs/Mistral-7B-Instruct-v0.2-4b-r16-task454 |
| Lots-of-LoRAs/Mistral-7B-Instruct-v0.2-4b-r16-task308 | Lots-of-LoRAs/Mistral-7B-Instruct-v0.2-4b-r16-task828 |
| Lots-of-LoRAs/Mistral-7B-Instruct-v0.2-4b-r16-task579 | Lots-of-LoRAs/Mistral-7B-Instruct-v0.2-4b-r16-task753 |
| Lots-of-LoRAs/Mistral-7B-Instruct-v0.2-4b-r16-task1404 | Lots-of-LoRAs/Mistral-7B-Instruct-v0.2-4b-r16-task1201 |
| Lots-of-LoRAs/Mistral-7B-Instruct-v0.2-4b-r16-task901 | Lots-of-LoRAs/Mistral-7B-Instruct-v0.2-4b-r16-task1567 |
| Lots-of-LoRAs/Mistral-7B-Instruct-v0.2-4b-r16-task1319 | Lots-of-LoRAs/Mistral-7B-Instruct-v0.2-4b-r16-task858 |
| Lots-of-LoRAs/Mistral-7B-Instruct-v0.2-4b-r16-task1200 | Lots-of-LoRAs/Mistral-7B-Instruct-v0.2-4b-r16-task492 |
| Lots-of-LoRAs/Mistral-7B-Instruct-v0.2-4b-r16-task675 | Lots-of-LoRAs/Mistral-7B-Instruct-v0.2-4b-r16-task094 |
| Lots-of-LoRAs/Mistral-7B-Instruct-v0.2-4b-r16-task1506 | Lots-of-LoRAs/Mistral-7B-Instruct-v0.2-4b-r16-task694 |
| Lots-of-LoRAs/Mistral-7B-Instruct-v0.2-4b-r16-task614 | Lots-of-LoRAs/Mistral-7B-Instruct-v0.2-4b-r16-task1390 |
| Lots-of-LoRAs/Mistral-7B-Instruct-v0.2-4b-r16-task1355 | Lots-of-LoRAs/Mistral-7B-Instruct-v0.2-4b-r16-task714 |
| Lots-of-LoRAs/Mistral-7B-Instruct-v0.2-4b-r16-task457 | Lots-of-LoRAs/Mistral-7B-Instruct-v0.2-4b-r16-task1565 |
| Lots-of-LoRAs/Mistral-7B-Instruct-v0.2-4b-r16-task834 | Lots-of-LoRAs/Mistral-7B-Instruct-v0.2-4b-r16-task642 |
| Lots-of-LoRAs/Mistral-7B-Instruct-v0.2-4b-r16-task732 | Lots-of-LoRAs/Mistral-7B-Instruct-v0.2-4b-r16-task1605 |
| Lots-of-LoRAs/Mistral-7B-Instruct-v0.2-4b-r16-task326 | Lots-of-LoRAs/Mistral-7B-Instruct-v0.2-4b-r16-task1292 |
| Lots-of-LoRAs/Mistral-7B-Instruct-v0.2-4b-r16-task716 | Lots-of-LoRAs/Mistral-7B-Instruct-v0.2-4b-r16-task1479 |
| Lots-of-LoRAs/Mistral-7B-Instruct-v0.2-4b-r16-task1147 | Lots-of-LoRAs/Mistral-7B-Instruct-v0.2-4b-r16-task153 |
| Lots-of-LoRAs/Mistral-7B-Instruct-v0.2-4b-r16-task1495 | Lots-of-LoRAs/Mistral-7B-Instruct-v0.2-4b-r16-task1196 |
| Lots-of-LoRAs/Mistral-7B-Instruct-v0.2-4b-r16-task1489 | Lots-of-LoRAs/Mistral-7B-Instruct-v0.2-4b-r16-task294 |
| Lots-of-LoRAs/Mistral-7B-Instruct-v0.2-4b-r16-task157 | Lots-of-LoRAs/Mistral-7B-Instruct-v0.2-4b-r16-task147 |
| Lots-of-LoRAs/Mistral-7B-Instruct-v0.2-4b-r16-task1197 | Lots-of-LoRAs/Mistral-7B-Instruct-v0.2-4b-r16-task754 |
| Lots-of-LoRAs/Mistral-7B-Instruct-v0.2-4b-r16-task1599 | Lots-of-LoRAs/Mistral-7B-Instruct-v0.2-4b-r16-task1420 |
| Lots-of-LoRAs/Mistral-7B-Instruct-v0.2-4b-r16-task1285 | Lots-of-LoRAs/Mistral-7B-Instruct-v0.2-4b-r16-task587 |
| Lots-of-LoRAs/Mistral-7B-Instruct-v0.2-4b-r16-task1338 | Lots-of-LoRAs/Mistral-7B-Instruct-v0.2-4b-r16-task116 |
| Lots-of-LoRAs/Mistral-7B-Instruct-v0.2-4b-r16-task713 | Lots-of-LoRAs/Mistral-7B-Instruct-v0.2-4b-r16-task431 |
| Lots-of-LoRAs/Mistral-7B-Instruct-v0.2-4b-r16-task067 | Lots-of-LoRAs/Mistral-7B-Instruct-v0.2-4b-r16-task934 |
| Lots-of-LoRAs/Mistral-7B-Instruct-v0.2-4b-r16-task617 | Lots-of-LoRAs/Mistral-7B-Instruct-v0.2-4b-r16-task696 |
| Lots-of-LoRAs/Mistral-7B-Instruct-v0.2-4b-r16-task846 | Lots-of-LoRAs/Mistral-7B-Instruct-v0.2-4b-r16-task933 |
| Lots-of-LoRAs/Mistral-7B-Instruct-v0.2-4b-r16-task865 | Lots-of-LoRAs/Mistral-7B-Instruct-v0.2-4b-r16-task671 |
| Lots-of-LoRAs/Mistral-7B-Instruct-v0.2-4b-r16-task1398 | Lots-of-LoRAs/Mistral-7B-Instruct-v0.2-4b-r16-task1518 |
| Lots-of-LoRAs/Mistral-7B-Instruct-v0.2-4b-r16-task628 | Lots-of-LoRAs/Mistral-7B-Instruct-v0.2-4b-r16-task1286 |
| Lots-of-LoRAs/Mistral-7B-Instruct-v0.2-4b-r16-task1596 | Lots-of-LoRAs/Mistral-7B-Instruct-v0.2-4b-r16-task298 |
| Lots-of-LoRAs/Mistral-7B-Instruct-v0.2-4b-r16-task645 | Lots-of-LoRAs/Mistral-7B-Instruct-v0.2-4b-r16-task903 |
| Lots-of-LoRAs/Mistral-7B-Instruct-v0.2-4b-r16-task594 | Lots-of-LoRAs/Mistral-7B-Instruct-v0.2-4b-r16-task413 |
| Lots-of-LoRAs/Mistral-7B-Instruct-v0.2-4b-r16-task719 | Lots-of-LoRAs/Mistral-7B-Instruct-v0.2-4b-r16-task672 |
| Lots-of-LoRAs/Mistral-7B-Instruct-v0.2-4b-r16-task1418 | Lots-of-LoRAs/Mistral-7B-Instruct-v0.2-4b-r16-task475 |
| Lots-of-LoRAs/Mistral-7B-Instruct-v0.2-4b-r16-task357 | Lots-of-LoRAs/Mistral-7B-Instruct-v0.2-4b-r16-task1387 |
| Lots-of-LoRAs/Mistral-7B-Instruct-v0.2-4b-r16-task1711 | Lots-of-LoRAs/Mistral-7B-Instruct-v0.2-4b-r16-task304 |
| Lots-of-LoRAs/Mistral-7B-Instruct-v0.2-4b-r16-task750 | Lots-of-LoRAs/Mistral-7B-Instruct-v0.2-4b-r16-task1320 |
| Lots-of-LoRAs/Mistral-7B-Instruct-v0.2-4b-r16-task034 | Lots-of-LoRAs/Mistral-7B-Instruct-v0.2-4b-r16-task692 |
| Lots-of-LoRAs/Mistral-7B-Instruct-v0.2-4b-r16-task1406 | Lots-of-LoRAs/Mistral-7B-Instruct-v0.2-4b-r16-task119 |
| Lots-of-LoRAs/Mistral-7B-Instruct-v0.2-4b-r16-task211 | Lots-of-LoRAs/Mistral-7B-Instruct-v0.2-4b-r16-task363 |
| Lots-of-LoRAs/Mistral-7B-Instruct-v0.2-4b-r16-task1087 | Lots-of-LoRAs/Mistral-7B-Instruct-v0.2-4b-r16-task083 |
| Lots-of-LoRAs/Mistral-7B-Instruct-v0.2-4b-r16-task1385 | Lots-of-LoRAs/Mistral-7B-Instruct-v0.2-4b-r16-task1308 |
| Lots-of-LoRAs/Mistral-7B-Instruct-v0.2-4b-r16-task1656 | Lots-of-LoRAs/Mistral-7B-Instruct-v0.2-4b-r16-task892 |
| Lots-of-LoRAs/Mistral-7B-Instruct-v0.2-4b-r16-task499 | Lots-of-LoRAs/Mistral-7B-Instruct-v0.2-4b-r16-task1409 |
| Lots-of-LoRAs/Mistral-7B-Instruct-v0.2-4b-r16-task1207 | Lots-of-LoRAs/Mistral-7B-Instruct-v0.2-4b-r16-task564 |
| Lots-of-LoRAs/Mistral-7B-Instruct-v0.2-4b-r16-task325 | Lots-of-LoRAs/Mistral-7B-Instruct-v0.2-4b-r16-task206 |
| Lots-of-LoRAs/Mistral-7B-Instruct-v0.2-4b-r16-task890 | Lots-of-LoRAs/Mistral-7B-Instruct-v0.2-4b-r16-task1520 |
| Lots-of-LoRAs/Mistral-7B-Instruct-v0.2-4b-r16-task703 | Lots-of-LoRAs/Mistral-7B-Instruct-v0.2-4b-r16-task318 |
| Lots-of-LoRAs/Mistral-7B-Instruct-v0.2-4b-r16-task366 | Lots-of-LoRAs/Mistral-7B-Instruct-v0.2-4b-r16-task335 |
| Lots-of-LoRAs/Mistral-7B-Instruct-v0.2-4b-r16-task600 | Lots-of-LoRAs/Mistral-7B-Instruct-v0.2-4b-r16-task477 |
| Lots-of-LoRAs/Mistral-7B-Instruct-v0.2-4b-r16-task138 | Lots-of-LoRAs/Mistral-7B-Instruct-v0.2-4b-r16-task291 |
| Lots-of-LoRAs/Mistral-7B-Instruct-v0.2-4b-r16-task074 | Lots-of-LoRAs/Mistral-7B-Instruct-v0.2-4b-r16-task1389 |
| Lots-of-LoRAs/Mistral-7B-Instruct-v0.2-4b-r16-task192 | Lots-of-LoRAs/Mistral-7B-Instruct-v0.2-4b-r16-task316 |
| Lots-of-LoRAs/Mistral-7B-Instruct-v0.2-4b-r16-task1609 | Lots-of-LoRAs/Mistral-7B-Instruct-v0.2-4b-r16-task296 |
| Lots-of-LoRAs/Mistral-7B-Instruct-v0.2-4b-r16-task666 | Lots-of-LoRAs/Mistral-7B-Instruct-v0.2-4b-r16-task1582 |
| Lots-of-LoRAs/Mistral-7B-Instruct-v0.2-4b-r16-task1728 | Lots-of-LoRAs/Mistral-7B-Instruct-v0.2-4b-r16-task701 |
| Lots-of-LoRAs/Mistral-7B-Instruct-v0.2-4b-r16-task275 | Lots-of-LoRAs/Mistral-7B-Instruct-v0.2-4b-r16-task107 |
| Lots-of-LoRAs/Mistral-7B-Instruct-v0.2-4b-r16-task079 | Lots-of-LoRAs/Mistral-7B-Instruct-v0.2-4b-r16-task1157 |
| Lots-of-LoRAs/Mistral-7B-Instruct-v0.2-4b-r16-task1167 | Lots-of-LoRAs/Mistral-7B-Instruct-v0.2-4b-r16-task403 |
| Lots-of-LoRAs/Mistral-7B-Instruct-v0.2-4b-r16-task359 | Lots-of-LoRAs/Mistral-7B-Instruct-v0.2-4b-r16-task517 |
| Lots-of-LoRAs/Mistral-7B-Instruct-v0.2-4b-r16-task351 | Lots-of-LoRAs/Mistral-7B-Instruct-v0.2-4b-r16-task964 |
| Lots-of-LoRAs/Mistral-7B-Instruct-v0.2-4b-r16-task904 | Lots-of-LoRAs/Mistral-7B-Instruct-v0.2-4b-r16-task1148 |
| Lots-of-LoRAs/Mistral-7B-Instruct-v0.2-4b-r16-task879 | Lots-of-LoRAs/Mistral-7B-Instruct-v0.2-4b-r16-task636 |
| Lots-of-LoRAs/Mistral-7B-Instruct-v0.2-4b-r16-task1509 | Lots-of-LoRAs/Mistral-7B-Instruct-v0.2-4b-r16-task207 |

| | |
|---|---|
| Lots-of-LoRAs/Mistral-7B-Instruct-v0.2-4b-r16-task228 | Lots-of-LoRAs/Mistral-7B-Instruct-v0.2-4b-r16-task209 |
| Lots-of-LoRAs/Mistral-7B-Instruct-v0.2-4b-r16-task128 | Lots-of-LoRAs/Mistral-7B-Instruct-v0.2-4b-r16-task710 |
| Lots-of-LoRAs/Mistral-7B-Instruct-v0.2-4b-r16-task1322 | Lots-of-LoRAs/Mistral-7B-Instruct-v0.2-4b-r16-task163 |
| Lots-of-LoRAs/Mistral-7B-Instruct-v0.2-4b-r16-task178 | Lots-of-LoRAs/Mistral-7B-Instruct-v0.2-4b-r16-task089 |
| Lots-of-LoRAs/Mistral-7B-Instruct-v0.2-4b-r16-task700 | Lots-of-LoRAs/Mistral-7B-Instruct-v0.2-4b-r16-task1581 |
| Lots-of-LoRAs/Mistral-7B-Instruct-v0.2-4b-r16-task927 | Lots-of-LoRAs/Mistral-7B-Instruct-v0.2-4b-r16-task101 |
| Lots-of-LoRAs/Mistral-7B-Instruct-v0.2-4b-r16-task123 | Lots-of-LoRAs/Mistral-7B-Instruct-v0.2-4b-r16-task1321 |
| Lots-of-LoRAs/Mistral-7B-Instruct-v0.2-4b-r16-task550 | Lots-of-LoRAs/Mistral-7B-Instruct-v0.2-4b-r16-task129 |
| Lots-of-LoRAs/Mistral-7B-Instruct-v0.2-4b-r16-task393 | Lots-of-LoRAs/Mistral-7B-Instruct-v0.2-4b-r16-task1214 |
| Lots-of-LoRAs/Mistral-7B-Instruct-v0.2-4b-r16-task277 | Lots-of-LoRAs/Mistral-7B-Instruct-v0.2-4b-r16-task1447 |
| Lots-of-LoRAs/Mistral-7B-Instruct-v0.2-4b-r16-task324 | Lots-of-LoRAs/Mistral-7B-Instruct-v0.2-4b-r16-task455 |
| Lots-of-LoRAs/Mistral-7B-Instruct-v0.2-4b-r16-task725 | Lots-of-LoRAs/Mistral-7B-Instruct-v0.2-4b-r16-task365 |
| Lots-of-LoRAs/Mistral-7B-Instruct-v0.2-4b-r16-task1316 | Lots-of-LoRAs/Mistral-7B-Instruct-v0.2-4b-r16-task1199 |
| Lots-of-LoRAs/Mistral-7B-Instruct-v0.2-4b-r16-task717 | Lots-of-LoRAs/Mistral-7B-Instruct-v0.2-4b-r16-task245 |
| Lots-of-LoRAs/Mistral-7B-Instruct-v0.2-4b-r16-task874 | Lots-of-LoRAs/Mistral-7B-Instruct-v0.2-4b-r16-task925 |
| Lots-of-LoRAs/Mistral-7B-Instruct-v0.2-4b-r16-task380 | Lots-of-LoRAs/Mistral-7B-Instruct-v0.2-4b-r16-task1712 |
| Lots-of-LoRAs/Mistral-7B-Instruct-v0.2-4b-r16-task1504 | Lots-of-LoRAs/Mistral-7B-Instruct-v0.2-4b-r16-task619 |
| Lots-of-LoRAs/Mistral-7B-Instruct-v0.2-4b-r16-task590 | Lots-of-LoRAs/Mistral-7B-Instruct-v0.2-4b-r16-task1186 |
| Lots-of-LoRAs/Mistral-7B-Instruct-v0.2-4b-r16-task736 | Lots-of-LoRAs/Mistral-7B-Instruct-v0.2-4b-r16-task069 |
| Lots-of-LoRAs/Mistral-7B-Instruct-v0.2-4b-r16-task377 | Lots-of-LoRAs/Mistral-7B-Instruct-v0.2-4b-r16-task181 |
| Lots-of-LoRAs/Mistral-7B-Instruct-v0.2-4b-r16-task859 | Lots-of-LoRAs/Mistral-7B-Instruct-v0.2-4b-r16-task144 |
| Lots-of-LoRAs/Mistral-7B-Instruct-v0.2-4b-r16-task632 | Lots-of-LoRAs/Mistral-7B-Instruct-v0.2-4b-r16-task641 |
| Lots-of-LoRAs/Mistral-7B-Instruct-v0.2-4b-r16-task064 | Lots-of-LoRAs/Mistral-7B-Instruct-v0.2-4b-r16-task630 |
| Lots-of-LoRAs/Mistral-7B-Instruct-v0.2-4b-r16-task1154 | Lots-of-LoRAs/Mistral-7B-Instruct-v0.2-4b-r16-task390 |
| Lots-of-LoRAs/Mistral-7B-Instruct-v0.2-4b-r16-task1188 | Lots-of-LoRAs/Mistral-7B-Instruct-v0.2-4b-r16-task625 |
| Lots-of-LoRAs/Mistral-7B-Instruct-v0.2-4b-r16-task607 | Lots-of-LoRAs/Mistral-7B-Instruct-v0.2-4b-r16-task495 |
| Lots-of-LoRAs/Mistral-7B-Instruct-v0.2-4b-r16-task1189 | Lots-of-LoRAs/Mistral-7B-Instruct-v0.2-4b-r16-task398 |
| Lots-of-LoRAs/Mistral-7B-Instruct-v0.2-4b-r16-task108 | Lots-of-LoRAs/Mistral-7B-Instruct-v0.2-4b-r16-task1347 |
| Lots-of-LoRAs/Mistral-7B-Instruct-v0.2-4b-r16-task1541 | Lots-of-LoRAs/Mistral-7B-Instruct-v0.2-4b-r16-task202 |
| Lots-of-LoRAs/Mistral-7B-Instruct-v0.2-4b-r16-task1723 | Lots-of-LoRAs/Mistral-7B-Instruct-v0.2-4b-r16-task1669 |
| Lots-of-LoRAs/Mistral-7B-Instruct-v0.2-4b-r16-task1089 | Lots-of-LoRAs/Mistral-7B-Instruct-v0.2-4b-r16-task1584 |
| Lots-of-LoRAs/Mistral-7B-Instruct-v0.2-4b-r16-task081 | Lots-of-LoRAs/Mistral-7B-Instruct-v0.2-4b-r16-task329 |
| Lots-of-LoRAs/Mistral-7B-Instruct-v0.2-4b-r16-task691 | Lots-of-LoRAs/Mistral-7B-Instruct-v0.2-4b-r16-task588 |
| Lots-of-LoRAs/Mistral-7B-Instruct-v0.2-4b-r16-task1593 | Lots-of-LoRAs/Mistral-7B-Instruct-v0.2-4b-r16-task724 |
| Lots-of-LoRAs/Mistral-7B-Instruct-v0.2-4b-r16-task149 | Lots-of-LoRAs/Mistral-7B-Instruct-v0.2-4b-r16-task1449 |
| Lots-of-LoRAs/Mistral-7B-Instruct-v0.2-4b-r16-task1313 | Lots-of-LoRAs/Mistral-7B-Instruct-v0.2-4b-r16-task1453 |
| Lots-of-LoRAs/Mistral-7B-Instruct-v0.2-4b-r16-task905 | Lots-of-LoRAs/Mistral-7B-Instruct-v0.2-4b-r16-task704 |
| Lots-of-LoRAs/Mistral-7B-Instruct-v0.2-4b-r16-task585 | Lots-of-LoRAs/Mistral-7B-Instruct-v0.2-4b-r16-task1209 |
| Lots-of-LoRAs/Mistral-7B-Instruct-v0.2-4b-r16-task249 | Lots-of-LoRAs/Mistral-7B-Instruct-v0.2-4b-r16-task1386 |
| Lots-of-LoRAs/Mistral-7B-Instruct-v0.2-4b-r16-task1400 | Lots-of-LoRAs/Mistral-7B-Instruct-v0.2-4b-r16-task751 |
| Lots-of-LoRAs/Mistral-7B-Instruct-v0.2-4b-r16-task1332 | Lots-of-LoRAs/Mistral-7B-Instruct-v0.2-4b-r16-task674 |
| Lots-of-LoRAs/Mistral-7B-Instruct-v0.2-4b-r16-task379 | Lots-of-LoRAs/Mistral-7B-Instruct-v0.2-4b-r16-task243 |
| Lots-of-LoRAs/Mistral-7B-Instruct-v0.2-4b-r16-task1318 | Lots-of-LoRAs/Mistral-7B-Instruct-v0.2-4b-r16-task428 |
| Lots-of-LoRAs/Mistral-7B-Instruct-v0.2-4b-r16-task488 | Lots-of-LoRAs/Mistral-7B-Instruct-v0.2-4b-r16-task705 |
| Lots-of-LoRAs/Mistral-7B-Instruct-v0.2-4b-r16-task698 | Lots-of-LoRAs/Mistral-7B-Instruct-v0.2-4b-r16-task1601 |
| Lots-of-LoRAs/Mistral-7B-Instruct-v0.2-4b-r16-task861 | Lots-of-LoRAs/Mistral-7B-Instruct-v0.2-4b-r16-task1510 |
| Lots-of-LoRAs/Mistral-7B-Instruct-v0.2-4b-r16-task077 | Lots-of-LoRAs/Mistral-7B-Instruct-v0.2-4b-r16-task509 |
| Lots-of-LoRAs/Mistral-7B-Instruct-v0.2-4b-r16-task734 | Lots-of-LoRAs/Mistral-7B-Instruct-v0.2-4b-r16-task720 |
| Lots-of-LoRAs/Mistral-7B-Instruct-v0.2-4b-r16-task1210 | Lots-of-LoRAs/Mistral-7B-Instruct-v0.2-4b-r16-task284 |
| Lots-of-LoRAs/Mistral-7B-Instruct-v0.2-4b-r16-task584 | Lots-of-LoRAs/Mistral-7B-Instruct-v0.2-4b-r16-task105 |
| Lots-of-LoRAs/Mistral-7B-Instruct-v0.2-4b-r16-task330 | Lots-of-LoRAs/Mistral-7B-Instruct-v0.2-4b-r16-task923 |
| Lots-of-LoRAs/Mistral-7B-Instruct-v0.2-4b-r16-task319 | Lots-of-LoRAs/Mistral-7B-Instruct-v0.2-4b-r16-task400 |
| Lots-of-LoRAs/Mistral-7B-Instruct-v0.2-4b-r16-task246 | Lots-of-LoRAs/Mistral-7B-Instruct-v0.2-4b-r16-task726 |
| Lots-of-LoRAs/Mistral-7B-Instruct-v0.2-4b-r16-task1568 | Lots-of-LoRAs/Mistral-7B-Instruct-v0.2-4b-r16-task1442 |
| Lots-of-LoRAs/Mistral-7B-Instruct-v0.2-4b-r16-task1640 | Lots-of-LoRAs/Mistral-7B-Instruct-v0.2-4b-r16-task280 |

Figure 9 presents the aggregated results across all layers, with error bars representing the standard deviation. For reference, the eigenvalue (scree) plot from Figure 3b is also reproduced in Figure 9. This plot depicts the proportion of variance explained by each principal component, computed across all weight matrices and layers from 500 independently trained Mistral models. The concentration of variance within the top $k$ components reveals the presence of a consistent low-dimensional subspace, offering strong empirical support for the universal subspace hypothesis.

The individual plots provide spectral analysis results for the key, query, and value matrices from all 32 layers of all 500 Mistral models. For clarity, only the top 128 principal directions are visualized, representing a subset of the full component basis. This truncation mitigates the visual distortion caused by the long tail of near-zero eigenvalues beyond the universal subspace, which would otherwise dominate the graph without contributing meaningful information.

To test subspace expressiveness, we reconstruct LoRA weights for both 5 seen (IID) and unseen (OOD) tasks by projecting them into the universal subspace. As shown in Figure 4, the reconstructed models retain high performance in both cases. In contrast, projection into the residual *Secondary Subspace* leads to a sharp performance drop, underscoring the importance of the principal subspace. Our method is also 19× more memory efficient, as it eliminates the need to store all 500 LoRAs.

Table 8: Models from HuggingFace used for the Universal Stable Diffusion-XL subspace extraction

| | | | |
|---|---|---|---|
| alphonse-mucha-style | directors-coen-brothers-style | larry-carlson-style | rene-magritte-style |
| beeple-mike-winkelmann-style | director-sergei-eisenstein-style | lascaux | richard-corben-style |
| character-design | director-sofia-coppola-style | laurel-burch-style | richard-dadd-style |
| director-christopher-nolan-style | director-terrence-malick-style | lawrence-alma-tadema-style | richard-hescox-style |
| director-lars-von-trier-style | director-tim-burton-style | leonid-afremov-style | richard-scarry-style |
| director-ridley-scott-style | director-wes-anderson-style | leonora-carrington-style | robert-adams-style |
| director-stanley-kubrick-style | director-wong-kar-wai-style | levitating-cube | robert-crumb-style |
| director-zhang-yimou-style | director-yorgos-lanthimos-style | liam-wong-style | robert-rauschenberg-style |
| olafur-eliasson-style | dixit-card-generator | lotte-reiniger-style | rodney-matthews-style |
| origami | dressed-animals | louis-comfort-tiffany-style | roger-ballen-style |
| simone-martini-style | dripping-art | lovis-corinth-style | roger-deakins-style |
| studio-ghibli-style | edward-gorey-style | lucas-cranach-style | romare-bearden-style |
| ukiyo-e-art | elizabeth-gadd-style | luc-schuiten-style | ryoji-ikeda-style |
| wu-guanzhong-style | erik-johansson-style | lyonel-feininger-style | sacha-goldberger-style |
| 1987-action-figure-playset-packaging | erik-madigan-heck-style | made-of-iridescent-foil | salomon-van-ruysdael-style |
| aardman-animations-style | euan-uglow-style | makoto-shinkai-style | sam-spratt-style |
| akos-major-style | felipe-pantone-style | marc-silvestri-style | sandy-skoglund-style |
| albumen-print | filip-hodas-style | marianna-rothen-style | santiago-caruso-style |
| alec-soth-style | folk-art | maria-sibylla-merian-style | shaun-tan-style |
| alejandro-jodorowsky-style | gabriel-pacheco-style | mark-catesby-style | shepard-fairey-style |
| alessandro-gottardo-style | gemma-correll-style | mark-ryden-style | sidney-nolan-style |
| alex-andreev-style | george-condo-style | martin-whatson-style | simon-stalenhag-style |
| alex-gross-style | gilbert-garcin-style | mary-cassatt-style | skottie-young-style |
| alfred-augustus-glendening-style | gregory-crewdson-style | maurice-de-vlaminck-style | sofonisba-anguissola-style |
| alex-pardee-style | gustave-dore-style | maurice-prendergast-style | sophie-gengembre-anderson-style |
| alternate-realities | hasui-kawase-style | maxfield-parrish-style | stained-glass-portrait |
| ando-fuchs-style | hiroshi-nagai-style | maxime-maufra-style | stanley-donwood-style |
| andre-derain-style | infrared-photos | mike-mignola-style | stephan-martiniere-style |
| andrei-tarkovsky-style | isometric-cutaway | mikhail-vrubel-style | stephen-gammell-style |
| andrew-wyeth-style | ivan-bilibin-style | moebius-jean-giraud-style | stop-motion-animation |
| angus-mckie-style | james-c-christensen-style | movie-poster | surreal-collage |
| anna-maria-garthwaite-style | james-jean-style | moving-meditations | surreal-harmony |
| atey-ghailan-style | james-r-eads-style | nadav-kander-style | surreal-plate |
| audrey-kawasaki-style | james-turrell-style | natalia-goncharova-style | syd-mead-style |
| avant-garde-fashion | jan-brueghel-style | n-c-wyeth-style | synthwave-t-shirt |
| banksy-style | jan-svankmajer-style | needlepoint | teamlab-style |
| bas-relief | jan-van-eyck-style | neon-night | terry-gilliam-style |
| century-botanical-illustration | jan-van-goyen-style | nicolas-poussin-style | thomas-cole-style |
| christopher-balaskas-style | j-c-leyendecker-style | noah-bradley-style | thomas-kinkade-style |
| christopher-ryan-mckenney-style | jean-baptiste-camille-corot-style | ohara-koson-style | thomas-moran-style |
| clay-animation | jean-baptiste-monge-style | okuda-san-miguel-style | thomas-schaller-style |
| color-palette | jean-baptiste-simeon-chardin-style | olly-moss-style | tim-walker-style |
| craig-mullins-style | jean-metzinger-style | op-art | tintoretto-style |
| crocheted | jean-michel-basquiat-style | parralel-dimensions | todd-hido-style |
| daniel-arsham-style | jessie-willcox-smith-style | pascal-campion-style | tove-jansson-style |
| dark-fantasy | jim-mahfood-style | paul-gustav-fischer-style | tracie-grimwood-style |
| dave-mckean-style | john-albert-bauer-style | paul-laffoley-style | vasily-vereshchagin-style |
| diorama | john-berkey-style | paul-signac-style | vertical-landscapes |
| director-agnes-varda-style | john-blanche-style | peter-doig-style | victor-brauner-style |
| death-stranding | john-constable-style | peter-paul-rubens-style | victor-moscoso-style |
| director-akira-kurosawa-style | john-everett-millais-style | philippe-druillet-style | video-installation |
| director-andrei-zvyagintsev-style | john-harris-style | photographer-elena-helfrecht-style | vintage-postage-stamps |
| director-bong-joon-ho-style | john-james-audubon-style | photographer-flora-borsi-style | weegee-style |
| director-darren-aronofsky-style | john-kenn-mortensen-style | photographer-maren-klemp-style | wendy-froud-style |
| director-david-fincher-style | john-martin-style | photographer-martin-kimbell-style | will-eisner-style |
| director-david-lynch-style | john-singer-sargent-style | photographer-reuben-wu-style | willem-haenraets-style |
| cute-animals | john-singleton-copley-style | pierre-auguste-renoir-style | willem-van-aelst-style |
| ben-aronson-style | john-william-waterhouse-style | pierre-bonnard-style | william-langson-lathrop-style |
| director-emir-kusturica-style | joseph-wright-of-derby-style | pieter-claesz-style | william-mctaggart-style |
| director-gaspar-noe-style | josh-agle-style | punk-collage | william-merritt-chase-style |
| director-jean-pierre-jeunet-style | josh-kirby-style | quentin-blake-style | winslow-homer-style |
| director-krzysztof-kieslowski-style | jules-bastien-lepage-style | raimonds-staprans-style | worthington-whittredge-style |
| director-martin-scorsese-style | kate-greenaway-style | ralph-bakshi-style | yaacov-agam-style |
| director-nicolas-winding-refn-style | kay-nielsen-style | ralph-steadman-style | yoh-nagao-style |
| director-park-chan-wook-style | kilian-eng-style | randolph-caldecott-style | yves-klein-style |
| director-pedro-almodovar-style | kirigami | ray-caesar-style | zanele-muholi-style |
| director-quentin-tarantino-style | konstantin-korovin-style | remedios-varo-style | |

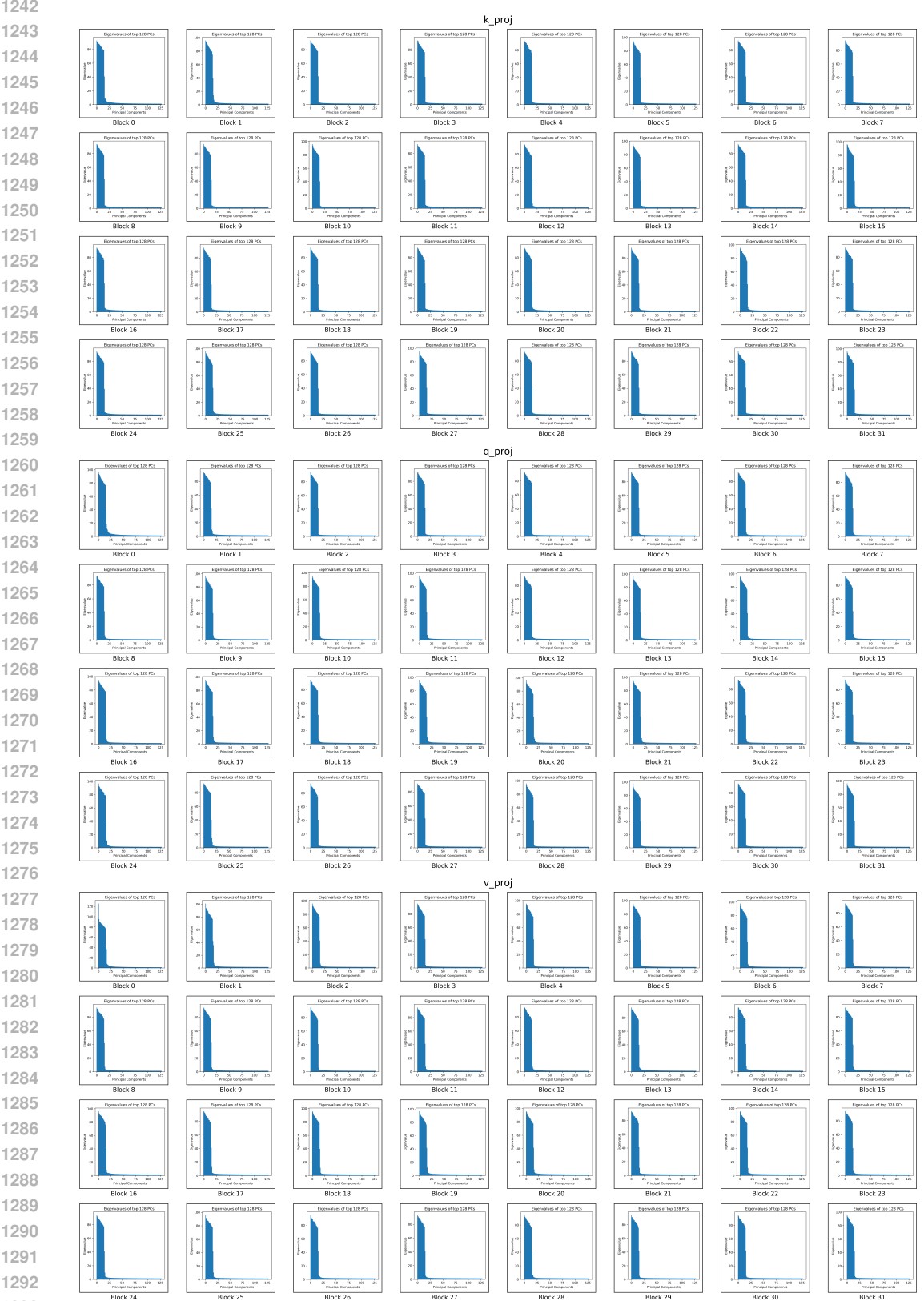

Figure 9: Layerwise Eigenvalue Plots of 500 Mistral-7B-Instruct-v0.2 models. Each layer has 3 sets of parameters - $k\_proj, q\_proj, v\_proj$

**Universal SDXL experiment details**   Our second experiment involves the complex and multimodal task of Text-to-Image generation using the Stable Diffusion-XL model Podell et al. (2023). We extract our low rank universal subspace from publicly available LoRA models on HuggingFace repository von Platen et al. (2022) - Table 8 lists all the SDXL models that we used to extract the Universal Subspace. As can be seen in Table 8, the models range wildly in styles on which they were finetuned. The fact that all these diverse models can be represented by a single low rank universal subspace model strongly verifies our hypothesis. We use top 16 components and 30 denoising steps. For each experiment model shown in Table 1 and Figure 5, that LoRA model is reconstructed using a universal subspace created using rest of the available LoRA adapters, essentially confirming the generalization capability of this subspace.

We then use this single SDXL universal subspace to generate images with similar styles to evaluate whether this subspace is capable of doing so, by projecting randomly chosen LoRA models into this subspace. Figure 5 shows that our universal subspace matches the visual quality and style nuances of individual LoRAs, resulting in significant memory savings. Table 1 shows quantitative results for our Universal subspace in terms of CLIP scores, where interestingly we can see that our Universal Subspace outperforms the individual LoRA models. This improvement may be attributed to our Universal SDXL removing noise from the subspace - a phenomenon previously observed by Sharma et al. (2023). The styles used in Table 1, which are in Table 8 are (from Style 1 to Style 10) Ukiyo-e Style, Todd Hildo Style , Olly Moss Style , Needlepoint Style , Studio Ghibli Style, Surreal Harmony Style , Dressed Animal Style , Lascaux Cave Art Style , Kirigami Style , Yaacov Agam Style.

## B.2   Low rank shared universal subspaces in classical weights

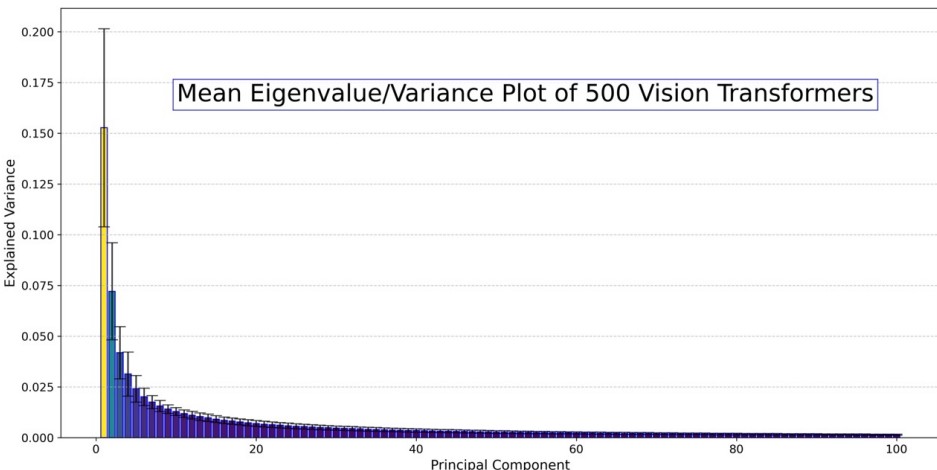

Figure 10: Spectral analysis of the Vision Transformer (ViT-base-patch16-224) model: Aggregated eigenvalue (scree) plot across  500 ViT models and all layers. The plot demonstrates that the majority of the variance is consistently captured by the top 16 principal directions, indicating the presence of a shared low-dimensional universal subspace.

In order to further solidify the evidence for our universal subspace hypothesis, we show that this universality does extend beyond adapter models to conventional weights. We do not focus on convolutional weight parameters as they can simply be equated with fully connected layers (Ma & Lu, 2017), and have been shown, in limited scope, to match Gabor-like filters (Krizhevsky et al., 2012). Therefore, our analysis trivially extends to these kinds of parameters as well. However, there are a few practical differences between the low rank adapter and classical weight subspace analysis. The classical weight subspace analysis is more computationally expensive relative to the LoRA one due to high dimensionality of the parameters, but in effect, same. Additionally, the number of sufficiently well trained models is understandably fewer than LoRA models. Further, there is also higher variance in terms of model quality in the classical weights as it is harder to optimize these models as compared to LoRA which often are optimized from a good initialization point (the pretrained base model). An outcome of this is that the universal subspace approximation that we obtain from the publicly

available pretrained models are noisier than their LoRA counterparts. Inspite of this, our universal subspace hypothesis remains validated.

To further support our universal subspace hypothesis, we extend our analysis beyond adapter models to standard full-rank weights. We exclude convolutional parameters from explicit consideration, as they are functionally equivalent to fully connected layers under certain conditions (Ma & Lu, 2017), and their learned representations (e.g., Gabor-like filters) have been studied, in limited scope, in prior work (Krizhevsky et al., 2012). Consequently, our analysis generalizes naturally to convolutional weights as well.

There are, however, practical differences between the subspace analysis of full-rank model weights and that of low-rank adapters. First, analyzing conventional weight matrices is significantly more computationally intensive because of their higher dimensionality. Second, the availability of a large number of independently and sufficiently well-trained models is more limited compared to LoRA models. Third, the classical weight models exhibit greater variance in model quality, since they must be trained from scratch, often without the benefit of a well-optimized initialization, unlike LoRA which builds upon a strong pretrained base.

As a result, the subspaces estimated from classical weights tend to be noisier, and the universality signal is less pronounced. Despite these challenges, we still observe consistent structure in the leading components, lending further empirical support to the universal subspace hypothesis.

**Universal ViT-base-patch16-224 experiment details**    We collect ∼500 pretrained ViT models from HuggingFace, shown in Table 9, spanning very diverse domains — many of which would be considered orthogonal to one another in terms of domain generalization. These models have been trained with varying losses, optimizers, and initializations. These models were used as-is, without curation or access to training data, to reflect real-world variability. Figure 10 shows the summarized scree plot for all relevant layers of ViT (sans first and last layers due to differences in shape and tasks) for all $\sim 500$ ViT models showing that the majority of variance is captured by the top 16 principal directions, revealing a highly compressible, shared subspace across layers. Only the top 100 components are visualized for clarity, although the available subspace is significantly larger, underlying the sparsity of this universal subspace. We observe this for layerwise analysis in Figure 11 as well. For the experimental results presented in Table 3, we randomly choose 4-5 IID and 4-5 OOD models from Table 9 for which evaluation dataset is available, and reconstruct these model weights by projecting them into our 16 component universal subspace. For the OOD case, we ensure that the models being evaluated are not present in the subset used for creating the universal subspace approximation. As seen from the results, our extremely sparse subspace model performs competitively compared to the fully trained versions. It is likely that with more careful choice of principal directions per layer would allow for at par or even better performance.

Table 9: Finetuned Models from HuggingFace used for the Universal Vision Transformer subspace extraction (vit-base-patch16-224)

| | |
|---|---|
| 0.50-200Train-100Test-vit-base | 2025-01-21-16-13-04-vit-base-patch16-224 |
| 2025-02-05-14-22-36-vit-base-patch16-224 | 21BAI1229 |
| Accomodation_room_classification | adam_VitB-p16-224-1e-4-batch_16_epoch_4_classes_24 |
| age_face_detection_base | AIvisionGuard-v2 |
| alea | amns |
| AnimeCharacterClassifierMark1 | autotrain-48ci8-roib9 |
| autotrain-8oqr6-image0807-20 | autotrain-ap-pass-fail-v1 |
| autotrain-g2g80-iwcfm | autotrain-google-vit-13epoch |
| autotrain-ht4es-gbvmt | autotrain-image-classifier-cats-and-dogs |
| autotrain-pknu0-o76h9 | autotrain-s0sds-erede |
| autotrain-test-image-classification | autotrain-vit-base-patch16-224-fog-or-smog-classification |
| beauty-ornot | beer-classifier |
| bg-classif | bigger-chord-finetuned |
| brain-tumor-44 | ButterflyClasifModel |
| camera-type | Caracam |
| cards-vit-base-patch16-224-finetuned-v1 | carmodel |
| cats123 | cats-dogs-2024 |
| cats-dogs-classification | CheXpert-ViT-U-MultiClass |
| CheXpert-ViT-U-SelfTrained | chord-final-model |
| chord_ViT-finetuned | cifar10-lt |
| city_multiclass_classification | clasificador_masas |
| corals_binary_classification | custom |
| detect_meme | dog-breeds-classification |
| dog-cat-demo-20240815 | dog-cats-model |
| dummy_classification_model | dvm-cars-vit-first-5k |
| ecg-image-multilabel-classification | emotion |
| EmotionAgeModel | emotion_model |
| emotion-recognition | emotion_recognition |

| | |
|---|---|
| emotion_recognition_results | emotion-vit |
| face_age_detection_base_v2 | face_age_detection_base_v3_weighted |
| final-run | finetune-cats |
| fine-tuned | finetuned-amazon |
| fine-tuned-augmented | finetuned-bin |
| finetuned-cifar10 | finetuned-indian-food |
| fine-tuned-model | finetuned_model |
| Fine-Tuned_Model | Fine-Tuned_Model2 |
| Fine-Tuned_Model3 | Fine-Tuned_Model3_Transfer_learning |
| finetune-vit-base-patch16-224 | finetune_vit_base_patch16_224_1epoch |
| Flowers | food |
| food-101-finetuned-model | Freshness-Fruit_Vegies |
| frost-vision-v2-google_vit-base-patch16-224 | frost-vision-v2-google_vit-base-patch16-224-v2024-11-09 |
| frost-vision-v2-google_vit-base-patch16-224-v2024-11-11 | frost-vision-v2-google_vit-base-patch16-224-v2024-11-14 |
| fruit_classification | fruits-360-16-7 |
| ft_stable_diffusion | gender |
| giecom-vit-model-clasification-waste | google-vit-base-patch16-224-batch32-lr0.0005-standford-dogs |
| google-vit-base-patch16-224-batch32-lr0.005-standford-dogs | google-vit-base-patch16-224-batch32-lr5e-05-standford-dogs |
| google-vit-base-patch16-224-batch64-lr0.005-standford-dogs | google-vit-base-patch16-224-OrganicAndInorganicWaste-classification |
| google-vit-base-patch16-224-Waste-O-I-classification | hf_vit_format_hap_pretrained_256_128 |
| Human-Action-Recognition-VIT-Base-patch16-224 | human-actions |
| image-classification | image_classification |
| image_strawbery-peach_classifier | isa-vit_model |
| lixg_food_model001 | Maggi-Parle-G_Classifier |
| mammals_multiclass_classification | MemeDetector |
| model | Model |
| model-vit-base-finetuned | MRI_vit |
| my_chest_xray_model | myclass |
| my_classification | MyPetModel |
| out | outputs |
| PagesClassificationModel | physiotheraphy-E2 |
| plant_disease_detection-beans | pokemon_classification |
| pokemon_model | pokemon-vit |
| recaptcha | recycled_waste_classification |
| results | rmsprop_VitB-p16-224-1e-4-batch_16_epoch_4_classes_24 |
| rmsprop_VitB-p16-224-2e-4-batch_16_epoch_4_classes_24 | road-conditions |
| rose_recognition | rotated2 |
| Ruster | S1_M1_R1_vit_42498800 |
| S1_M1_R1_vit_42509509 | S1_M1_R1_ViT_42616100 |
| S1_M1_R2_vit_42498972 | S1_M1_R2_ViT_42618476 |
| S1_M1_R3_vit_42499444 | S1_M1_R3_ViT_42618486 |
| S2_M1_R1_vit_42499480 | S2_M1_R1_ViT_42618522 |
| S2_M1_R2_vit_42499499 | S2_M1_R2_ViT_42618530 |
| S2_M1_R3_vit_42499514 | S2_M1_R3_ViT_42618549 |
| S5_M1_fold1_vit_42499955 | S5_M1_fold1_ViT_42618571 |
| S5_M1_fold2_vit_42499968 | S5_M1_fold2_ViT_42618583 |
| S5_M1_fold3_vit_42499983 | S5_M1_fold3_ViT_42618589 |
| S5_M1_fold4_vit_42499997 | S5_M1_fold4_ViT_42618593 |
| S5_M1_fold5_vit_42500027 | S5_M1_fold5_ViT_42621111 |
| Screenshots_detection_to_classification | sign-lan-model |
| square_run_32_batch | square_run_age_gender |
| square_run_first_vote_full_pic_50 | square_run_first_vote_full_pic_50_age_gender |
| square_run_first_vote_full_pic_75 | square_run_first_vote_full_pic_75_age_gender |
| square_run_second_vote | square_run_second_vote_full_pic_50 |
| square_run_second_vote_full_pic_50_age_gender | square_run_second_vote_full_pic_75 |
| square_run_second_vote_full_pic_75_age_gender | square_run_second_vote_full_pic_age_gender |
| square_run_second_vote_full_pic_stratified | square_run_square_run_first_vote_full_pic_25 |
| square_run_square_run_first_vote_full_pic_25_age | square_run_square_run_first_vote_full_pic_25_age_gender |
| square_run_square_run_first_vote_full_pic_25_age_gender_double_check | square_run_square_run_second_vote_full_pic_25 |
| square_run_square_run_second_vote_full_pic_25_age_gender | square_run_with_16_batch_size |
| square_run_with_actual_16_batch_size | stool-condition-classification |
| swaddling-classifier | swin-tiny-patch4-window7-224-finetuned-eurosat-kornia |
| tarread | telidermai |
| test-cifar-10 | traffic-levels-image-classification |
| Train-Augmentation-vit-base | trainer_output |
| Train-Test-Augmentation-V3D-vit-base | UL_base_classification |
| UL_bedroom_classification | UL_exterior_classification |
| UL_interior_classification | vehicle_multiclass_classification |
| ViT_ASVspoof_DF | vit-augmentation |
| vit-b16-plant-village | vit_base |
| vit-base-1e-4-15ep | vit-base-1e-4-20ep |
| vit-base-1e-4-randaug | vit-base-1stGen-Pokemon-Images |
| vit-base-25ep | Vit-Base-30VN |
| vit-base-3e-5-randaug | vit-base-5e-4 |
| vit-base-add-2-decay | vit-base-augment |
| vit-base-batch-32 | vit-base-beans |
| vit-base-brain-mri | vit-base-cat_or_dog |
| vit-base-change-arg | vit-base-cocoa |
| ViT-Base-Document-Classifier | vit-base-fashion |
| vit-base-finetuned-cephalometric | vit-base-food101 |
| vit-base-fruits-360 | vit-base-hate-meme |
| vit-base-nationality | vit-base-org-plot |
| vit-base-oxford-brain-tumor | vit-base-oxford-brain-tumor_try_stuff |
| vit-base-oxford-brain-tumor_x-ray | vit-base-oxford-iiit-pets |
| vit-base-oxford-pets-krasuluk | vit-base-patch16-224 |

| | |
|---|---|
| vit-base-patch16-224-13_model | vit-base-patch16-224-30-vit |
| vit-base-patch16-224-9models | vit-base-patch16-224-abhi1-finetuned |
| vit-base-patch16-224_augmented-v2_fft | vit-base-patch16-224_augmented-v2_tl |
| vit-base-patch16-224-blur_vs_clean | vit-base-patch16-224-brand |
| vit-base-patch16-224-classifier | vit-base-patch16-224-clothes-filter |
| vit-base-patch16-224-cl-v1 | vit-base-patch16-224-crochets-clothes-classification |
| vit-base-patch16-224-Diastar | vit-base-patch16-224-Diastarallclasses |
| vit-base-patch16-224-dmae-va-U | vit-base-patch16-224-dmae-va-U5-100-iN |
| vit-base-patch16-224-dmae-va-U5-10-45-5e-05 | vit-base-patch16-224-dmae-va-U5-20-45-5e-05 |
| vit-base-patch16-224-dmae-va-U5-40-45-5e-05 | vit-base-patch16-224-dmae-va-U5-42B |
| vit-base-patch16-224-dmae-va-U5-42C | vit-base-patch16-224-dmae-va-U5-42D |
| vit-base-patch16-224-ethos | vit-base-patch16-224-ethos-25 |
| vit-base-patch16-224-ethos-8 | vit-base-patch16-224-ethos-data |
| vit-base-patch16-224-ethosrealdata | vit-base-patch16-224-fatigue |
| vit-base-patch16-224-finalterm | vit-base-patch16-224-finetuned |
| vit-base-patch16-224-finetuned-barkley | vit-base-patch16-224-finetuned-brain-tumor-classification |
| vit-base-patch16-224-finetuned-Brain-Tumor-Classification | vit-base-patch16-224-finetuned-cassava-leaf-disease |
| vit-base-patch16-224-finetuned-cedar | vit-base-patch16-224-finetuned-cifar10 |
| vit-base-patch16-224-finetuned-combinedSpiders | vit-base-patch16-224-finetuned-context-classifier |
| vit-base-patch16-224-finetuned-covid_ct_set_full | vit-base-patch16-224-finetuned-covid_ct_set_resumed |
| vit-base-patch16-224-finetuned-crochets-clothes | vit-base-patch16-224-finetuned-dangerousSpiders |
| vit-base-patch16-224-finetuned-eurosat | vit-base-patch16-224-finetuned-feature-maps-v3 |
| vit-base-patch16-224-finetuned-feature-map-v2 | vit-base-patch16-224-finetuned-fibre |
| vit-base-patch16-224-finetuned-flower | vit-base-patch16-224-finetuned-flower-classify |
| vit-base-patch16-224-finetuned-flowers | vit-base-patch16-224-finetuned-food101 |
| vit-base-patch16-224-finetuned-food102 | vit-base-patch16-224-finetuned-foveated-features |
| vit-base-patch16-224-finetuned-foveated-features-v2 | vit-base-patch16-224-finetuned-galaxy10-decals |
| vit-base-patch16-224-finetuned-hateful-meme-restructured | vit-base-patch16-224-finetuned-hateful-meme-restructured-balanced |
| vit-base-patch16-224-finetuned-imagegpt | vit-base-patch16-224-finetuned-ind-17-imbalanced-aadhaarmask |
| vit-base-patch16-224-finetuned-ind-17-imbalanced-aadhaarmask-new-parameter | vit-base-patch16-224-finetuned-landscape-test |
| vit-base-patch16-224-finetuned-lora-oxford-pets | vit-base-patch16-224-finetuned-masked-hateful-meme-restructured |
| vit-base-patch16-224-finetuned-noh | vit-base-patch16-224-finetuned-original-images |
| vit-base-patch16-224-finetuned-pneumonia-detection | vit-base-patch16-224-finetuned-polyterrasse |
| vit-base-patch16-224-finetuned-skin | vit_base_patch16_224-finetuned-SkinDisease |
| vit-base-patch16-224-finetuned-teeth_dataset | vit-base-patch16-224-finetuned-trash-classifications-albumentations |
| vit-base-patch16-224-finetuned-turquoise | vit-base-patch16-224-finetuned-Visual-Emotional |
| vit-base-patch16-224-finetuned-vit | vit-base-patch16-224-finetune_test |
| vit-base-patch16-224-food101-16-7 | vit-base-patch16-224-food101-24-12 |
| vit-base-patch16-224-for-pre_evaluation | vit-base-patch16-224-fruits-360-16-7 |
| vit-base-patch16-224-high-vit | vit-base-patch16-224-jvadlamudi2 |
| vit-base-patch16-224-masaratti | vit-base-patch16-224-mascotas |
| vit-base-patch16-224-mascotas-DA | vit-base-patch16-224-MSC-dmae |
| vit-base-patch16-224-newly-trained | vit-base-patch16-224-oxford-pets-classification |
| vit-base-patch16-224-perros-y-gatos | vit-base-patch16-224-pure-ViT |
| vit-base-patch16-224-R1-10 | vit-base-patch16-224-R1-40 |
| vit-base-patch16-224-Rado_5 | vit-base-patch16-224_rice-disease-02 |
| vit-base-patch16-224_rice-leaf-disease-augmented_fft | vit-base-patch16-224_rice-leaf-disease-augmented_tl |
| vit-base-patch16-224_rice-leaf-disease-augmented-v4_fft | vit-base-patch16-224_rice-leaf-disease-augmented-v4_tl |
| vit-base-patch16-224_rice-leaf-disease-augmented-v4_v5_fft | vit-base-patch16-224_rice-leaf-disease-augmented-v4_v5_pft |
| vit-base-patch16-224-rotated-dungeons-v101 | vit-base-patch16-224-rotated-dungeons-v103 |
| vit-base-patch16-224-RU2-10 | vit-base-patch16-224-RU2-40 |
| vit-base-patch16-224-RU3-10 | vit-base-patch16-224-RU3-40 |
| vit-base-patch16-224-RU4-10 | vit-base-patch16-224-RU4-40 |
| vit-base-patch16-224-RU5-10 | vit-base-patch16-224-RU5-10-8 |
| vit-base-patch16-224-RU5-40 | vit-base-patch16-224-RU9-24 |
| vit-base-patch16-224-RX1-24 | vit-base-patch16-224-RX2-12 |
| vit-base-patch16-224-RXL1-24 | vit-base-patch16-224-type |
| vit-base-patch16-224-U6-10 | vit-base-patch16-224-U7-10 |
| vit-base-patch16-224-U8-10 | vit-base-patch16-224-U8-10b |
| vit-base-patch16-224-U8-10c | vit-base-patch16-224-U8-40 |
| vit-base-patch16-224-U8-40b | vit-base-patch16-224-U8-40c |
| vit-base-patch16-224-U8-40d | vit-base-patch16-224-ve-b-U10-12 |
| vit-base-patch16-224-ve-b-U10-24 | vit-base-patch16-224-ve-b-U10-40 |
| vit-base-patch16-224-ve-U10-12 | vit-base-patch16-224-ve-U10-24 |
| vit-base-patch16-224-ve-U11-12 | vit-base-patch16-224-ve-U11-b-24 |
| vit-base-patch16-224-ve-U11-b-40 | vit-base-patch16-224-ve-U11-b-80 |
| vit-base-patch16-224-ve-U12-b-24 | vit-base-patch16-224-ve-U12-b-80 |
| vit-base-patch16-224-ve-U13-b-120 | vit-base-patch16-224-ve-U13-b-24 |
| vit-base-patch16-224-ve-U13-b-80 | vit-base-patch16-224-ve-U13b-80R |
| vit-base-patch16-224-ve-U13-80RX | vit-base-patch16-224-ve-U13b-80RX1 |
| vit-base-patch16-224-ve-U13b-80RX3 | vit-base-patch16-224-ve-U13b-R |
| vit-base-patch16-224-ve-U14-b-24 | vit-base-patch16-224-ve-U15-b-80 |
| vit-base-patch16-224-ve-U16-b-80 | vit-base-patch16-224-ve-Ub |
| vit-base-patch16-224-vit | vit-base-patch16-224-vit-base-patch16-224-vit-base-patch16-224-dogORnot |
| vit-base-pets | vit-base-PICAI |
| vit-base-seed-1e-4 | vit-base-seed-3e-4 |
| vit-base-travel-document-classification | vit-base-v1-eval-epoch-maxgrad-decay-cosine |
| vit-beans-classifier | vit-beta1-0.85 |
| vit-beta1-0.88 | vit-beta1-0.95 |
| vit-beta2-0.99 | vit-beta2-0.995 |
| vit-beta2-0.9995 | vit-bird |

| | |
|---|---|
| ViT_bloodmnist | ViT_bloodmnist_std_0 |
| ViT_bloodmnist_std_15 | ViT_bloodmnist_std_30 |
| ViT_bloodmnist_std_45 | ViT_bloodmnist_std_60 |
| ViT_breastmnist | ViT_breastmnist_std_0 |
| ViT_breastmnist_std_15 | ViT_breastmnist_std_30 |
| ViT_breastmnist_std_45 | ViT_breastmnist_std_60 |
| VIT-cats-vs-dogs | vit-cifar10-fine-tuned |
| vit-class-weight | vit-cxr4 |
| vit-demo | ViT_dog_food |
| vit-dropout-0.2 | vit-dropout-0.3 |
| vit-dropout-0.4 | vit-dropout-0.5 |
| vit-ds-processed | vit-emotion-model |
| vit-epsilon-1e-7 | vit-epsilon-1e-9 |
| vit-epsilon-5e-9 | vit-face-project-piyush |
| vit-fine-tune-classification-cats-vs-dogs | vit-finetuned-1 |
| vit-food-classification-chrisis2 | vit-geometric-shapes-base |
| vit-google-model-30-classes | vit_google_vehicle_classification_model |
| vit-historical-page | vit_Liveness_detection_v1.0 |
| vit-lr-0.0001 | vit-lr-0.001 |
| vit-lr-0.01 | vit-lr-cosine-restarts |
| vit-lr-cosine-warm-restarts | vit-lr-cosine-warmup |
| vit-lr-exponential | vit-lr-inverse-sqrt |
| vit-lr-linear | vit-lr-poly |
| vit-lr-reduce-plateau | vit-lr-step |
| vit-mae-base-finetuned-eurosat | vit-molecul |
| vit-ori-dataset-exp | vit-plant-classification |
| vit-plantnet300k | vit-plants |
| vit-real-fake-classification-v1 | vit-real-fake-classification-v2 |
| vit-real-fake-classification-v3 | vit-real-fake-classification-v4 |
| vit-skin-demo-v1 | vit-skin-demo-v2 |
| vit-skin-demo-v3 | vit-skin-demo-v4 |
| vit-skin-demo-v5 | vit-spam |
| vit-sports-cls | vit-transfer-learning |
| vit_transformer_eye_disease | vit_tumor_classifier |
| vit-vit | vit-vit-base-patch16-224-finetuned-chest-xray |
| vit-weight-decay-1e-2 | vit-weight-decay-1e-3 |
| vit-weight-decay-1e-4 | vit-weight-decay-1e-5 |
| wmc_v2_vit_base_wm811k_cls_contra_learning_0916 | wmc_v2_vit_base_wm811k_cls_contra_learning_0916_9cls |
| wmc-wmk811-v0-vit-special_map_det_0917 | WS800_ViT_42820348 |
| WS800_ViT_42895082 | xraynewww |
| yet-another-amber-mines | zdravJEM_CV_BERT |

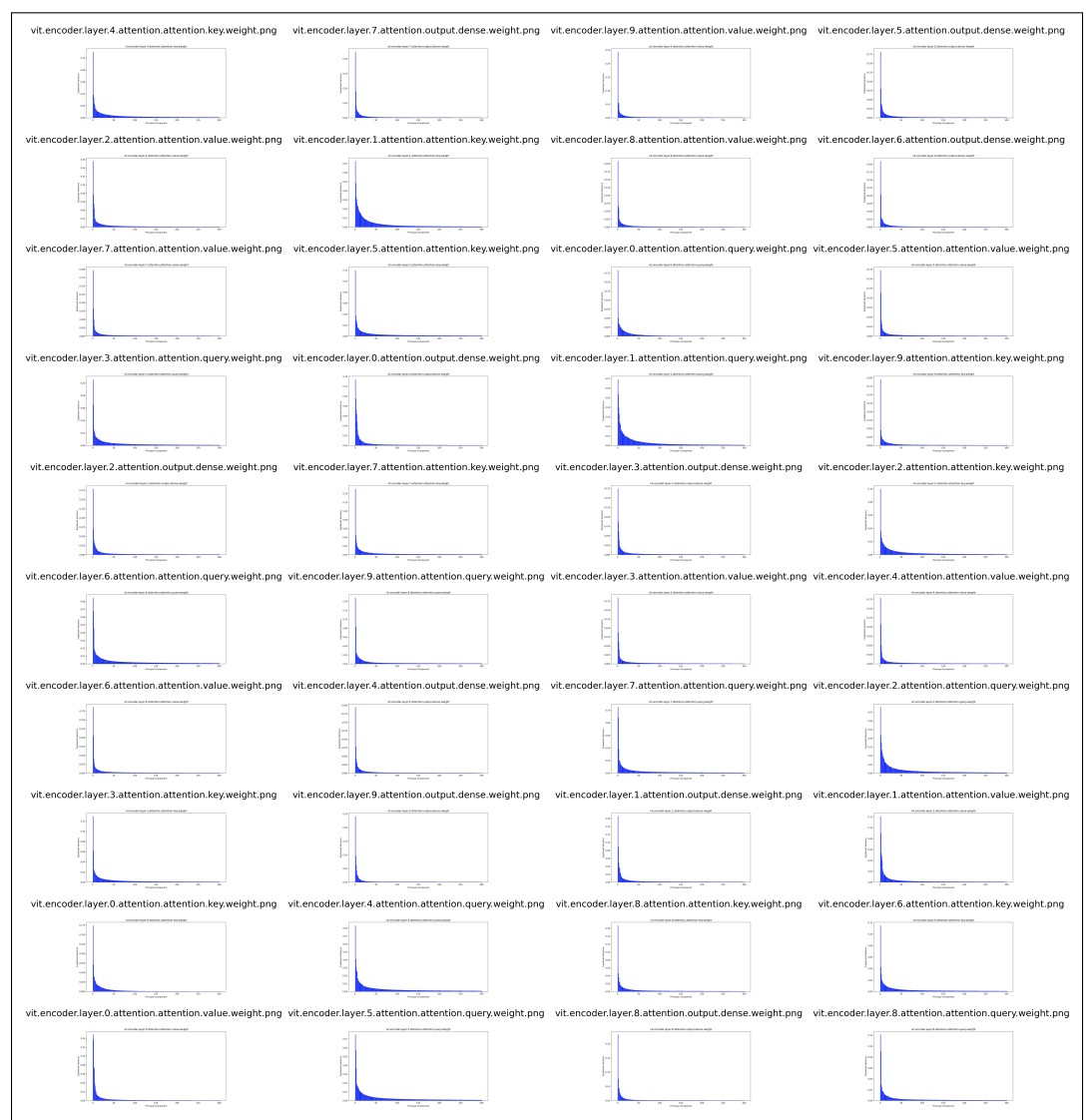

Figure 11: Layerwise Eigenvalue Plots of 500 ViT models.

**Universal LLaMA3-8B Experiment Details**  To further stress-test our universal subspace hypothesis on classical weight matrices, we extract a shared subspace from approximately 50 finetuned LLaMA3 models, each with 8 billion parameters. These models were obtained from publicly available repositories on HuggingFace. Due to their scale, we do not apply any model selection or filtering, and instead include the entire available set.

As shown in Figure 12, which presents the aggregated scree plot across all layers and all 50 models, the principal variance is concentrated in the top few components—consistent with the emergence of a low-rank universal subspace. For reference, the plot displays only the top 300 components, which represent a small fraction of the full rank, highlighting the inherently low-dimensional structure.

The models included in this analysis span a diverse range of domains, including medical applications, multilingual dialogue systems, and general-purpose assistants, as listed in Table 10. To the best of our knowledge, this is the first work to demonstrate that such a large and heterogeneous collection of high-capacity language models can be jointly represented within a single low-rank subspace.

The layerwise spectral analysis, shown in Figure 13, corroborates this finding: across all layers, the majority of eigenvalues fall below a threshold of $< 0.001$, indicating that most directions in

parameter space contribute negligibly to variation across models. The plots are cropped to show only the leading components due to the large number of total dimensions. We recommend zooming in for clearer visualization.

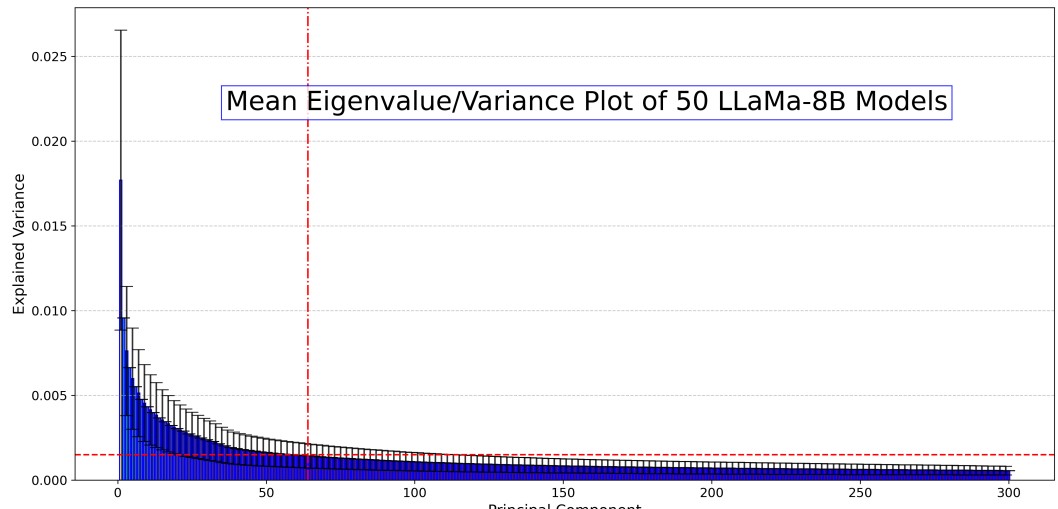

Figure 12: Spectral analysis of 50 LLaMA-3-8B model: Aggregated eigenvalue (scree) plot across 50 LlaMa-8B models and all layers. The plot demonstrates that the majority of the variance is consistently captured by few top principal directions, indicating the presence of a shared low-dimensional universal subspace.

Table 10: Models from HuggingFace used for the Universal LlaMa3-8B subspace extraction

| Meta-Llama-3-8B-Instruct-Jailbroken | Llama-3-13B-Instruct | large_crafting_sft_success | suzume-llama-3-8B-multilingual |
|---|---|---|---|
| summary-llama3-8b-f16-full | Llama-3-13B-Instruct-v0.1 | Llama-3-8B-ProLong-64k-Base | LLaMAntino-3-ANITA-8B-Inst-DPO-ITA |
| ai-medical-model-32bit | filtered_crafting_train_data_shorter_length | Llama-3-portuguese-Tom-cat-8b-instruct | Llama-3-MAAL-8B-Instruct-v0.1 |
| Human-Like-LLama3-8B-Instruct | LLaMA-3-8B-Instruct-TR-DPO | CabraLlama3-8b | chartgpt-llama3 |
| KoLlama-3-8B-Instruct | honeypot-llama3-8B | Llama-SEA-LION-v2-8B | TR |
| Llama3-8B-Instruct-Turkish-Finetuned | Llama-3-15B-Instruct-zeroed | Llama-3-8B-Instruct-TAR-Bio-v2 | Bio-Medical-Llama-3-8B |
| filtered_construction_train_data | shisa-v1-llama3-8b | REFUEL-Llama-3-Armo-iter_1 | llama3-instrucTrans-enko-8b |
| Llama-3-8B-Instruct-Ja | llama3-passthrough-chat | RoLlama3-8b-Instruct | Lloro-SQL |
| Summary_L3_1000steps_1e7rate_SFT2 | CyberSentinel | Meta-Llama-3-8B-Instruct-function-calling-json-mode | MARS |
| Llama-3-8B-Instruct-Finance-RAG | LLaMA3-Instruct-8B-FR-Spec | Llama-3-8B-Japanese-Instruct | Llama3-8B-Chinese-Chat |
| llama-3-chinese-8b-instruct-v2 | Athene-RM-8B | Llama-3-OffsetBias-RM-8B | large_cooking_sft_success |
| suzume-llama-3-8B-japanese | llama-3-chinese-8b-instruct-v3 | Waktaverse-Llama-3-KO-8B-Instruct | llama-3-8b-gpt-4o-ru1.0 |
| Llama-3-Aplite-Instruct-4x8B-MoE | Llama-3-8B-Instruct-DPO-v0.3 | | |

**Universal Flan-T5 Experiment Details** We collected Flan-T5 models fine-tuned on individual datasets from the GLUE (Wang et al., 2019) benchmark. We extract the joint subspace from these models and trends similar to those observed above are seen. This shows that across diverse datasets and tasks a low-rank subspace emerges.

Table 11: Finetuned Flan-T5 Models from HuggingFace used for the Universal Flan-T5 subspace extraction

| tanganke/flan-t5-base_glue-cola | tanganke/flan-t5-base_glue-mnli |
|---|---|
| tanganke/flan-t5-base_glue-mrpc | tanganke/flan-t5-base_glue-qnli |
| tanganke/flan-t5-base_glue-rte | tanganke/flan-t5-base_glue-qqp |
| tanganke/flan-t5-base_glue-sst2 | tanganke/flan-t5-base_glue-stsb |

### B.3 ABLATING NUMBER OF MODELS AND SUBSPACE EFFECTIVENESS

Although this is implicitly addressed through our large-scale experiments (500 ViTs, 500 Mistral-7B and 300 Stable Diffusion LoRAs, 50 LLaMA3-8B, 177 GPT-2s, Flan-T5, and ResNet50 models) in all Figures and Tables, which demonstrate consistent behavior at different scales. Theorem 2.5 provides insights on the saturation dynamics where we see that the rate of convergence of the shared subspace to the true subspace is in the order $O(1/T)$, where T is the number of tasks, indicating increasingly effective coverage as T increases. In practice, the minimum number of models per

Table 12: Lots of LoRAs (Mistral-7B) OOD evaluation per increasing number of models used to extract Universal Subspace

| Method | Model Number | Rouge-L Score |
|---|---|---|
| Normal Model | - | 73.7 |
| Universal model | 50 | 55.8 |
| Universal model | 150 | 66.1 |
| Universal model | 250 | 71.9 |
| Universal model | 450 | 72.3 |

architecture needed to achieve saturation point depends on the quality of the trained models, the diversity of data they have been trained on, and on the architecture itself. Ablating these would require access to all the data for all the models, and very careful training on every training for each data, and then running permutation with all possible combinations of models. All of this is out of reach for most researchers simply due to time, data and compute constraints. We, however, do provide an initial ablation here. For LoRA models shown in Table 7, we choose 9 random (OOD) tasks (39 ,190 ,280 ,290 ,391 ,442 ,1342 ,1391 ,1598) and extract the Universal Subspace from rest of the the tasks, sampled randomly for increasing number of models. The coefficients for OOD tasks are analytically reconstructed to effectively evaluate the universal subspace created from varying number of models. Table 12 shows that the adequate principal components are quickly extracted, and increasing the number of models has diminishing returns.

## C  FINDING UNIVERSAL SUBSPACES AND APPLYING THEM TO FUTURE TASKS

In this section, we present two tasks, GLUE (Wang et al., 2019) and Image Classification. For each experiment, the joint subspace is created using all other models in subset. For Image Classification, we use $k = 4$ and train only 8 epochs using learning rate of 1e-4. Importantly, only the coefficients are trained for the experiment. It is important to note that our shared subspace model performs quite well despite using very few (4-5) models to extract the subspace. For GLUE, we use 16-32 components for our subspace, with learning rate of 4e-4, batch size of 64, and 30-80 epochs for each task. In addition, it is likely that our model might perform similarly or better if trained longer or with optimized hyperparameters.

**Compute Resources**   We conduct all our experiments using a single A5000 GPU, and a CPU with 8 workers. For the universal subspace extraction, all calculation can be done on the CPU. However, GPU would increase the speed of calculation as the layerwise subspace extraction can be parallelized.

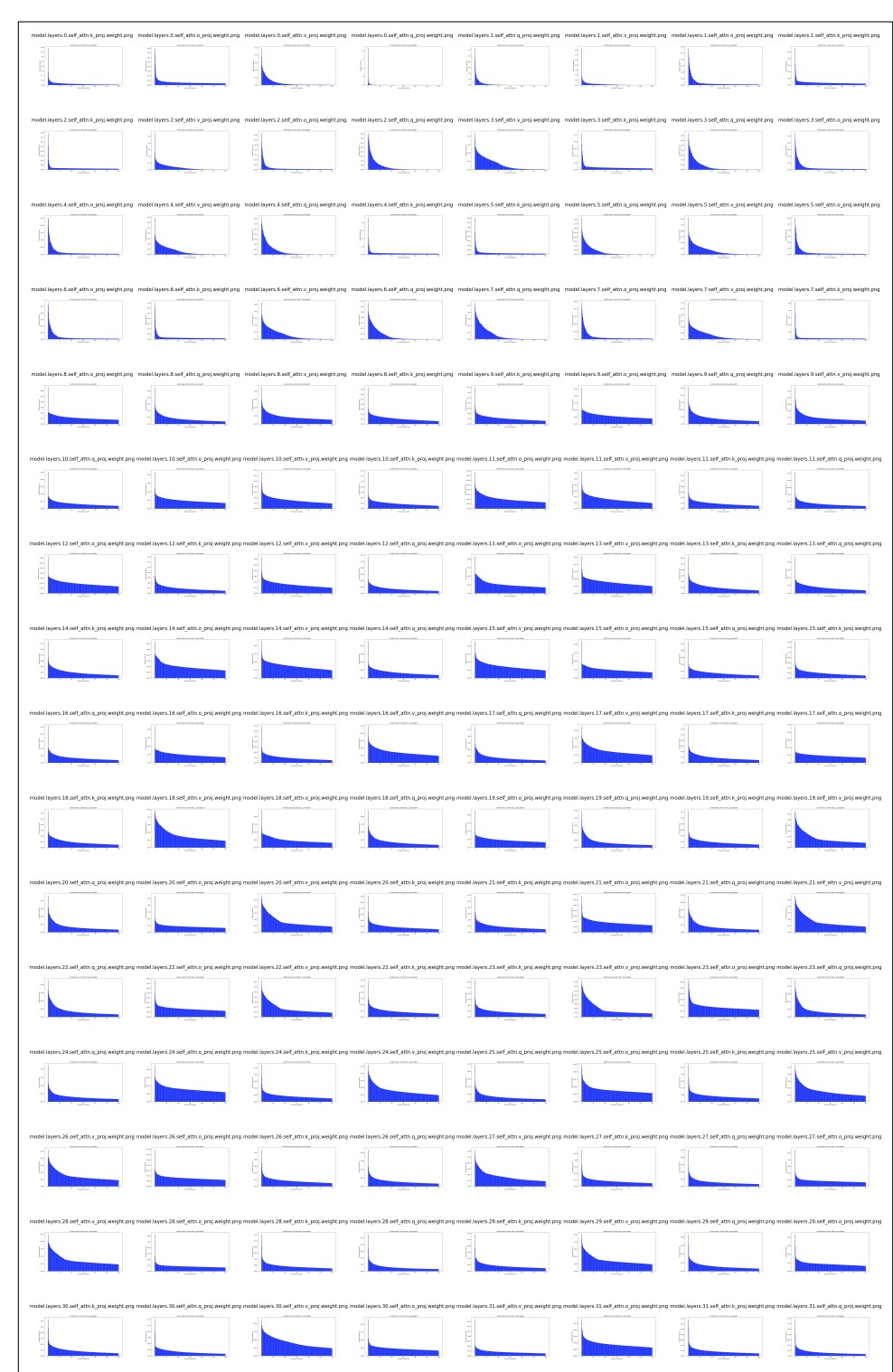

Figure 13: Layerwise Scree Plots for 50 LLaMA-3-8B Models. For enhanced clarity, each subplot presents a truncated view of the total possible principal directions. These plots consistently demonstrate that the dominant information, as represented by explained variance, resides within a small number of leading principal directions for all models. Components beyond this initial set are characterized by eigenvalues approaching zero, signifying their redundancy for the universal subspace.

## D DISCUSSION AND BROADER IMPACT

Our findings suggest that deep neural networks trained across diverse tasks and modalities systematically converge to shared, low-dimensional subspaces within their parameter space. The existence of such universal subspaces challenges conventional assumptions about the independence and diversity of model and task-specific finetuning trajectories. Instead, it highlights a powerful regularity in the way deep models encode task-specific knowledge - one that can be exploited for significantly improved training and deployment efficiency. By leveraging these subspaces, we demonstrate that models can be adapted to new tasks by learning only a small number of coefficients, rather than retraining or storing full sets of weights. This facilitates more robust multi-task learning, model merging, and scalable fine-tuning, with theoretical guarantees and empirical validation across multiple architectures.

The broader societal impact of this work is substantial. Our approach enables large-scale models to be reused and extended with dramatically reduced computational overhead, addressing both the financial and environmental costs associated with training and deploying deep learning systems. This contributes directly to the goals of sustainable and accessible AI. By lowering the hardware and energy requirements for adaptation and inference, we empower under-resourced researchers, institutions, and communities to build upon state-of-the-art models without needing extensive compute infrastructure. Furthermore, by supporting modular model design and data-free model merging, our work lays the foundation for more interpretable, maintainable, and equitable AI systems.

