# OpenReview forum: "Do Neural Networks Learn Similar Subspaces? An Empirical Exploration of Joint Parametric Subspaces in Deep Neural Networks"
_ICLR.cc/2026/Conference — Submitted to ICLR 2026_

### Official Review · Reviewer_c4a8 · 2025-10-30

**Soundness:** 3
**Presentation:** 4
**Contribution:** 3
**Rating:** 6
**Confidence:** 4

**Summary:**

This submission presents empirical evidence that the training of neural networks generates low-dimensional parametric subspaces, as identified through spectral decomposition, which the authors claim to be universal. These subspaces are able to capture the majority of the variance of a collection of neural network models drawn from the same architecture into a few principal directions.

Potential downstream applications of these findings are the compression of neural networks into a smaller subset of coefficients of the found principal directions, allowing to significantly save memory and compute.

Experimental results show evidence of these subspaces in ResNets training on multiple computer vision datasets,  LoRA adapters from Mistral-7B-Instruct-v0.2, and LoRA adapters from Stable Diffusion SDXL models. Finally, the last set of experiments was run on ViT, GPT-2, LLaMa-8B, and Flan T5 models. In all experiments, the reported subspaces emerge given the spectral decomposition method introduced by the authors.

I think this is a fascinating paper and very much appreciate this work as it might indeed change the way we look at neural networks and the structures they represent after training. I would have some questions, which I outline in more detail below.

**Strengths:**

- **(S1)**: Given that neural network models are becoming larger and more models are being made publicly available, finding such parametric subspaces is not only timely but can also provide novel directions for model compression, potentially leading to less compute needed in the future.

- **(S2)**: I appreciate the paper's motivation and theoretical formulations. It provides an interesting connection to the empirical results.

- **(S3)**: This work provides a large-scale set of experiments investigating if subspaces form in collections of very different neural network models composed of different architectures, datasets being trained with, losses and tasks being optimised for, and modalities. This is truly helping to understand empirically how universal such subspaces are. I also appreciate the details and additional results listed in the appendix.

- **(S4)**: This paper is easy to read and follow, given its systematic build-up.

**Weaknesses:**

- **(W1)**: One of the stronger limitations has already been outlined by the authors. The proposed spectral decomposition can only be done on models from the same neural network architecture. It would be great if one could bridge the found subspaces between architectures to be truly universal. Therefore, I would also be careful in using the term "universal" or, said the other way around, properly defining what the term means in the context of this work.

**Questions:**

I would have some questions:

- **(Q1)**: How does this work compare to WeightWatcher [1] using spectral decomposition to find similar structures in NN weight matrices, and follow-up work learning lower-dimension manifolds of populations of neural networks [2]. The submitted work appears to be quite similar to [1] and has a connection to [2], but is missing in the comparison of methods paragraph (line 71-81).

[1] Martin & Mahoney, Predicting trends in the quality of state-of-the-art neural networks without access to training or testing data, Nature Communications, 2021
[2] Schuerholt et al, Towards Scalable and Versatile Weight Space Learning, ICML 2024


- **(Q2)**: How do you perform classification in subspace? Results from Fig. 2 (a), Table 2, and Table 4 show accuracy results of the corresponding models being in their universal subspaces. How does inference work in these subspaces? Or do you project the original model into the found subspace and back by reconstructing it only from the first major (principal) directions? I found it briefly mentioned in the appendix. Can you provide more details on this?

- **(Q3)**: I have a similar question for the "model compression" example of 500 ViT models (line 381-388). Can you outline why you understand this as "merging"? Could you provide more information about this? Also, what exactly do you project into the found subspace? You mention that the task variable first and last layers are ignored. Can you outline what you mean by this?

- **(Q4)**: Same question for Section 3.2.2. The idea is to learn only their task-specific coefficients for the new task. Can you provide details on how you apply gradient descent in the subspace to learn the task-specific coefficients?

- **(Q5)**: Last question, have you compared the coefficient of the original models to their sparsified counterparts (for example, for magnitude pruning or variational dropout sparsification). Basically, my question is, do you think the subspace would change significantly or would the coefficients of the sparsified model change as compared to the original models?

As I already said, I think this is fascinating and I would really appreciate more details about the downstream tasks and how they are performed. I would be happy to increase my rating once I fully understand the interaction between the weight space and universal subspace.

**Details Of Ethics Concerns:**

-

---

> ### Author Response · Authors · 2025-11-28
> **Response (Part 1)**
>
> We thank the reviewer for their insightful comments and queries.
>
> W1:**” .. bridge the found subspaces between architectures to be truly universal.”**
>
> Thank you for this suggestion. We agree that, in the current form, our notion of “universality” is fundamentally architecture‑specific. In the revised draft we have made this more explicit by consistently describing our findings as universal subspaces within shared architectures - including in the abstract, introduction, and discussion. We have updated the wording in several places to use terms such as “architecture‑specific universal subspace” or “shared universal subspace within an architecture” to avoid overclaiming, and we will adopt this terminology systematically in the final version. As you suggest, developing mathematical tools for genuinely cross‑architectural analysis is a natural and important direction for future work, and as you noted, we explicitly list this as a key open problem in the limitations section.
>
> ---
> **Q1: "How does this compare to WeightWatcher and Schuerholt et al.?"**
>
> Thank you for citing these highly relevant works. We have added citations for both and will include a detailed discussion in the related work section (Section A1).
>
> Both works, while not explicitly analyzing or hypothesizing shared low-rank subspaces of neural networks, offer valuable insights that are complementary to our findings.
>
> The first is the theory of Heavy-Tailed Self-Regularization (HTSR) by Martin et al. This framework leverages Random Matrix Theory (RMT) to analyze the distribution of eigenvalues, proposing that the correlation structure of learned weights follows a heavy-tailed power law, which serves as a metric for generalization capability. While this work focuses on the eigenvalue distribution analysis of individual layers or an average across one model, our work investigates the properties of a joint, shared subspace underlying all models specific to a given architecture. Both works shed light on the black-box nature of neural network learning and propose complementary theories. Specifically, HTSR can provide indirect information about the universal subspace - measuring $\alpha$ (HTSR) is effectively measuring the "peakedness" or "strength" of the Universal Subspace without needing to calculate the subspace explicitly.
>
> Schuerholt et al. introduce SANE, a parameter generation method focused on hyper-representation learning. Unlike previous works that required all model weights to be used as a combined input, SANE tokenizes individual weights, which are then input sequentially to a large autoencoder model. This approach allows them to scale their method to larger networks. However, the scope of their work is somewhat limited compared to ours, focusing predominantly on convolutional image classification networks. They utilize metrics proposed in the aforementioned work (and their earlier publications) for evaluation. Although not directly related, we hypothesize that the Universal Subspace hypothesis could allow for a rephrasing of SANE or similar methods by only requiring the learning and generation of coefficients or eigenvalues for the extracted joint, layer-wise subspace.
>
> ---
> **Q2: "How do you perform classification in subspace?"**
>
> We use the shared subspace in two closely related ways.
>
> 1. Projection and reconstruction of existing / unseen models.
>
> For a trained model with layer weights $\(W\)$, we first learn a layer-wise basis $\(U\)$ from a collection of models, then project and reconstruct via $\hat{W} = U U^\top W,$ equivalently solving $\(\min_{C} \lVert W - U C \rVert^2\)$ using the SVD/PCA of the stacked weights (no gradient training is needed).
>
> We then run inference with $\(\hat{W}\)$ in place of $\(W\)$.
>
> This is the procedure used in Fig.2(a), Table 2 (model merging (newly added)), and Table 3.
>
> In the OOD setting of Table 3, the OOD models are **not** used to estimate $\(U\)$; they are only projected afterwards. The fact that their accuracy remains high after projection indicates that the learned subspace genuinely generalizes across tasks within the architecture.
>
> 2. Learning task-specific coefficients with frozen bases.
>
> In Sec.3.3.1 and Table 5, we instead **freeze** the shared bases $\(U\)$ and learn only lightweight coefficients $\(C\)$ for a new task, parameterizing each layer as $\(W_{\text{task}} = U C\)$.
>
> We initialize $\(C\)$ randomly and update it by standard gradient descent on the task loss, while $\(U\)$ stays fixed; this drastically reduces the number of trainable parameters and leads to faster, more stable convergence, as reflected by the speedups in Table 4 and the parameter counts in Table 5.
>
> Both regimes rely only on standard spectral decomposition and gradient-based optimization, and we plan to release a small library that creates Universal variants of different architectures and efficient finetuning and inference using our method.

---

> ### Author Response · Authors · 2025-11-28
> **Response (Part 2)**
>
> **Q3: I have a similar question for the "model compression" example of 500 ViT models (line 381-388). Can you outline why you understand this as "merging"? Could you provide more information about this? Also, what exactly do you project into the found subspace? You mention that the task variable first and last layers are ignored. Can you outline what you mean by this?**
>
> Thank you for asking for clarification; we agree that “merging” can be interpreted in multiple ways. In Section 3.3 and Figure 6, what we do for the 500 ViT models is best described as basis sharing and compression, rather than task‑arithmetic‑style weight averaging. We refer to this as merging because Conceptually, hundreds of independent ViTs become one set of shared bases {stored once), plus a set of small coefficient tensors for each of the original models. This yields up to 100x reduction in total memory while preserving per‑model performance (Table 3 and discussion around lines 432-485), which is why we spoke of “merging into a single universal representation.” We are happy to rephrase this more precisely as “compression into a shared subspace representation” to avoid confusion.
>
> We do, however, have added comparison with model merging methods in new Table 2.
>
> For each ViT, we consider all internal layers except the first (patch embedding) and last (classification head), because those differ in input resolution and number of classes across tasks. For each such layer
> $l$, we flatten the weight matrices from all 500 models (for mode-1 HOSVD), stack them, and perform spectral decomposition to obtain a shared basis. Each model’s layer is then represented via its coefficients in this basis.
>
> We exclude first/last layers as these layers are inherently task‑specific (different input sizes and label spaces). Including them would conflate architectural and task variation and make extracting a shared subspace ill‑posed. Line 429 and Appendix B (lines 947-950) provides explicit justification: "We do this universal subspace analysis on all weight parameters in every neural network layer except the first and last neural network layer. This is because these layers may differ across models due to differences in input shapes and types, loss functions, and the tasks being trained."
>
> ---
> **Q4: "How do you apply gradient descent to learn task-specific coefficients in subspace?"**
>
> In Section 3.3.1 and Tables 4-5, the optimization is standard SGD/Adam, but applied only to the coefficients in the subspace.
>
> - For each layer $\ell$, we fix the basis $U_\ell$ learned from a set of source models.
> - For a new task, we initialize trainable coefficients $C^{(\ell)}$ (vector or small matrix, depending on layer shape).
> - The forward pass computes reconstructed weights $W^{(\ell)} = U_\ell C^{(\ell)}$ (or the appropriate reshaping variant) and then proceeds exactly as in a normal network.
> - Backpropagation computes gradients $\nabla_{C^{(\ell)}} \mathcal{L}$ via the chain rule through $W^{(\ell)} = U_\ell C^{(\ell)}$; since $U_\ell$ is fixed, only $C^{(\ell)}$ is updated.
>
> This yields:
>
> - much smaller parameter count (e.g., 10K vs. 86M in Table 5),
> - smoother optimization (learning only linear coefficients in a well‑conditioned subspace), and
> - competitive performance on GLUE and image classification benchmarks.
>
> We will add a short algorithmic box summarizing this “coefficient‑only” training procedure.
>
> ---
> **Q5: "have you compared the coefficient of the original models to their sparsified counterparts (for example, for magnitude pruning or variational dropout sparsification). Basically, my question is, do you think the subspace would change significantly or would the coefficients of the sparsified model change as compared to the original models?"**
>
> We have not yet performed a large, systematic experimental study comparing coefficients before and after explicit sparsification, so we prefer not to over‑claim here. However, we conducted a preliminary experiment with a ViT‑B/16 trained on CIFAR‑100 (initialized from ImageNet‑1k) and sparsified to 98\% using magnitude pruning, then reconstructed by projecting this model into our universal ViT subspace.
>
> |                  Model                  | CIFAR100 |
> |:---------------------------------------:|:--------:|
> | Vit-B/16 + Magnitude Pruning (98%)      |   57.56  |
> | Magnitude Pruning (98%) -> UniversalViT |   57.45  |
>
> We observe essentially no statistically significant drop in performance after projection, which suggests that, at least in this setting, sparsification at this level preserves the dominant principal directions captured by the universal subspace. This is consistent with the intuition that, for moderate pruning, changes are mainly reflected in the coefficients rather than in the underlying basis, whereas at very high sparsification one should expect to lose at least some of the shared basis structure.

---

### Official Review · Reviewer_g5Q1 · 2025-11-01

**Soundness:** 2
**Presentation:** 2
**Contribution:** 2
**Rating:** 4
**Confidence:** 2

**Summary:**

This paper suggests that weights of the most of the available pretrained deep learning model have a joint universal subspace. The authors claim that thanks to this fact, huge memory/computation gains can be achieved. In the paper, there are emprical results that show most of the explained variance belong to few directions, which is common across models architectures and tasks. Authors also provide some mathematical proofs to strengthen their claims.

**Strengths:**

One good thing regarding this paper is that they conducted lots of experiments with variety of models and tasks. Also, they made a theoretical analysis.

**Weaknesses:**

I think the language and explanation is weak and some parts are confusing regardless of the content. When it comes to the content, first of all, the claim that the weights of pretrained models are having low rank structure is not new. There are several works that claim and made analysis on such low rankness, even if your claim is more broader which is universal low rank structure. Moreover, in Table 4, it is the universal model is compared with the full training. Which is I believe not fair. If you have pretrained weights in universal model, it would be fair to compare it with another finetuning on pretrained model, for example just finetuning last layer etc. Such a table gives impression like you trained a 86M model from scratch  and you 10K model from scratch and your model gives almost the same performance, however the universal ViT has a strong pre-knowledge.

**Questions:**

1 - To be honest, I did not quite understand the core point of the project. How are you getting the Figure 1 ? So far, it seems that you plot the explained variances for each principan components, and provide a bar plot. However, this does not mean that these weight matrices are having a common subspace. The directions of the first eigenvalues can be different. Probably, I miss an important point here.

---

> ### Author Response · Authors · 2025-11-26
> **Response (Part 1)**
>
> Concern 1. **I think the language and explanation is weak and some parts are confusing regardless of the content.**
>
> The work presents a novel and fundamental concept alongside numerous results, necessitating a high degree of information density, which can make the paper dense to read.
>
> To address this, we have improved the language, explanations, and nearly all main text figures (e.g., the updated Figure 1). We have also added highlighted sections in the main text to provide brief, less dense summaries for reviewers who prefer them instead.
>
> Please specify parts which you find weak or confusing and we will be happy to provide an explanation, if not already present in the main text.
>
> ---
> C2: **Low-rank structure is not novel; prior work on spectral decomposition and rank collapse exists..**
>
> You are correct that low-rank structure has been studied in a few prior works (see discussion in Section A1 and the second paragraph of our Introduction). However, our contribution is distinct: we demonstrate a **shared** or **aligned** low-rank structure across diverse tasks. This is the property of **universality**, as highlighted in the very first line of the introduction (and also mentioned in the abstract, analysis, discussion, and appendix sections).
>
> “We show that backpropagated neural networks trained on a variety of datasets - which could be disjoint and unrelated - diverse hyper-parameter settings, initializations and regularization methods, often learn an architecture-specific, layer-wise similar, low-rank joint subspaces (we refer to this as the Universal Subspace). We provide the first large-scale empirical analysis - across a diverse set of models - that neural networks tend to converge to these joint subspaces, largely independent of their initialization or the specific data used for training.”
>
> This enables novel applications, such as compressing 500 Vision Transformers (ViTs) into a single subspace model, achieving parameter-efficient adaptation (up to greater than 100x), and data-free model merging, etc.
>
> Furthermore, even among works studying the low-rank structures of backpropagated neural networks, to the best of our knowledge, no previous work presents the scale and diversity of experiments that we provide.
>
> If the reviewer has a specific concern about novelty, please clarify.
>
> ---
> C3: **"Table 4 comparison unfair - comparing against full training, not finetuning baselines"**
>
> The **Full Training** baseline is finetuned on the respective datasets and is initialized using an Imagenet pretrained ViT, following the normal convention for ViT finetuning. Allowing all parameters to be trained enables the best possible optimization.
>
> The **Universal ViT**, on the other hand, has **frozen principal components** which have been extracted from models that have **not seen** the datasets mentioned in the Table. Only lightweight coefficients for these components are then finetuned.
>
> The models in Section 3.3.1 are initialized using the joint subspace extracted from pretrained models (excluding the data/model that it is subsequently trained on). The PCs/basis are then **frozen**, and coefficients are tuned on data that is **Out-of-Distribution (OOD)** for the PCs, before being tested on unseen data (**explained in line 452**). If our low-rank universal subspace hypothesis were incorrect, the model would fail to learn, as all basis vectors are frozen, and a new set of basis vectors would be required for the new data. This is particularly relevant since the Universal subspace was extracted from a small number of models. However, our experiments show that even this Universal subspace is adequate for good results, further validating the **Universal subspace property** and showcasing an application of **efficient learning**.
>
> Regarding the comment, "universal ViT has a strong pre-knowledge" - It is unclear what the reviewer means by this. Please clarify. If the reviewer is agreeing that the Universal ViT has a universal subspace that can be finetuned efficiently for any future tasks, then we agree; this is precisely what Section 3.3.1 demonstrates.
>
> ---

---

> ### Author Response · Authors · 2025-11-26
> **Response (Part 2)**
>
> Question 1a. **“To be honest, I did not quite understand the core point of the project.”**
>
> Our abstract, Figure 1, and the first line of the introduction provide a concise overview and objective of our work.
>
> “We show that backpropagated neural networks trained on a variety of datasets - which could be disjoint and unrelated - diverse hyper-parameter settings, initializations and regularization methods, often learn an architecture-specific, layer-wise similar, low-rank joint subspaces (we refer to this as the Universal Subspace)”
>
> “We provide the first large-scale empirical analysis - across a diverse set of models - that neural networks tend to converge to these joint subspaces, largely independent of their initialization or the specific data used for training.”
>
> Our work is simply an analysis of the fundamental properties of backpropagated neural networks. However, as discussed in Section 4, Figure 7, and Appendix D, the implications of this observed property are significant.
>
> ---
> Q1b: **"Figure 1 shows explained variance but doesn't prove shared subspace - directions could be different"**
>
> Please refer to the updated Figure 1 for more details, and note the highlighted points throughout the main text. Figure 1 plots the explained variance with respect to a specific principal component for a specific layer (the graphs for all layers are summarized into one for brevity).
>
> This suggests that all models share the same subspace, as we subsequently show in our paper: discarding the remaining, unneeded subspace has no significant change in the performance of individual models. Since this 'universal subspace' is shared, we can replace all the models with a single universal one.

---

### Official Review · Reviewer_rMn9 · 2025-11-02

**Soundness:** 1
**Presentation:** 1
**Contribution:** 2
**Rating:** 2
**Confidence:** 2

**Summary:**

The paper claims to show empirical evidence for the hypothesis that the parameters of DNNs with the same architecture trained on different datasets and/or trained with different hyperparameters and initializations, lie in a low-dimensional linear subspace. It does so by doing a form of singular-value decomposition on the covariance matrices of the concatenated weights of the networks. Estimation is done by means of a "sample" of trained DNNs, already publicly available (on Huggingface). The main experimental evidence is two-fold:
* it is shown that a large part of the explained variance in the parametric spaces is located in a limited number of principal component directions.
* it is shown that after projection onto this low-dimensional subspace of a limited number of principal components, the performance of the DNNs remains (largely) intact.

**Strengths:**

The focus on (effective) linear subspaces of the parameter space of DNNs is an interesting direction of research. This seems to be orthogonal to the study of (effective) linear subspaces of embedding spaces, which has been studied extensively in recent years.

**Weaknesses:**

* the main weakness of the paper is its unclarity, which makes it extremely hard to assess what the paper claims exactly and what precisely is done in the experiments. This weakness makes it impossible for me to give a positive assessment of the paper in its current form. Here are some examples of this unclarity:
    * the theoretical hypothesis the paper claims to introduce is nowhere stated in an explicit and (mathematically) precise manner.
    * the main empirical claims follow from a spectral decomposition that is outlined in Algorithm 1, but this is poorly written. For example, what does $I_n$ mean and what does $N$ mean? And how are the ranks $r_n$ chosen? And what is happening in line 4 of the algorithm?
    * the experimental setup is also rather opaque. For example, it is not made clear how exactly the multiple trained DNNs with the same architecture are being used. I am assuming they are used as a "sample" on whose weight space SVD is applied, but this is nowhere explained in a detailed manner.
    * also the experimental results are not clear. For example, what is the relation between figures 3b and 3a? What does the word "summarized" in the caption of figure 3b mean? Another example is Table 3, where it is unclear what "mode-2" and "mode-3" mean.
* a rather disturbing weakness is that the paper contains (at least) two non-existing references. The two cited papers by Guth et al. do not exist and the mentioned arXiv numbers refer to totally different papers. I kindly ask the authors to give an explanation for this and to also let the reviewers know whether the paper contains more citation errors.
* a minor weakness is that the presentation is sometimes sloppy. The text contains main typo's or grammatical mistakes. Sometimes the text refers to the Appendix, but it is not clear to which section in the Appendix. In Appendix A.2 Theorem A.3 is stated, which I believe is a copy of Theorem 2.5, but Theorem A.2 is never referred to anymore.

**Questions:**

* The Tables 3 and 4 you are working with "universal" DNNs, which seem to be some kind of projections onto the universal subspace of parameters. How does this work precisely? For example, how are the dimensions of the universal subspace determined? Is this done per layer of the DNN? And which model is projected onto this subspace?
* My other questions are contained in my overview of Weaknesses.

---

> ### Author Response · Authors · 2025-11-27
> **Response (Part 1)**
>
> Concern 1a: **”the main weakness of the paper is its unclarity, which makes it extremely hard to assess what the paper claims exactly and what precisely is done in the experiments”**
>
> We understand that the reviewer may have found the paper challenging to follow. Our work introduces a novel and fundamental concept alongside numerous results, which necessarily leads to a high density of information.
>
> To significantly enhance accessibility, we have thoroughly revised our figures, particularly Figure 1, which should now clarify several of the reviewer’s concerns.
>
> We have also deliberately highlighted key points throughout the text to present the work's dense information more concisely.
>
> For a focused objective of our work, please refer to the first few sentences of the introduction. As stated in the very first sentence:
> “We show that backpropagated neural networks trained on a variety of datasets - which could be disjoint and unrelated - diverse hyper-parameter settings, initializations and regularization methods, often learn an architecture-specific, layer-wise similar, low-rank joint subspaces …”
>
> ---
> C1b. **the theoretical hypothesis the paper claims to introduce is nowhere stated in an explicit and (mathematically) precise manner.**
>
> We are unclear which theoretical hypothesis the reviewer is referring to - can they please clarify. We provide theoretical analysis (assumptions 2.3, 2.4 and theorem 2.5) to understand the characteristics of the proposed shared subspace while our universal subspace hypothesis is noted in our abstract, first line of the introduction as well as primary contributions (in introduction). Our analysis-focused paper pertains to empirically demonstrating the existence of the lower dimensional shared subspace. There has been no such rigorous study towards analyzing the parametric (LoRA as well as classical) subspaces of various types of models from diverse modalities including large foundation models like Llama, Mistral, etc. We have highlighted this hypothesis. To this end, we kindly ask the reviewer to precisely elaborate their concern in this regard for us to better address it.
>
> ---
> C1c: **what does $N$ mean and what does mean? And how are the ranks chosen? And what is happening in line 4 of the algorithm**
>
> As shown in line 218, $I\_1\times…I\_N$ represent the dimensions of Tensor $\mathcal{X}$. We have further added $n\\in\[1,N\]$ for clarity. Ranks $r\_n$ are chosen based on the explained variance of the leading eigenvectors either through visual inspection or using a threshold (See discussion in Appendix B). Please find the updated algorithm in our revised draft where we further clarify the notations, operators we have used in it.
>
> ---
> C1d. **the experimental setup is also rather opaque. For example, it is not made clear how exactly the multiple trained DNNs with the same architecture are being used. I am assuming they are used as a "sample" on whose weight space SVD is applied, but this is nowhere explained in a detailed manner.**
>
> We regret that the reviewer finds our experimental setup opaque, but we assert that it is thoroughly explained (and is easily reproducible) in our paper. Our paper’s core objective - to demonstrate that neural networks learn low-rank joint subspaces - is introduced in the first paragraph (please refer to updated Figure 1 and highlighted text in Introduction for brief overview)..
> To verify the existence of these 'universal' joint subspaces, our experimental setup is designed around a diverse set of models and tasks, encompassing nine distinct experiments. This is briefly introduced in lines 86-92 (Introduction) and detailed extensively in Section 3 and Section 3.2. The paper's organization is outlined in lines 101-107 (Introduction). Furthermore, Section 2 (last paragraph) explains our methodology and explicitly refers to **Algorithm 1**.
>
> Regarding the specific example of applying SVD on the concatenated weight space, this is precisely what Algorithm 1 proposes for general, higher-order weight tensors (in which case we use HOSVD instead of SVD). We have now further refined Algorithm 1 for improved clarity.
>
> The experimental setup is further clarified across all subsections of Section 3 and in **Appendix Section B**, which is individually referenced for each experiment:
>
> LoRA experiments **(B.1)**: "For every layer, the rank vectors of all available LoRA matrices are extracted and concatenated into a single matrix. This matrix is then normalized by subtracting the feature-wise mean from each vector, after which principal directions are extracted …. ". Please refer to the section for further details.
>
> ViT experiments **(B.2)**: "We collect 500 pretrained ViT models from HuggingFace... Figure 9 shows the summarized scree plot... we randomly choose 4 IID and 4 OOD models... and reconstruct these model weights by projecting them into our 16 component universal subspace."
>
> Appendix B provides detailed specifications for all experimental settings.
>
> ---

---

> ### Author Response · Authors · 2025-11-27
> **Response (Part 2)**
>
> C1e: **"Experimental results unclear - what is the relation between Figures 3a and 3b? What does 'summarized' mean?"**
>
> We have provided concise (due to space limits) explanations and the experimental setup for every experiment, with references to the Appendix where more details are needed.
> To address potential confusion regarding the figures:
> Figure 3a: "Eigenvalue/Variance plot for Orthogonal Spectral Components for 500 unique LoRAs of different layers of Mistral-7B model" clearly shows the per-layer decomposition.
> Figure 3b: "Summarized eigenvalue plot of all LoRAs corresponding to all 31 layers of all 500 Mistral-7B models" aggregates the results across all layers via averaging to show the overall spectral structure.
> Specifically, Figure 3b is the summarized/averaged/mean representation of all the graphs shown in Figure 3a (including those in Section B1). To further aid the reviewer, we explicitly define the term ‘summarized’ in the caption of Figure 2b.
>
> Regarding terminology, the modes in (now) Table 4 can be referenced from line 4 in Algorithm 1. While ‘mode’ is common terminology in HOSVD to refer to the order of the tensor, to simplify this for the reviewer, we have changed ‘mode’ to ‘order’ throughout, though either can be used interchangeably.
>
> ---
> C2: **: "Citation errors"**
>
> Both Guth et al. references have been fully corrected. The initial error was likely caused by an outdated or corrupted bibliography file overriding the correct references. We have now meticulously confirmed the accuracy of all citations. Any minor transcription errors in the submission are unintentional and do not impact the validity or interpretation of the cited results.
>
> ---
> C3. **The text contains *main typo's* or grammatical mistakes. Sometimes the text refers to the Appendix, but it is not clear to which section in the Appendix. In Appendix A.2 Theorem A.3 is stated, which I believe is a copy of Theorem 2.5, but Theorem A.2 is never referred to anymore.**
>
> We appreciate the reviewer's feedback regarding the presentation. Although a thorough review of our text revealed no major typos or grammatical errors, we have implemented improvements as previously detailed, including enhancing the figures (especially Figure 1), refining the overall text and explanations, and addressing all minor issues.
>
> The reviewer appears to be mistaken about the reference to Section A.2, which is explicitly cited in line 189: "Proof of Theorem 2.5 can be found in appendix Section A.2."
>
> ---

---

> ### Author Response · Authors · 2025-11-27
> **Response (Part 3)**
>
> Q1: **The Tables 3 and 4 you are working with "universal" DNNs, which seem to be some kind of projections onto the universal subspace of parameters. How does this work precisely? For example, how are the dimensions of the universal subspace determined? Is this done per layer of the DNN? And which model is projected onto this subspace?**
>
> The details have been provided in Section 3.3.1, along with the referenced tables. Line 449 states: "we reuse the shared principal directions and learn only their task-specific coefficients for the new task. Learning these low-rank coefficients is substantially cheaper than optimizing full-rank weights of size, reducing both computation and memory." This approach is analogous to learning or finding coefficients for a set of fixed basis vectors or principal components.
>
> For GLUE (Line 475), we initialize our universal subspace using a leave-one-out setup, where the subspace is calculated using components from all but one LoRA adapter for which the coefficients are subsequently learned. For image classification (Line 477), we utilize publicly available ViT LoRAs to extract our universal subspaces, ensuring that the data used for any of these pretrained LoRAs has not been seen during the coefficient training. Further implementation details are referenced in Appendix Section C.
>
> The dimensions of the layer-wise universal subspace can be determined visually or by using an eigenvalue threshold (refer to the discussion in Appendix Section B). Yes, the principal components are defined layer-wise, as consistently mentioned throughout the paper (abstract, introduction, theory, analysis, and appendix sections).
>
> The reviewer's question, **"And which model is projected onto this subspace?,"** is somewhat unclear. The models discussed in Section 3.3.1 are initialized using the joint subspace extracted from pretrained models, excluding the data/model on which they are subsequently trained. The Principal Components (PCs)/basis are then frozen, and only the coefficients are tuned on an out-of-distribution (OOD for the PCs) before unseen data (explained in line 454). If our low-rank universal subspace hypothesis were incorrect, the model would fail to learn, as all basis vectors are frozen, requiring a new set of basis for the new data. This is particularly relevant since the Universal subspace is extracted from a small number of models. However, our experiments demonstrate that even this limited Universal subspace is adequate for achieving good results, further validating the Universal subspace property and showcasing an application of efficient learning.

---

### Official Review · Reviewer_pdW6 · 2025-11-05

**Soundness:** 2
**Presentation:** 3
**Contribution:** 2
**Rating:** 4
**Confidence:** 4

**Summary:**

This paper first provides large-scale empirical evidence demonstrating that neural networks share similar low-dimension parameteric subspaces regardless of initialization, task, or domain. By projecting the parameter space to the low-dimension subspace, it is feasible to save storage and training consumption in future research.

**Strengths:**

1. The first to provide large-scale empirical study to demonstrate same neural networks share similar subspace parameter space.
2. Experiments are conducted among many model series, such as LLaMA, Mistral and etc.
3. The Universal Weight Subspaces for the same series models takes less storage but show comparable performance, which means it is feasible to use just one model for many tasks.

**Weaknesses:**

1. The paper claims universality “…neural networks…regardless of initialization, task, or domain” in abstract, but all experiments use models within the same architecture families (e.g., ResNet, ViT, LLaMA/Mistral) under the same domain tasks (different datasets). Thus, the evidence supports within-architecture similarity rather than true cross-domain or cross-architecture universality, making the claim overstated.

2. The study employs only a single tensor decomposition method (HOSVD), and the observed phenomena may strongly depend on this choice. Additional experiments using alternative high-order decompositions (e.g., HOOI) are needed to confirm that the findings are not artifacts of the specific method used.

3. In Section 3.2.1, the authors evaluate subspace expressiveness on IID and OOD tasks but do not specify what those tasks actually are. Similarly, Table 1 lists “Style 1 – Style 10” without any explanation, leaving the experimental setup and interpretation unclear.

4. In the ViT experiments, the authors skip the first and last layers during subspace decomposition, while other models do not, yet no justification or comparison is provided—would including these layers harm performance? Moreover, Table 2 does not specify what the IID and OOD tasks are, leaving the experimental design under-explained.

5. Each experiment relies on a large number of models per architecture, yet the relationship between model count and subspace effectiveness is not examined. It would be important to determine whether there exists a saturation point beyond which adding more models yields diminishing contributions to the shared subspace.

**Questions:**

See Weaknesses.

---

> ### Author Response · Authors · 2025-11-27
>
> **Concern 1: "Claims universality regardless of initialization, task, or domain, but experiments only within shared architecture families"**
>
> The paper's abstract (and later introduction and other sections) explicitly define this scope. Line 22 (Abstract) states the work focuses on "within shared architectures across diverse tasks and datasets." Section 3 (line 201) explains that cross-architecture comparison is currently infeasible, clarifying that "there is no current method that enables us to compare subspaces of models with different architectures". Line 59 (Introduction) now also specifies this. Cross-architecture subspace comparison, for our work, requires mathematical advancements that are yet to materialize. If any such mathematical tools exist, please cite them, and we will gladly add the analysis for dissimilar architectures.
>
> We are unsure what the reviewer means by "domain" - could you please clarify? Our results demonstrate the Universality property's applicability to IID and OOD setups, showing that data from distinct domains, such as medical and remote sensing, can exist in the same low-rank subspace.
>
> Rather than overstating our claims, we have carefully circumscribed universality. This rigorous definition is a novel and scientifically precise aspect of our work, not a limitation.
>
> ---
> C2:**: "Single decomposition method (HOSVD) - phenomena may depend on this choice"**
>
> We have added HOOI results in Table 4, and observe similar results, showing that the phenomenon is not an outcome of HOSVD formulation.
>
> ---
> C3: **"IID/OOD tasks in Section 3.2.1 not specified; Table 1 lists Style 1-10 without explanation"**
>
> For IID/OOD tasks shown Figure 4, refer to Appendix B.1. For Table 1, the style names were not added due to space constraints (they are mentioned in Figure 5) - they have been added to Appendix Section B1 (last paragraph). For a list of all tasks in Table 1, refer to Table 7 in Appendix.
>
> ---
> C4: **"ViT first/last layer omission - no justification or comparison provided"**
>
> Line 429 and Appendix B (lines 947-950) provides explicit justification: "We do this universal subspace analysis on all weight parameters in every neural network layer except the first and last neural network layer. This is because these layers may differ across models due to differences in input shapes and types, loss functions, and the tasks being trained."
>
> This is a principled choice: embedding layers (shape/input mismatch) and classification heads (task-specific outputs) are naturally excluded. There is no coherent mathematical tool to compare such different subspaces for our analysis.
>
> For Table 2 (now Table 3), as explained in Appendix B2. (lines 1378-1383), the IID and OOD tasks are chosen randomly from the list of tasks (for 5 independent runs) provided in Appendix Table 8, therefore there are no determinate task names. For determinate results of Image Classification (as this one) with task names, please refer to Figure 2a, Table 2 and Table 5.
>
> ---
> C5: **"Relationship between model count and subspace effectiveness not examined - saturation point unknown"**
>
>
> This is implicitly addressed through our large-scale experiments (500 ViTs, 500 Mistral-7B and 300 Stable Diffusion LoRAs, 50 LLaMA3-8B, 177 GPT-2s, Flan-T5, and ResNet50 models) in Figure 1 to Figure 12, and Tables 1 - 5, which demonstrate consistent behavior at different scales. Theorem 2.5 provides insights on the saturation dynamics where we see that the rate of convergence of the shared subspace to the true subspace is in the order \(O(1/T)\), where T is the number of tasks, indicating increasingly effective coverage as T increases. In practice, the minimum number of models per architecture needed to achieve saturation point depends on the quality of the trained models, the diversity of data they have been trained on, and on the architecture itself. Ablating these would require access to all the data for all the models, and very careful training on every training for each data, and then running permutation with all possible combinations of models. All of this is out of reach for most researchers simply due to time, data and compute constraints.
>
> We, however, do provide one ablation for the Mistral-7B experiment in Appendix B3. For 9 randomly chosen OOD tasks, we randomly sample an increasing number of models from Table 7 and analytically reconstruct the OOD models by projecting them into the extracted Universal subspaces. These reconstructed models are then evaluated on the OOD tasks and compared with models directly finetuned on those OOD tasks. We observe that an adequate set of principal components is quickly extracted, achieving performance close to finetuned performance starting around 100 model samples.

---

### Author Response · Authors · 2025-12-01
**General Comment**

We thank the reviewers for their time and valuable feedback. We note that while many concerns raised were already addressed in depth within our paper, we have uploaded a **revised draft** that addresses **all** raised concerns through new experiments, ablations, and extensive textual refinements.

**We find that all reviewers agree on the novelty and experimental depth of our work.**

As a few reviewers (pdW6, rMn9, g5Q1) were confused about the contribution and claims of our work, we'll restate it here (from our Introduction): **"We show that backpropagated neural networks trained on a variety of datasets - which could be disjoint and unrelated - diverse hyper-parameter settings, initialization and regularization methods, often learn an architecture-specific, layer-wise similar, low-rank joint subspaces (dubbed the Universal Subspace). We provide the first large-scale empirical analysis - across a diverse set of models - that neural networks tend to converge to these joint subspaces, largely independent of their initialization or the specific data used for training."**

We'd like to note there has been no prior work on this specific topic, and few, if any, works in similar mechanistic interpretability or analysis have provided the experimental breadth and depth that we have.

**Addressing Readability and "Density"**
We acknowledge that this paper is dense, a byproduct of the scale of our experimental analysis (spanning ViTs, LLMs, Diffusion models, and CNNs). However, we respectfully note that this is purely subjective, as although some reviewers found the density challenging, others (c4a8) noted: "This paper is easy to read and follow, given its systematic build-up."

We have, nevertheless, put substantial effort into improving readability and have significantly updated our draft:

- **Completely Overhauled Figure 1**: We provide a comprehensive visual summary of the findings to immediately anchor the reader. Other main text figures have also been updated.
- **Added "Highlights"**: We have bolded and highlighted key takeaways throughout the text to guide the reader through the dense results.
- **Clarified Nomenclature**: We have refined our definitions of "universality" (scoped to shared architectures). The algorithm and all major terms and experimental setups have been further elaborated, cited, and refined.

This paper uncovers an **intrinsic property of deep learning optimization**. The implications are profound:

- **Efficiency**: Compressing 100s/1000s of models into a single subspace model (up to 100x memory reduction).
- **Sustainability**: Drastically reducing the carbon footprint of model finetuning and inference.
- **Theory**: Providing empirical grounding for why overparameterized models generalize and transfer so well.
*And more.*

**Summary of Key Improvements & New Experiments**
We have ameliorated all specific technical concerns raised by the reviewers:

 - **Dependency on Decomposition Method (Reviewer pdW6)**: We added results using HOOI (Higher-Order Orthogonal Iteration) in addition to HOSVD. The results (Table 4) are consistent, proving the phenomenon is a fundamental property of the weights, not an artifact of the decomposition method.

- **Saturation & Model Count (Reviewer pdW6)**: We added an ablation study (Appendix B.3, Table 12) demonstrating the relationship between the number of models and subspace effectiveness. We show that the subspace generalizes quickly, with performance saturating after a certain number of models.

- **Baselines & Comparisons (Reviewer g5Q1, c4a8)**: We added Table 2, comparing our subspace merging against SOTA model merging techniques, showing superior performance with significantly fewer parameters.

- **Clarifications (Reviewer rMn9)**: We have fixed the bibliography errors, clarified Algorithm 1 (notations), and explicitly defined the IID/OOD tasks in the Appendix.

We respectfully urge the Area Chairs to evaluate this work based on its **scientific hypothesis and empirical evidence**, rather than subjective preferences regarding presentation style or minor typos (which have now been resolved). We believe this work offers a novel perspective that **"might indeed change the way we look at neural networks"** (Reviewer c4a8).

---

### Meta-Review · Area_Chair_WPQA · 2025-12-14

**Summary:**

The reviewers’ concerns focused primarily on clarity and presentation, the scope and precision of the paper’s central claim of “universality,” the relationship to prior work, and the interpretation and fairness of the experimental comparisons. The authors responded with extensive clarifications, additional experiments, revised figures, corrected citations, and a more explicit definition "universality". While the experimental breadth is monumental, the paper could be improved by more clearly highlighting its findings, structuring the mathematical and empirical results in a more digestible way, better explaining their meaning, and further streamlining the presentation to improve accessibility.

**Reviewer Concerns:**

Addressed

* Concerns that the paper overstated “universality” were addressed by explicitly redefining the claim, clarifying scope, introduction, and discussion
* Questions about dependence on a single decomposition method were addressed by adding results using an alternative tensor decomposition
* Multiple concerns about unclear experimental setup, notation, and algorithms were addressed through revisions
* Requests to examine the relationship between the number of models and subspace quality were addressed through an ablation study and discussion
* Criticism regarding missing or incorrect citations was addressed by correcting bibliography errors
* Concerns about fairness of comparisons were addressed by clarifying the experimental protocol and by adding comparisons
* Questions regarding how inference and learning are performed were addressed with explicit explanations

Unresolved

* Clarity and framing can still be improved
    + The current revision is rather minor in its form, and it's unclear whether this addresses the reviewers' concerns, especially the point of "overselling" the results
    + The plots are not easily readable (axis labels too small, too many plots, unnecessary background coloring, unclear within-plot annotations, etc), often unclear, and visually not appealing
    + The beginning of the paper is dense, with technically heavy mathematical results, which is particularly not reader-friendly
* The degree of novelty relative to prior work remains a point of interpretation

**Reviewer Scores:**

* Reviewer pdW6: Likely unchanged or slightly higher, as all specific technical concerns (claim scope, decomposition dependence, experimental clarity, saturation behavior) were directly addressed with new experiments and clarifications.
* Reviewer rMn9: Likely modestly higher, given extensive clarification of notation, algorithms, experimental setup, and correction of citation issues, though concerns about overall clarity and hypothesis formulation may partially remain.
* Reviewer g5Q1: Likely modestly higher, as questions about novelty, fairness of comparisons, and interpretation of results were directly addressed, though some conceptual skepticism about the core claim may persist.
* Reviewer c4a8: Likely unchanged or slightly higher, as their main concerns regarding scope of universality, relation to prior work, and downstream task explanations were directly addressed with clarifications, added citations, and expanded methodological details.

---

### Decision · Program_Chairs · 2026-01-26

Reject